# OPUS-DSD: deep structural disentanglement for cryo-EM single-particle analysis

Zhenwei Luo[1,2,3], Fengyun Ni[1], Qinghua Wang [4] & Jianpeng Ma [1,2,3]✉

Cryo-electron microscopy (cryo-EM) captures snapshots of dynamic macromolecules, collectively illustrating the involved structural landscapes. This provides an exciting opportunity to explore the structural variations of macromolecules under study. However, traditional cryo-EM single-particle analysis often yields static structures. Here we describe OPUS-DSD, an algorithm capable of efficiently reconstructing the structural landscape embedded in cryo-EM data. OPUS-DSD uses a three-dimensional convolutional encoder–decoder architecture trained with cryo-EM images, thereby encoding structural variations into a smooth and easily analyzable low-dimension space. This space can be traversed to reconstruct continuous dynamics or clustered to identify distinct conformations. OPUS-DSD can offer meaningful insights into the structural variations of macromolecules, filling in the gaps left by traditional cryo-EM structural determination, and potentially improves the reconstruction resolution by reliably clustering similar particles within the dataset. These functionalities are especially relevant to the study of highly dynamic biological systems. OPUS-DSD is available at https://github.com/alncat/opusDSD.

Macromolecules are dynamic machines that use specialized motions to carry out their functions[1]. Structural biology aims to determine the three-dimensional (3D) structure of macromolecules to high resolution, and at the same time understand their functions by reconstructing the underlying dynamics. Cryo-electron microscopy (cryo-EM) single-particle analysis (SPA) is a powerful method for obtaining high-resolution 3D structures[2–4]. Conventional cryo-EM SPA reconstruction often produces only a single static 3D model. However, the large number of snapshots captured by cryo-EM preserves a huge amount of conformational and/or compositional heterogeneity that may be functionally important. Moreover, the flexibility of 3D macromolecules is a major bottleneck to the achievement of high resolution in cryo-EM SPA[2]. Therefore, powerful analysis methods are needed to reliably recover structural heterogeneity and help improve the resolution of cryo-EM reconstruction for macromolecules.

The conventional tool for resolving heterogeneity in cryo-EM datasets is 3D classification[5,6], which models different conformations as individual 3D volumes. However, 3D classification scales poorly with respect to the number of conformations, and falls short of resolving structural dynamics composed of a large number of transitional conformations. Existing approaches such as multi-body refinement in RELION[7], and 3D variability analysis in cryoSPARC[8] model continuous conformation changes using linear combinations of reaction coordinates. The multi-body refinement in RELION is particularly effective at modeling large-scale conformational dynamics of rigid-body components, yielding high-resolution structures for otherwise unresolved densities. It becomes less effective, however, for macromolecular systems in which dynamic structural components do not move as rigid bodies.

Recently, deep learning has emerged as a viable solution for handling structural heterogeneity. There are a number of deep learning approaches to probe structural heterogeneity, including Multi-CryoGAN[9], e2gmm[10] and 3DFlex in cryoSPARC[11]. Multi-CryoGAN is based on a generative adversarial network and its validity has been demonstrated using synthetic data. E2gmm represents the 3D structure using a set of Gaussians, while 3DFlex uses a neural network to fit the 3D

[1]Multiscale Research Institute of Complex Systems, Fudan University, Shanghai, China. [2]Zhangjiang Fudan International Innovation Center, Fudan University, Shanghai, China. [3]Shanghai AI Laboratory, Shanghai, China. [4]Center for Biomolecular Innovation, Harcam Biomedicines, Shanghai, China. ✉e-mail: jpma@fudan.edu.cn

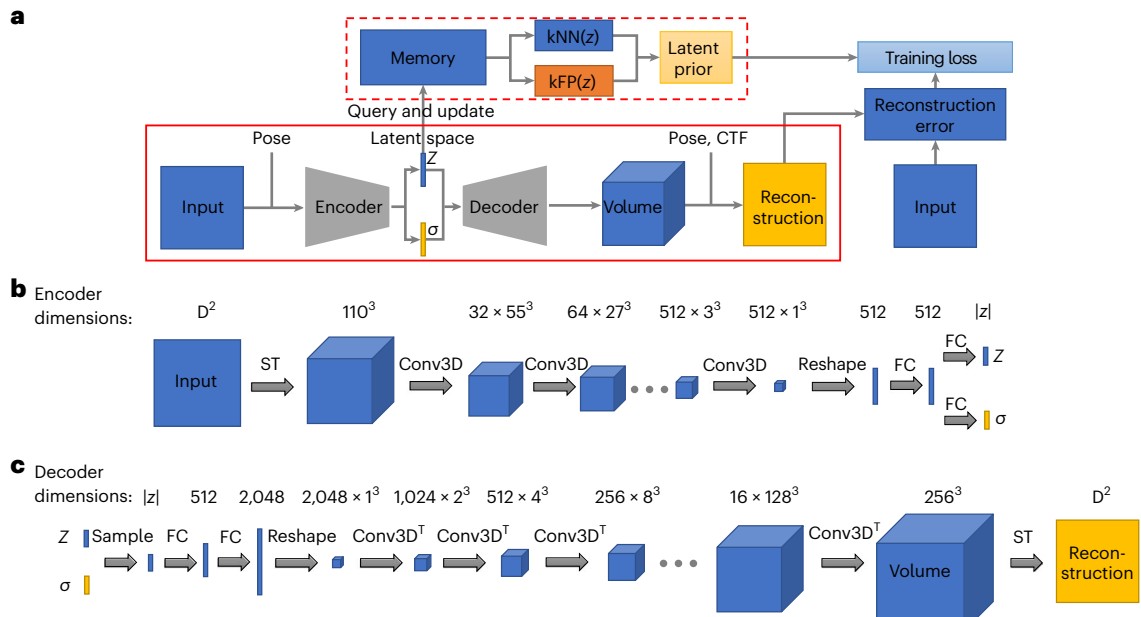

**Fig. 1 | Architecture of OPUS-DSD. a**, Schematic diagram of the architecture. Pose refers to the projection direction of input with respect to the consensus model. For simplicity, the standard Gaussian prior for latent distribution as well as the smoothness and sparseness priors for the 3D volume are omitted in this chart. **b**, Architecture of the encoder of OPUS-DSD. This diagram shows the encoder that translates a 2D cryo-EM image into the latent encoding. The top row denotes the dimensions of the intermediate tensors. The arrow links the input and output of the operation above. FC, fully connected layer; Conv3D, 3D convolution; ST, spatial transformer (which back-projects the 2D image to a 3D volume). The number of channels of the convolution kernel can be derived from the dimensions of its input and output. The ellipsis represents the repeating of the preceding operation until the tensor reaches the output dimension. All convolutions and fully connected layers except the last one had LeakyReLU[33] (leaky rectified linear unit) non-linearity with a negative slope of 0.2. **c**, Architecture of the decoder of OPUS-DSD. This diagram shows the decoder that translates the latent encoding $z$ into a reconstructed 2D projection. Conv3D[T], 3D transposed convolution; ST, spatial transformer, which here renders the 3D volume into the 2D image of desired resolution. All transposed convolutions except the last one and fully connected layers have LeakyReLU non-linearity with a negative slope of 0.2.

displacement field of each particle. By contrast, a neural network-based method, cryoDRGN, uses two-dimensional (2D) images as input, and the particle poses are determined by consensus refinement[12]. This method adopted a neural representation for 3D structures and leveraged a variational autoencoder (VAE)[13,14] to train the neural network that translates 2D images to 3D volumes end-to-end.

In the 2D image to 3D structure translation formulation as implemented in cryoDRGN[12], the landscape of 3D structures is in a low-dimension latent space. Translating a 2D image to a 3D structure is equivalent to mapping the 2D image to the encoding of its underlying 3D structure in latent space using neural networks. Both the encoding of the 2D image to latent code and the decoding of the latent code to a 3D structure are performed by neural networks. The continuous transformation between 2D image and 3D structure can then be learned end-to-end. Despite the use of neural networks, recovering the landscape of 3D structures using only 2D images is still inherently ill-posed, that is, the unknown 3D structures are of much higher dimensionality than the available 2D data. Furthermore, the pose parameters determined by consensus refinement are provided only for resolved densities.

The smoothness of latent space is critical for both resolving continuous structural heterogeneity and reliably recovering the distribution of conformations in cryo-EM datasets. A smooth latent space ensures that the neural network can generate continuous conformations when traversing the latent space, and guarantees the clustering of images with similar structures. Popular methods to encourage the smoothness of latent space are the VAE and its extensions, including β-VAE[13,14], which apply variational Bayesian learning to the distribution of latent variables (more details can be found in 'Training objectives' subsection in Methods).

In this study, we present a neural network-based method, OPUS deep structural disentanglement (OPUS-DSD), for 3D structural heterogeneity analysis. Built upon cryoDRGN1.0 (ref. 12), OPUS-DSD incorporates a large number of methodological improvements, including the use of 3D convolutional architecture, that is, neural volumes[15–17], and latent priors to encourage the smoothness of the latent space. By systematic testing on synthetic and real cryo-EM datasets, we demonstrate the performance of OPUS-DSD in resolving structural heterogeneity, even at lower signal-to-noise ratios. OPUS-DSD not only reconstructs the large-scale continuous dynamics, but also shows the associated compositional changes. By effectively reducing the level of structural heterogeneity in selected particles, OPUS-DSD also improves structural determination in cryo-EM SPA reconstructions.

## Results

### Design of OPUS-DSD

OPUS-DSD is designed to elucidate the structural heterogeneity in a cryo-EM dataset using 2D images as input. Overall, it contains two primary components: an encoder–decoder network to convert the 2D cryo-EM image to its corresponding 3D structure (solid red box in Fig. 1a), and a prior in latent space that facilitates the encoding of structural information in 2D images (dashed red box in Fig. 1a).

The encoder–decoder network works as follows. The encoder takes a 2D cryo-EM image as input, and estimates the distribution of its associated latent code $z$ in a low-dimension space $\mathbb{R}^d$. The decoder then produces a 3D volume by sampling a latent code from the distribution estimated by the encoder (Fig. 1a). The latent code is assumed to follow a Gaussian distribution, for which the mean $z \in \mathbb{R}^d$ and standard variation $\sigma \in \mathbb{R}^d$ are estimated by the encoder[14]. The 3D volume is represented as a discrete grid of voxels, $V(\boldsymbol{x})$, where $\boldsymbol{x} \in \mathbb{R}^3$ is a point

in 3D space. This explicit 3D voxel grid $V(x)$ is then transformed into a 2D reconstruction with specified pose and contrast transfer function (CTF) parameters according to the differentiable cryo-EM image formation model. The neural architecture of OPUS-DSD can then be trained end-to-end by reconstructing each input image and minimizing the squared reconstruction error.

In addition, we incorporated a latent prior to promote a specific type of geometry in latent space (Fig. 1a). The structural heterogeneity is resolved within the encoder–decoder architecture of OPUS-DSD by integrating structural information into the latent space. That is, the variation of latent codes should primarily correspond to the variation in 3D structures. This kind of latent space specifies a geometry in which the similarity between 3D structures is proportional to the distance between their latent codes. In other words, the latent codes of similar 3D structures are clustered together, while the latent codes of distinct 3D structures are far apart. This not only ensures the smoothness of the latent space but also enhances the correlation between latent codes and structural variations. The formulation of this multi-component latent prior is detailed in Methods. For a given latent code, the latent codes of similar 3D structures can be identified as its $k$ nearest neighbors (kNN), whereas those of different 3D structures can be found as its $k$ furthest points (kFP) (Fig. 1a). The latent prior is implemented by querying the kNN and kFP from a dynamically updated memory bank that stores the latent codes of all images. Furthermore, we implemented the reconstruction loss to guide the neural network to generate an ensemble of 3D structures congruent with the cryo-EM dataset. Concurrently, the latent prior refines the geometry of the latent space, providing a regularizing effect that enables the neural network to better capture the structural dynamics.

The encoder network converts the 2D input to latent vector $z$ by going through a series of intermediate representations (shown as different shapes in Fig. 1b). The imputation of pose information to the 2D projection is performed by back-projection as follows. The 2D projection of size $D^2$ (a square) is first converted into a pseudo 3D volume (first cube) by repeating along the $z$ axis $D$ times. Given that the pose of the pseudo 3D volume remains the same as the original 2D projection, a spatial transformer module[18] is introduced to transform the pseudo volume into the canonical pose, matching the consensus model. The spatial transformer enables neural networks to disentangle object pose from shape and texture in images, thereby realizing spatial invariance[18]. In OPUS-DSD, the spatial transformer aligns pseudo volumes with the consensus model by performing rotations using predetermined poses, thereby enabling different pseudo volumes with varying poses to be brought back to the same canonical pose, aiding the encoder to distinguish structural heterogeneities from pose variations. It is worth noting that the spatial transformer in this context is not trainable. Subsequently, the aligned pseudo 3D volume is down-sampled to a $1^3$ cube with 512 channels using six consecutive strided convolutions[16] with a kernel size of 4. Then the $1^3$ cube with 512 channels is flattened into a 512-dimension vector (first rectangle) that is transformed into the distribution of latent code by estimating the mean $z$ and standard deviation $\sigma$ via the application of two fully connected layers with non-linearity.

To ensure the smoothness of the 3D density maps generated by the neural network and avoid overfitting[3,19], the decoder in OPUS-DSD uses a convolutional architecture, neural volumes[17], that converts the latent vector to a smooth 3D volume (Fig. 1c). First, a series of fully connected layers with non-linearity are applied to transform the sampled latent vector (second rectangle) into a 2,048-dimensional representation, resulting in a $1^3$ cube with 2,048 channels. Then the $2,048 \times 1^3$ cube is up-sampled into a $1 \times 256^3$ 3D volume (last cube) using eight consecutive transposed convolutions[16]. The kernel size is 2 for the first two convolutions and 4 for the remaining convolutions. The 2D reconstruction (square) is generated from the $256^3$ volume by the spatial transformer with predetermined pose and CTF parameters[18].

Although only the process to generate a $256^3$ volume from a latent code is shown, the decoder can produce a 3D volume in any size by changing the intermediate tensors.

Given that OPUS-DSD is designed to improve on cryoDRGN 1.0 (ref. 12), the performance of these two methods is compared using synthetic and real datasets. For simplicity, cryoDRGN 1.0 is referred to as cryoDRGN hereafter.

### Pre-catalytic spliceosome

OPUS-DSD was used to analyze the structural heterogeneity of the pre-catalytic spliceosome (EMPIAR-10180, ref. 20) to enable direct comparison with other methods[7,10,12]. We trained a 12-dimensional latent variable model using the consensus refinement results deposited in EMPIAR-10180. Uniform manifold approximation and projection (UMAP) visualization[21] of the latent space of OPUS-DSD shows 20 classes using the Kmeans algorithm (Fig. 2a). By reconstructing the density maps using the cluster centers as input for the decoder of OPUS-DSD, those clusters were found to represent different combinations of spliceosome subcomplexes, namely core, foot, helicase and SF3b, as previously defined[7] (Fig. 2b and Extended Data Fig. 1a). For instance, class 12 represents the spliceosome with all four expected subcomplexes, class 10 shows a spliceosome with an additional U2 core[22], while class 18 has only the core and helicase, and class 14 lacks SF3b. Similar compositional heterogeneities were also detected by cryoDRGN[12] but not by e2gmm[10]. The densities for each of the four subcomplexes in these structures agreed very well with their ground-truth high-resolution structures obtained by multi-body refinement[7] (Extended Data Fig. 1b). Furthermore, traversal along the first principal component (PC1) in principal component analysis (PCA) of the latent space showed distinct conformations. The structures at the two ends of PC1 correspond to two opposite conformations. The structure at the negative end of PC1 shows an 'open' conformation in which SF3b stays on the top of the core and the regions highlighted in the yellow ellipse and circle are far apart (Fig. 2c). By contrast, the structure at the positive end of PC1 shows a 'closed' conformation in which SF3b is almost folded onto the core, the regions enclosed in the yellow ellipse and circle come together in space, and a chain appears in the yellow ellipses connecting helicase and SF3b. The emergence of this chain was not previously reported. Traversal along the PC1 in Supplementary Video 1 shows the continuous dynamics between the open and closed conformations, highlighting the synchronous movements of the subcomplexes, that is, the helicase bends towards the foot while SF3b folds onto the core. Moreover, displacement of SF3b between these two conformations was observed (Fig. 2c, green arrows).

### *Plasmodium falciparum* 80S ribosome

OPUS-DSD was used to analyze the structural heterogeneity of the *Plasmodium falciparum* 80S (*Pf*80S) ribosome (EMPIAR-10028, ref. 23), which was studied by cryoDRGN and multi-body refinement[7,12]. We trained a 12-dimensional latent variable model using the consensus refinement results from RELION 3.1. The UMAP visualization of the latent space of OPUS-DSD showed clusters with a greater degree of separation (Fig. 3a) than that of cryoDRGN[12]. Furthermore, we obtained 20 classes after Kmeans clustering in latent space and reconstructed their density maps by supplying the cluster centers to the decoder of OPUS-DSD. Classes 6, 7, 8, 9, 15 and 16 all harbor an RNA strand in the region highlighted by a red ellipse, which is absent in other classes (Fig. 3b). The existence of this heterogeneity was confirmed by comparing class 8 with the consensus model at a lower contour level (Extended Data Fig. 2), and was reported by cryoDRGN[12] as well. Traversal along PC1 in the latent space uncovered distinct conformations and their interconnections (Supplementary Video 2). The structures at the positive or negative end of PC1 are denoted as the positive or negative conformations, respectively, with the red arrows indicating the relative displacement between them (Fig. 3c,d). Following the division of the

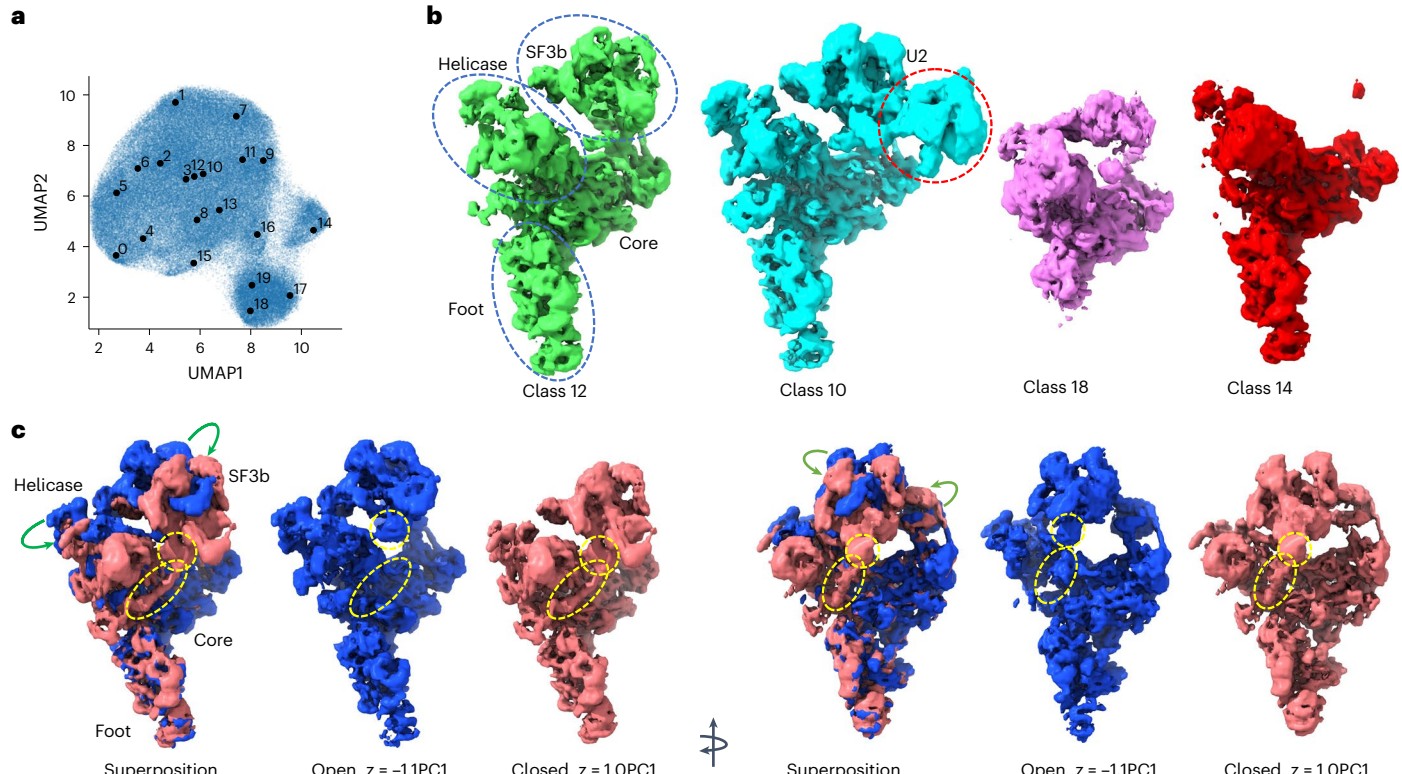

**Fig. 2 | Heterogeneity analysis on the pre-catalytic spliceosome (EMPIAR-10180). a**, UMAP visualization of the 12-dimensional latent space of all particles encoded by OPUS-DSD. Solid black dots represent the cluster centers for labeled classes. **b**, Conformations with major compositional heterogeneities determined using OPUS-DSD. The density maps were generated by the decoder of OPUS-DSD with the corresponding class centers shown in **a**. Class 12 is the expected structure of the pre-catalytic spliceosome, class 10 is a complete structure with the U2 core, class 18 only has the core, and class 14 lacks the SF3b subcomplex. **c**, Open and closed conformations of the spliceosome generated along the PC1 of the latent space. The open and closed conformations were generated at the PC1 with an amplitude of −1.1 and 1.0, respectively. The superposition shows the displacement of SF3b and helicase between these two conformations as indicated by the green arrows. The yellow dashed ellipses highlight the occupancy differences of a chain between helicase and SF3b in different conformations. The yellow dashed circles mark the domain in SF3b that connects to this chain. All maps were contoured at the same level and visualized using ChimeraX[34].

$Pf$80S ribosome into three subunits, head, large subunit (LSU) and small subunit (SSU), in the published multi-body refinement[7], the two sides of the LSU move in opposite directions. In the positive conformation there is an RNA (marked by the yellow ellipse) connecting the SSU and LSU, which becomes absent in the negative conformation (Fig. 3c). A series of intermediate structures along the PC1 in Supplementary Video 2 show how this RNA slides out of its original site in the SSU and docks onto the LSU. This kind of visualization is more difficult for multi-body analysis because it involves a series of small-scale occupancy changes during transition. In addition, at the back of the $Pf$80S ribosome a peripheral RNA strand gradually moves closer to its binding site on the LSU in the positive conformation and forms a new link (marked by the orange ellipse) with the LSU, while the head and LSU perform coordinated movements during this transition (Fig. 3d and Supplementary Video 3).

### *Saccharomyces cerevisiae* 80S ribosome

OPUS-DSD was used to analyze the structural heterogeneity of a smaller dataset of the *Saccharomyces cerevisiae* 80S (*Sc*80S) ribosome containing 60,363 images (EMPIAR-10002, ref. 24). We trained an eight-dimensional latent space model using 90% of images in this dataset. UMAP visualization of the latent space of OPUS-DSD and cryo-DRGN showed similar clustering patterns (Fig. 4a and Extended Data Fig. 3a). However, OPUS-DSD uncovered many interesting structural heterogeneities. First, a single 60S class was identified (class 0 in Fig. 4b) uniquely by OPUS-DSD. Second, OPUS-DSD showed the movements of

an RNA strand that binds with the 60S subunit (Fig. 4b, highlighted in red dashed boxes; Extended Data Fig. 4a). This movement can also be observed in Supplementary Video 4 (which consists of a series of states along the PC1 of the latent space) and appears to be coordinated with the rotation of the 40S subunit. Traversal of the PC1 of the latent space in the cryoDRGN results also suggested this movement (Supplementary Video 5). Third, the red solid boxes in Fig. 4b mark two different binding locations of an RNA strand, which were not observed in the cryoDRGN results at this contour level (Extended Data Fig. 3b). Last, OPUS-DSD showed the large-scale displacement of the 40S subunit between different classes (Fig. 4c, Extended Data Fig. 4b and Supplementary Video 4). At the same contour level, the density map of class 9 reconstructed by the trained decoder of OPUS-DSD showed a much more complete 40S subunit than the consensus model by RELION using all particles (Fig. 4d), which also confirmed the high mobility of this region. The density map from class 8 by cryoDRGN also partially recovered the missing densities of the 40S subunit (Extended Data Fig. 3c, red ellipses) when contoured at the same level.

### Synthetic NEXT complex data

OPUS-DSD was further tested on the synthetic data of the nuclear exosome-targeting (NEXT) complex, a highly mobile system. The NEXT complex has a dumbbell shape, in which two MTR4 helicase domains[25] located at the two ends are connected by a ZCCHC8 center formed by two stranded helices. Starting from the structure of the NEXT complex

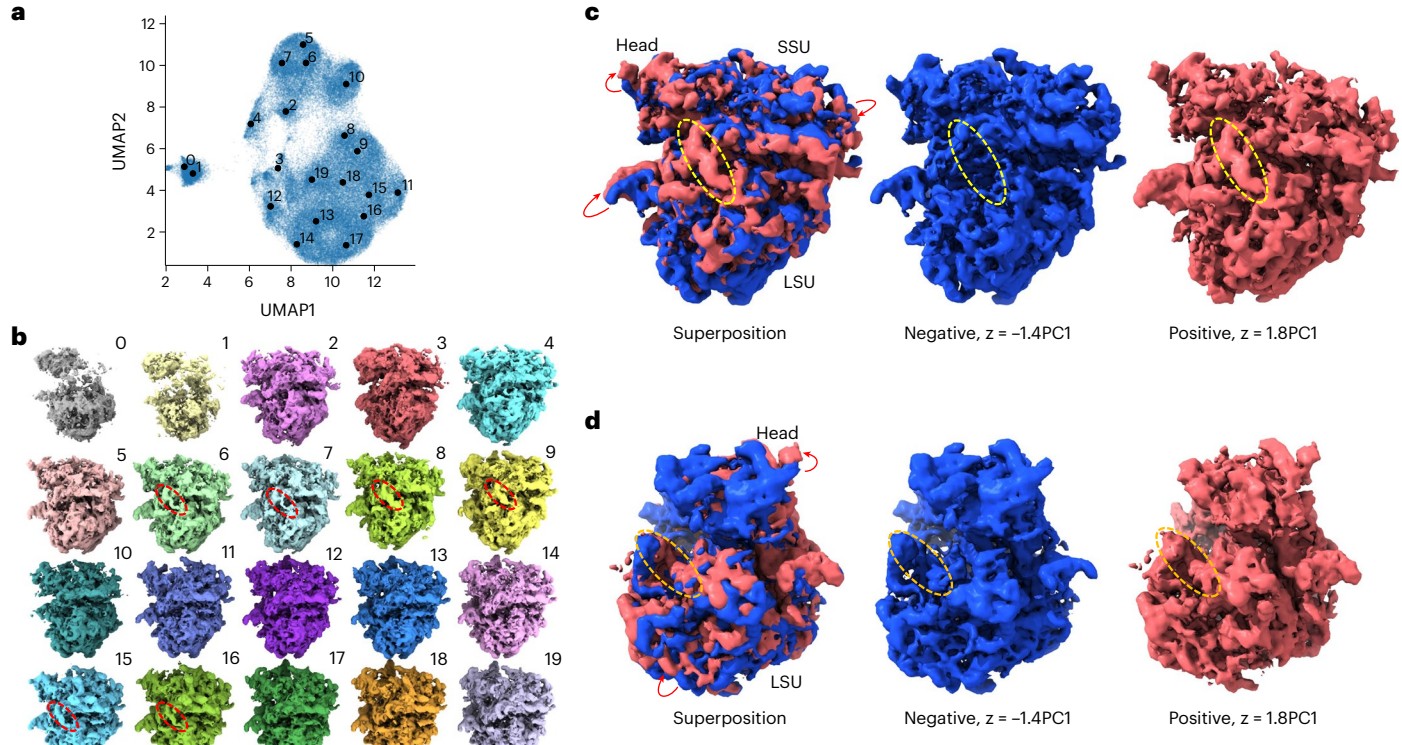

**Fig. 3 | Heterogeneity analysis of the *Pf*80S ribosome (EMPIAR-10028).**
**a**, UMAP visualization of the 12-dimensional latent space of all particles encoded by OPUS-DSD. Solid black dots represent the cluster centers for labeled classes. **b**, Twenty density maps generated by the decoder of OPUS-DSD using the cluster centers shown in **a** as inputs. Red dashed ellipses highlight the presence of a peripheral RNA between SSU and LSU in the corresponding classes. **c**, Change in occupancy of an RNA according to conformation. The negative and positive conformations were generated at the PC1 with an amplitude of −1.4 and 1.8,

respectively. Their superposition shows the displacement of different subunits as indicated by the red arrows. The yellow dashed ellipses highlight compositional differences of that RNA. **d**, Change in occupancy of a chain between a peripheral RNA and the LSU in the positive and negative conformations. The orange dashed ellipses highlight the compositional differences of that chain. Red arrows show the displacement of different subunits. All maps were contoured at the same level and visualized using ChimeraX.

determined in our own group (Fig. 5a and Supplementary Table 1), eight different conformations were generated by shifting the MTR4 helicase domains relative to the ZCCHC8 center (Fig. 5b). The synthetic data were constructed with a uniform distribution of the eight different conformations, that is, each conformation has 8,000 particles. The synthetic NEXT complex dataset with 64,000 particles yielded a consensus model using cryoSPARC (Fig. 5c).

When the signal-to-noise ratio was 0.05, OPUS-DSD and cryoDRGN differed significantly in the clustering results and 3D density maps (Fig. 5d). In the UMAP visualization of latent space, clusters of OPUS-DSD were better separated than those of cryoDRGN (Fig. 5d). Judging by the quality of the density map generated by the decoders of both methods, OPUS-DSD delivered better density maps with resolutions similar to that of the consensus model (Extended Data Fig. 5a), while cryoDRGN produced density maps with very low-resolution MTR domains and noise in the ZCCHC8 center (Extended Data Fig. 5b). OPUS-DSD identified some clusters with high accuracy (Fig. 5e). For example, class 1 recovered approximately 67% of particles belonging to conformation *f* (Fig. 5e), and the 3D reconstruction using class 1 also resembled conformation *f* (Extended Data Fig. 5c). In contrast, cryoDRGN yielded a more even distribution of the particles with high classification errors in all classes (Fig. 5f), even for class 7 with the highest dominating conformation, only 28% of particles were from conformation *f*. Using entropy as a metric, the average entropy value for clusters in OPUS-DSD (at 0.756) was lower than that for cryoDRGN (at 0.882), where the entropy of a uniform distribution of conformations was 0.903. Comparing the density maps generated by cryoSPARC using classes from either method,

the maps from OPUS-DSD were generally of higher quality than those from cryoDRGN (Extended Data Fig. 5c,d). In translation–rotation plots, the classes from cryoDRGN showed little difference and the rotations of MTR4 relative to ZCCHC8 were clustered within 10° (Fig. 5g,h, blue dots). The results of OPUS-DSD, however, showed a much wider range of structural changes (Fig. 5g,h, red dots), agreeing with the structural changes in ground-truth conformations (Fig. 5g,h, green dots). For example, the translations and rotations of MTR4 in class 1 of OPUS-DSD closely resembled conformation *f*, while classes 3 and 5 closely resembled conformations *g* and *b*, respectively (Fig. 5g,h). Hence, OPUS-DSD clearly recovered more ground-truth structural heterogeneities in this case. Furthermore, the average resolution of density maps reconstructed by classes from OPUS-DSD was also higher, suggesting its improved classification accuracy (Extended Data Fig. 5c,d).

When the signal-to-noise ratio increases to 0.1, both methods performed similarly and recovered most ground-truth conformations (Extended Data Fig. 6). However, there were still noticeable differences between the 3D density maps from their decoders (Extended Data Fig. 6g,h). The density maps from OPUS-DSD (Extended Data Fig. 6g) had a similar resolution to the consensus model across the entire complex. In contrast, the density maps from cryoDRGN were of a higher resolution than the consensus model for the central ZCCHC8 domain but a lower resolution for the MTR domains, especially in classes with larger shifts (Extended Data Fig. 6h).

The synthetic data of the NEXT complex also provided an ideal test for assessing the contributions of some of the methodological designs in OPUS-DSD. At a signal-to-noise ratio of 0.05, we found that turning

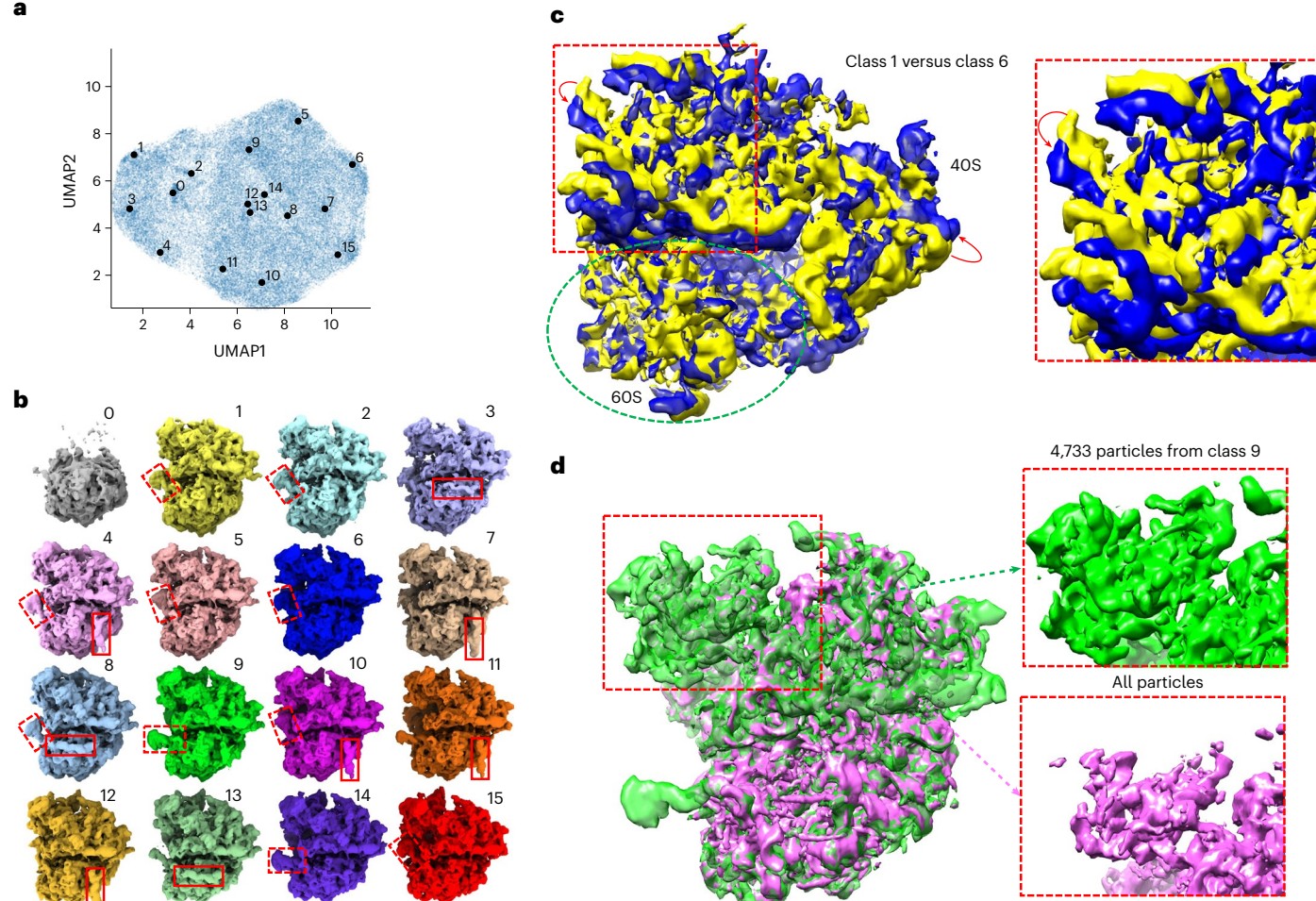

**Fig. 4 | Heterogeneity analysis of the *Sc*80S ribosome (EMPIAR: 10002).**
**a**, UMAP visualization of the eight-dimensional latent space of all particles encoded by OPUS-DSD. Solid black dots represent the cluster centers for labeled classes. **b**, Sixteen density maps reconstructed by the decoder of OPUS-DSD using the Kmeans clustering centers in its eight-dimensional latent space as inputs. The different locations of two RNA strands in different conformations are highlighted by the red solid and dashed boxes, respectively. **c**, Superimposition of class 1 and class 6 to show the displacement of 40S subunits highlighted in the red dashed boxes. Red arrows indicate the displacement directions. 60S is highlighted in a green dashed ellipse. **d**, The density map of class 9 is more complete than that reconstructed using all particles. Both maps are contoured at the same level and visualized by ChimeraX.

off the disentanglement prior tended to reduce the classification accuracy of clustering results, most significantly for classes 1, 2, 3, 5 and 7 (Extended Data Fig. 7a,b). Similar behaviors were observed when turning off data augmentation, where the classification accuracy decreased in the latent space of OPUS-DSD, most significantly for classes 2, 3, 5, 6, and 7 (Extended Data Fig. 7a,c). In addition, the presence of data augmentation decreased the validation error and improved the generalization on the *Sc*80S ribosome (Extended Data Fig. 7d,e).

**Real cryo-EM data of the NEXT complex**
Next, OPUS-DSD was tested on experimental data of the NEXT complex collected in our own group, which contained 224,354 particles after multiple rounds of 2D and 3D classifications (Supplementary Table 1 and Supplementary Fig. 1). Given that the complete NEXT complex is too dynamic to be determined to high resolution, signals of one MTR4 helicase domain were subtracted by RELION to facilitate cryo-EM SPA reconstruction[26]. Consensus refinement using the 224,354 particles of the NEXT complex produced a resolution of 5.59 Å measured using the gold-standard Fourier shell correlation (FSC) curve at 0.143 (ref. 27) (Fig. 6a). OPUS-DSD and cryoDRGN were used to analyze the structural heterogeneity by learning a latent space with the consensus refinement result (Fig. 6a).

For OPUS-DSD, the UMAP for the latent space of all particles had a bipolar distribution (Fig. 6b). Classes 0, 1 and 2 on the left side of the UMAP correspond to reconstructions in which NEXT complex densities were not complete, while classes 3–9 on the right side of the UMAP correspond to reconstructions in which NEXT complex densities were complete (Fig. 6c). In the translation–rotation plot, five reconstructions from OPUS-DSD were within 2° of rotation and 1 Å of translation, forming a cluster of 116,712 particles, while another two classes had a much larger extent of motion at rotations of 3° and 6° (classes 9 and 5, respectively) (Fig. 6e). Non-uniform refinement[28] using the 116,712 particles improved the resolution of the reconstruction from 5.59 Å to 4.45 Å (Fig. 6f and Extended Data Fig. 8a). In comparison, using cryoDRGN on the same dataset (Fig. 6b) resulted in reconstructions that all had complete NEXT complex densities (Fig. 6d). In the translation–rotation plot, one reconstruction had a large rotation and translation (class 9, ~4° and ~3 Å, respectively) while the remaining nine reconstructions (classes 0–8) were clustered within 2° of rotation and 1 Å of translation (Fig. 6g). The 200,154 particles from these nine classes (blue circle) yielded a reconstruction with a resolution of 6.12 Å in non-uniform refinement (Fig. 6h and Extended Data Fig. 8b).

We also performed focused refinement on the ZCCHC8 dimeric center, leaving the pose parameter of MTR4 helicase

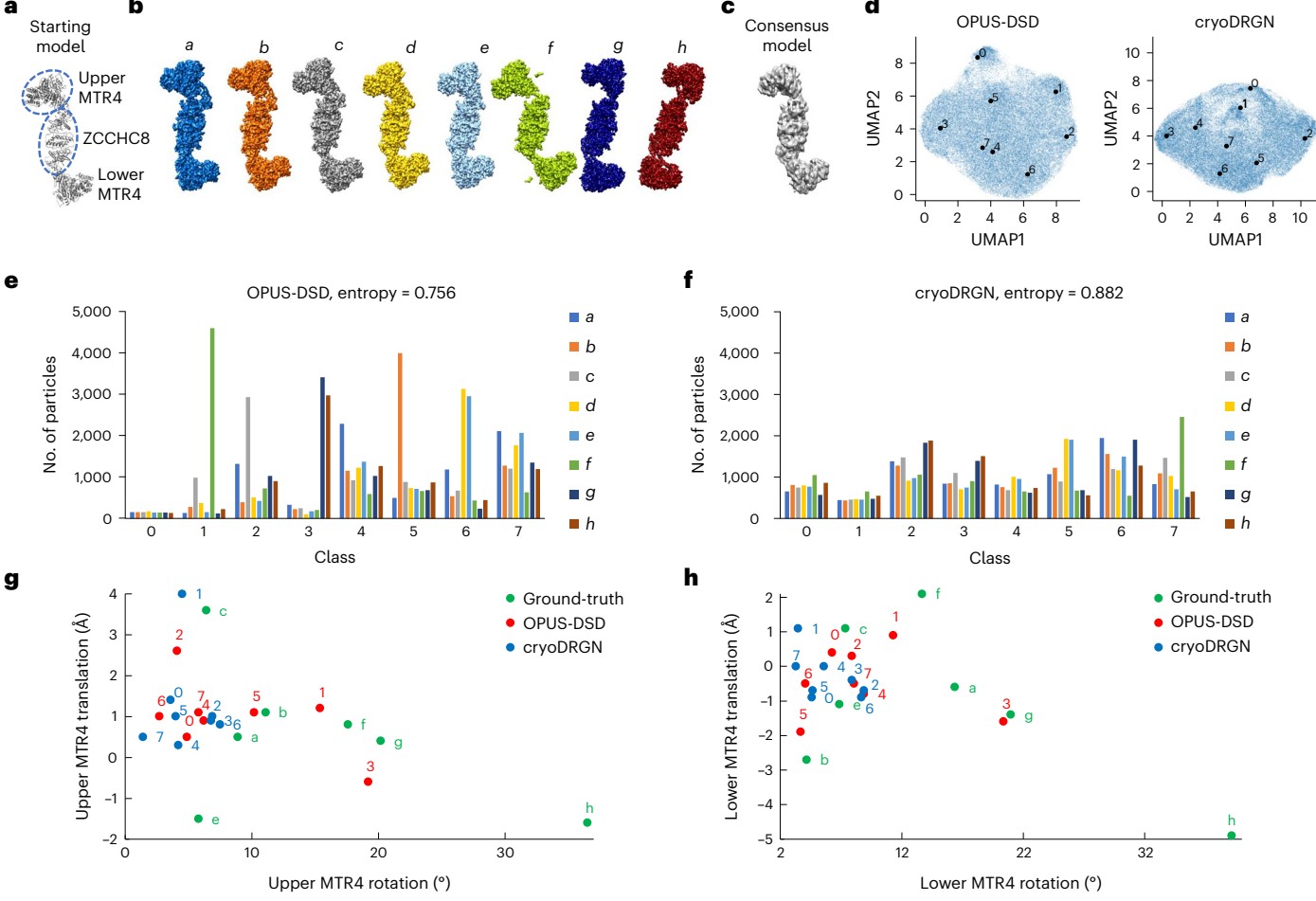

**Fig. 5 | Heterogeneity analysis of the synthetic dataset of the NEXT complex with a signal-to-noise ratio of 0.05. a**, The starting atomic model of the NEXT complex. **b**, The ground-truth density maps of the NEXT complex. Of the eight conformations, conformations *a,d,e* are similar and difficult to distinguish, and conformations *f,g,h* have larger shifts in the MTR domains. **c**, The consensus map reconstructed by cryoSPARC. **d**, UMAP visualization of the eight-dimensional latent space of all particles encoded by OPUS-DSD and cryoDRGN. Solid black dots represent the cluster centers for labeled classes, identified using the Kmeans algorithm. **e**, The distribution of particles in the clusters in the latent space of OPUS-DSD. Bars are colored as the ground-truth

density maps in **b**. Entropy is defined as the average of $-\sum_i p_i \log p_i$ for each cluster, where $p_i = \frac{n_i}{\sum_i n_i}$, and $n_i$ is the number of particles in the ground-truth conformation *i*. **f**, The distribution of particles in the clusters in the latent space of cryoDRGN. **g**, Translation–rotation plot for the upper MTR4 domain for results from both methods. **h**, Translation–rotation plot for the lower MTR4 domain for results from both methods. The movements were measured for the upper (**g**) or lower (**h**) MTR4 domains relative to the ZCCHC8 of each conformation using Dyndom[35] with the ground-truth conformation *d* as a reference.

undetermined (Extended Data Fig. 9a). The UMAP for the latent space from OPUS-DSD was bipolar (Extended Data Fig. 9b), in which classes 0–4 on the left side of UMAP correspond to complete complexes, whereas classes 5–9 on the right side of UMAP represent incomplete complexes (Extended Data Fig. 9c). By contrast, the UMAP from cryoDRGN was circular (Extended Data Fig. 9b), and nine out of 10 reconstructions showed complete densities with even particle distributions (Extended Data Fig. 9d). The translation–rotation plots from both methods (Extended Data Fig. 9e,g) showed a larger range of motion than previous analysis in Fig. 6. Classes with a small range of motion were then combined for further refinement. For OPUS-DSD, three classes with a rotation of approximately 5° (classes 2–4; 85,570 particles) were selected (Extended Data Fig. 9e), which yielded a resolution of 4.52 Å (Extended Data Fig. 9f,i). For cryoDRGN, four classes with rotations within 5° (classes 0, 4, 5 and 7; 96,466 particles) were selected for refinement (Extended Data Fig. 9g), which resulted in a reconstruction with a resolution of 5.60 Å (Extended Data Fig. 9h,j), on par with the consensus refinement (resolution of 5.59 Å) using the entire set of 224,354 particles (Fig. 6a).

It is worth noting that in the case of the NEXT complex with focused refinement, both cryoDRGN and OPUS-DSD showed the large-scale movements of MTR4 relative to ZCCHC8. The fact that cryoDRGN achieved a higher resolution in focused refinement suggests that its ability to resolve structural heterogeneity may be influenced by the degree of heterogeneity in a dataset.

We next tested a smaller dataset of 84,530 particles of the NEXT complex, which was obtained by applying several rounds of heterogenous refinements in cryoSPARC, with the best resolution of 4.39 Å (Extended Data Fig. 10a). This scenario represents the stage after conventional classification is converged.

In UMAP visualization, classes 0, 1 and 3 from OPUS-DSD were separated from classes 2 and 4 (Extended Data Fig. 10b). Classes 0, 1 and 3 all had complete density of the NEXT complex, while classes 2 and 4 were incomplete (Extended Data Fig. 10c). The particles in classes 0, 1 and 3 (63,368 particles in total) yielded a resolution of 4.48 Å (Extended Data Fig. 10e). Although the resolution does not improve, the average *B* factor decreased from 121 Å² to 109 Å², suggesting a somewhat overall improvement in map quality (Extended Data Fig. 10e).

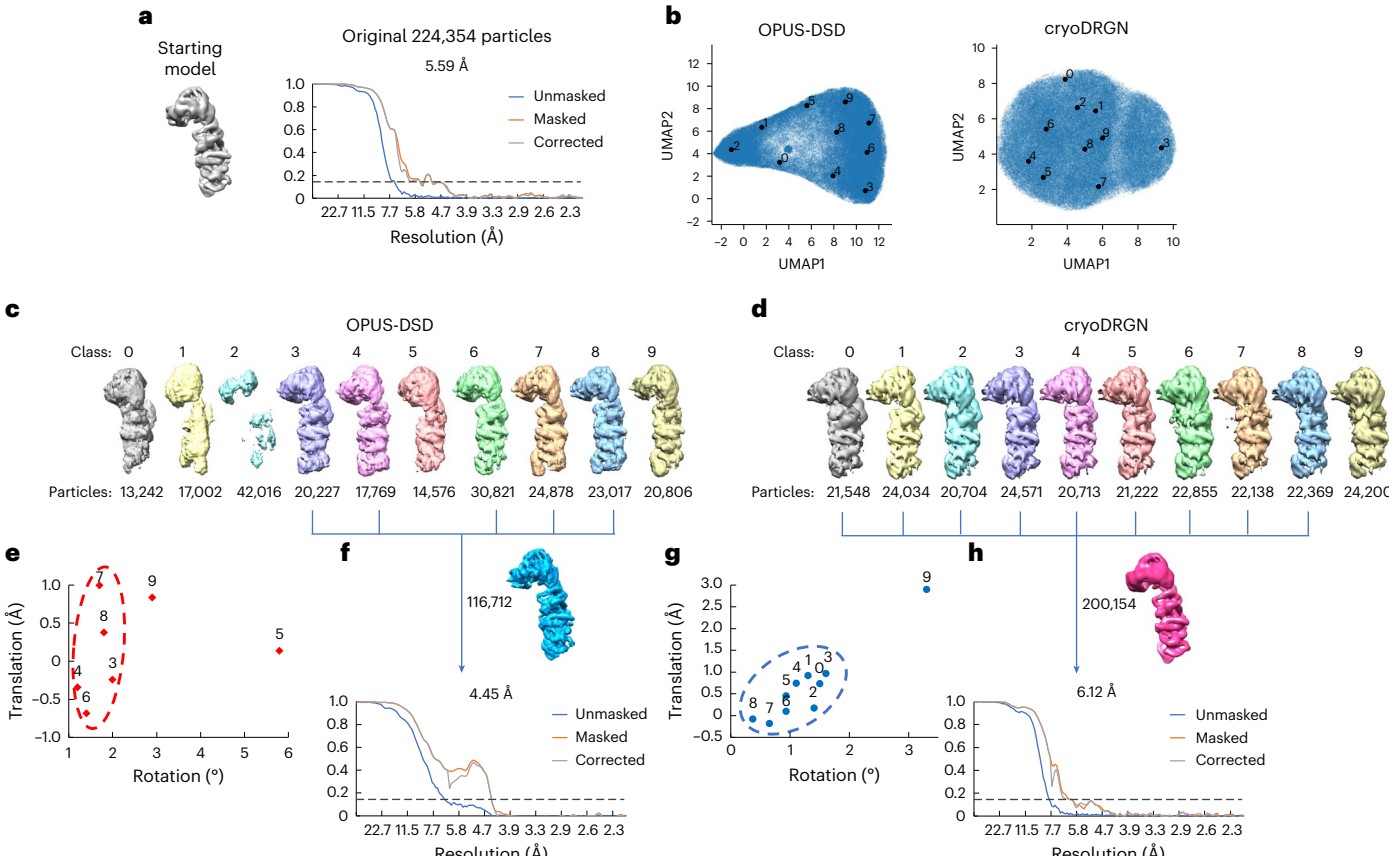

**Fig. 6 | Heterogeneity analysis of 224,354 particles of the NEXT complex.**
**a**, Starting consensus model and its gold-standard FSCs for all particles. The dashed horizontal line denotes FSC = 0.143. **b**, UMAP visualization of the latent space learned by OPUS-DSD and cryoDRGN. Solid black dots represent the cluster centers for labeled classes, identified using the Kmeans algorithm. **c**, Ten classes reconstructed by OPUS-DSD at points shown in **b** and the corresponding number of particles in each class. There were seven classes with complete densities on the right-hand side of the UMAP (classes 3–9), and three classes with incomplete densities on the left-hand side of the UMAP (classes 0–2). **d**, Ten classes reconstructed by cryoDRGN at points shown in **b** and the

corresponding number of particles in each class. **e**, Translation–rotation plot for reconstructions from OPUS-DSD. The movement was measured for the MTR4 domain relative to the ZCCHC8 domain of each conformation by Dyndom using the consensus model as a reference. **f**, Density map and gold-standard FSCs for 116,712 particles by combining the OPUS-DSD clusters that had small structural variations (grouped by the red circle in **e**). **g**, Translation–rotation plot for reconstructions from cryoDRGN. **h**, Density map and gold-standard FSCs for 200,154 particles by combining the cryoDRGN clusters that had small structural variations (grouped by the blue circle in **g**). The dashed horizontal line in **f** and **h** denotes FSC = 0.143.

In contrast, the classes from cryoDRGN exhibited a more uniform distribution in UMAP (Extended Data Fig. 10b) and similar reconstructions (Extended Data Fig. 10d). Non-uniform refinement on classes 1–4 (68,879 particles) from cryoDRGN resulted in a resolution of 5.58 Å and *B* factor of 291 Å² (Extended Data Fig. 10f). As a baseline comparison, reconstructions using four sets of particles in which ~25% of particles were randomly discarded yielded resolutions of 5.60–6.35 Å (Extended Data Fig. 10g).

The test on the 84,530 particles of the NEXT complex shows that OPUS-DSD can still detect real heterogeneity in the dataset at the stage where conventional classification has been converged.

## Discussion

Built upon the PyTorch version of cryo-EM data processing utilities provided by cryoDRGN1.0, OPUS-DSD introduces a set of methodological improvements aiming to effectively and reliably capture structural heterogeneities.

One critical component of OPUS-DSD is the 3D convolutional neural network with translational equivariance. Consensus 3D refinement can determine only the well-ordered portions of a macromolecule to high resolution, while averaging out the densities of flexible parts across space as exemplified by the pre-catalytic spliceosome (Extended

Data Fig. 2). Hence, the pose parameters of well-ordered structural elements in each image are better determined than those of flexible structural elements. If the unresolved mobile elements undergo rigid-body movements, the problem of resolving structural heterogeneity can be simplified as multi-body refinement[7] in which the pose parameters of dynamic elements are determined in relation to the well-ordered structures. By contrast, neural networks can resolve structural heterogeneity end-to-end by reconstructing the underlying conformations directly from an image without prior knowledge. To accomplish this, the neural network needs to be equivariant to variations of objects in input. Specifically, for a neural network that is equivariant to translation, any amount of shift of an object in input results in the same degree of shift of the same object in output. Convolutional neural networks are well-known for translational equivariance[29], which contributes critically to the effectiveness of OPUS-DSD in modeling structural heterogeneity using consensus refinement results. In contrast, cryoDRGN leverages a different mechanism by operating in reciprocal space, that is, using the Fourier shift theorem, in which shifting a function by Δ in real space results in a multiplication of the Fourier transform of that function by $e^{-i2\pi\Delta s}$ in reciprocal space. Consequently, a local change in real space causes a global change to every voxel in reciprocal space. Therefore, cryoDRGN needs to approximate

the change in every voxel in reciprocal space to high precision to reliably capture the structural heterogeneity.

By contrast, cryo-EM images contain high levels of noise. Even for the well-resolved structures in a complex determined by consensus refinement, their projection poses estimated by consensus refinement could still contain significant errors. These errors could compromise the quality of the 3D density map reconstructed by the neural network. To overcome this, OPUS-DSD leverages real-space 3D volume as an intermediate representation[16,17]. This representation enables fast sampling of different projections near the given direction without reevaluating the decoder. This set-up enables OPUS-DSD to incorporate a reconstruction loss in which an experimental image is compared with an ensemble of projections at its estimated pose determined from consensus refinement and neighboring poses. Conceptually similar to the local angle search in RELION[30], the inclusion of this loss increases the robustness of OPUS-DSD to errors in pose parameters. In addition, the quality of the 3D volume representation is improved by encouraging its smoothness and sparseness via traditional regularizers such as total variation and LASSO[19,31,32], which are integrated into the training objective function of OPUS-DSD.

Despite these methodological improvements, training a neural network to reconstruct 3D dynamic objects using only 2D images remains challenging. The most formidable challenge comes from the inherent ill-posedness of the 2D to 3D translation, in which the encoder must disentangle the 3D structural information from other non-structural factors in 2D cryo-EM images. To this end, OPUS-DSD uses a multifaceted approach. The ill-posedness is first mitigated by reducing the pose variations in the input distribution through a back-projection step, in which the input is aligned with the same reference frame as the consensus model, which assists the encoder to disentangle structural heterogeneity from pose variations in 2D images. The ill-posedness is further alleviated by encouraging the smoothness of the latent space with respect to structural variations so that similar 3D structures are encoded by similar latent codes. By encouraging the smoothness of the latent space, the number of training samples available for the decoder at any meaningful latent code increases, ultimately improving the quality of the 3D density map reconstructed by the decoder. In OPUS-DSD, the smoothness of the latent space is encouraged by β-VAE as in cryoDRGN[12,13], and additionally by a multi-component latent prior that facilitates the clustering of images with similar structures in latent space. Furthermore, data augmentation is implemented in OPUS-DSD to reduce its sensitivity to contrast variations in 2D images.

We expect OPUS-DSD to facilitate structural determination and analysis of macromolecular complexes with unresolved structural elements due to high mobility, regardless of whether the mobile elements undergo rigid-body movements or not. Moreover, the end-to-end neural network-based approaches such as cryoDRGN and OPUS-DSD require no prior knowledge about the division of subcomplexes and can automatically reconstruct them and the corresponding dynamics. This enables the reconstruction of synchronous movement among different subcomplexes. One trade-off of this approach is that the resolution of reconstructed subcomplexes is not as high as in multi-body refinement. Additionally, due to the 3D volume representation in real space, OPUS-DSD has a somewhat higher computational cost than cryoDRGN, which only needs to output a 2D slice in the Fourier domain during training. By contrast, OPUS-DSD is able to output a 3D volume much faster in one pass during inference time, while cryoDRGN needs to evaluate the 3D volume voxel by voxel.

## Online content

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

## Methods

### Notation

The notation used in this paper is as follows. $V$ represents the 3D structure. $X$ represents the 2D projection of the 3D structure. For a vector $x \in \mathbb{R}^N$, $\|x\|^2 = \sum_i x_i^2$ is the sum of squares of the vector $x$. $\|x\|_2 = \sqrt{\sum_i x_i^2}$ represents the $l_2$ norm of the vector $x$. $\|x\|_1 = \sum_i |x_i|$ represents the $l_1$ norm of the vector $x$. $\cdot^T$ represents the transpose of a matrix. Var represents the variance of a random variable. SO(3) represents the 3D rotation group.

### Image formation model

The image formation model of cryo-EM is typically defined in the frequency domain. Given that the images collected in cryo-EM are 2D projections of a 3D molecular structure, their Fourier transform has the following relationship with the Fourier transform of the 3D molecular structure according to the projection-slice theorem. Let the 3D molecular structure be $V$, and its Fourier transform be $v$. Assume an $N^2$ image $X$ is formed by rotating an $N^3$ 3D volume $V$ with the Euler angle set, $\phi$, and projecting along the $z$ axis; using the projection-slice theorem, the Fourier transform of the image $x$ can be expressed as

$$x_{(h,k)} = \mathrm{CTF}_{h,k} \sum_{h'=1}^{N} \sum_{k'=1}^{N} \sum_{l'=1}^{N} P^{h,k}_{h'k'l'}(\phi) v_{h'k'l'} + \in_{h,k}, \quad (1)$$

where $x_{h,k}$ is a component of the Fourier transform of the image $X$ with the spatial frequency vector $[h,k]$, $\mathrm{CTF}_{h,k}$ is a component of the CTF (ref. [36]), $P^{h,k}_{h'k'l'}(\phi)$ is the slice operator that slices a plane in the 3D Fourier transform $v$ according to the Euler angle set $\phi$, and $\epsilon_{h,k}$ is the noise that corrupts the projection. The 2D image $X$ can further undergo a set of 2D rigid transformations such as rotation and translation in the projection plane, which, however, are omitted in our discussions because they can be easily corrected using known pose parameters.

We elaborate on the slice operator $P^\phi$ by giving its formal definition. Let the index of a voxel in the 3D Fourier transform $v$ be $[h',k',l']$, and the index of the corresponding pixel of the Fourier transform of the image is $[h,k]$; the slice operator $P^\phi$ transforms the 3D index to a 2D index by the following equation:

$$\begin{pmatrix} h \\ k \\ 0 \end{pmatrix} = R_\phi^T \begin{pmatrix} h' \\ k' \\ l' \end{pmatrix}, \quad (2)$$

where $R_\phi$ is a rotation matrix parameterized by the Euler angle set $\phi$. The counterpart of the slice operator in real space is the projection operator. Let $V$ be the template volume, and $V'$ be the volume rotated by the Euler angle set $\phi$; the projection operator rotates the 3D template volume by the following equation:

$$V'\left(\begin{pmatrix} x \\ y \\ z \end{pmatrix}\right) = V\left(R_\phi^T \begin{pmatrix} x \\ y \\ z \end{pmatrix}\right). \quad (3)$$

The projection operator then generates the 2D projection $X$ by summing along the $z$ axis of the rotated volume $V'$:

$$X\left(\begin{pmatrix} x \\ y \end{pmatrix}\right) = \sum_{z=1}^{N} V'\left(\begin{pmatrix} x \\ y \\ z \end{pmatrix}\right). \quad (4)$$

In summary, the real-space 3D volume in OPUS-DSD undergoes a rotation in 3D and 2D rigid transformations before transforming to a 2D projection. The 2D cryo-EM reconstruction is obtained by first rotating the 3D volume by the spatial transformer[18], which is then projected along the $z$ axis to generate a 2D projection that further undergoes defocus correction to generate the 2D cryo-EM reconstruction

by applying a CTF as in equation (1). The 2D cryo-EM image $X$ is thus a function of the 3D volume $V$, the projection angle $P$ and the defocus parameters of microscope $u$, all of which defines the CTF, that is, $X(V,P,u)$. Therefore, the entire image formation process is differentiable and suitable for end-to-end training.

### cryoDRGN

CryoDRGN proposed a neural representation to model the 2D projection in 3D Fourier volume. The projection can be described as a continuous function $V : \mathbb{R}^3 \to \mathbb{R}$, which maps the 3D coordinate to a voxel value. CryoDRGN approximates the function $V$ directly by using a multilayer perceptron (MLP)[37]. Instead of supplying the 3D coordinates into the MLP, cryoDRGN encodes the coordinates using a specific positional encoding[12]. The 2D projection at a specific angle is computed from positional encodings of the coordinates in its corresponding slice (equation (2)), pixel by pixel. Different structures are generated by concatenating their latent codes to the positional encodings and supplying those combinations to the MLP. CryoDRGN is trained using β-VAE[13].

### Structural disentanglement prior

In the framework of the encode–decoder network, resolving the structural heterogeneity in a cryo-EM dataset can be formulated as learning a latent space that captures the 3D structural information. Each latent code represents a unique 3D structure in this space, $z \sim V$. Given a latent code, the decoder network can produce the corresponding 3D volume $g(V|z)$. The similarity between 3D structures correlates with the distance between their latent codes. From the perspective of the decoder, encoding structural information in the latent space can be encouraged by explicitly supplementing the non-structural information such as poses and defocus parameters into the reconstruction of the 2D projection from the 3D volume. If reliably modeled, the non-structural information in 2D images will not propagate from the decoder into the latent space during backpropagation. However, when the pose assignments of the 2D images are erroneous, the 3D reconstruction from the decoder will be distorted to account for the pose assignment errors. Therefore, this approach is greatly affected by the accuracy of pose assignment, which depends heavily on the signal-to-noise ratios of a dataset. The power of a neural network for resolving structural heterogeneity will quickly deteriorate as the signal-to-noise ratios of a dataset drop. The disentanglement of structural content from pose parameters in cryoDRGN 1.0 was achieved via this approach[38]. The most recent version of cryoDRGN2 introduced a pose update to mitigate this issue[39].

From the perspective of the encoder, learning a structural latent space can be achieved by learning a projection and defocus invariant encoding for all 2D cryo-EM images from the same 3D structure, namely, $f(X(V,P,u)) = f(X(V,P',u')), \forall P,P' \in \mathrm{SO}(3), \forall u,u' \in \mathbb{R}^n$. The 2D cryo-EM image is a function, $X(V, P, u)$. It can vary considerably by changing the projection angle and defocus parameters while fixing the underlying 3D structure, namely, $X(V,P,u) \neq X(V,P',u')$ as long as $P \neq P'$ or $u \neq u'$. In order to approximate such a latent space that primarily encodes 3D structural information, the encoder network should disentangle the 3D structural variations in 2D inputs from other factors such as projection angles and defocus parameters.

An important component of OPUS-DSD is a latent space prior disentangling the 3D structural heterogeneity from the non-structural information and encoding the 3D structural heterogeneity. The encoder network in OPUS-DSD is encouraged to generate latent encodings with larger variations for 3D structural changes in 2D inputs, while being relatively insensitive to the poses and contrast changes, namely, $\mathrm{Var}(\partial f(X(V,P,u))/\partial V) \gg \mathrm{Var}(\partial f(X(V,P,u))/\partial \Delta), \Delta = P$ or $u$, where variances are taken over all images. Specifically, the disentanglement is achieved by adding attracting forces and repelling forces between the encodings of a specific combination of images. The attracting forces restrain the distance between the latent codes of images from

similar 3D structures, while the repelling forces encourage the separation between the latent codes of different 3D structures.

The latent space prior is composed of two components, namely the intraclass prior for 2D images of similar projection angles and the interclass prior for 2D images of different projection angles. First let us consider the intraclass prior. The 3D structural heterogeneity is most discernible in 2D cryo-EM images that have the same projection angle, namely, the variation of $X(V,P,u)$ can be mainly attributed to $V$ when conditioned on $P$. The repelling force is added between the latent codes of those pairs of images to amplify the difference between latent codes for images from different 3D structures, which encourages the encoder to discern images from different structures. Next, to encourage the smoothness of latent codes, the distances of the latent codes of similar images are restrained to prevent over-separation. To formally define the prior, given the projection angles form an SO(3) group that can be discretized into a number of classes using HEALPix[40], 2D images can be classified into different projection classes according to their projection angles. Suppose the encoder network is $f$, the projection class of image $X_i$ is $P_i$, the projection class of image $X_j$ is $P_j$, inspired by the objective function of UAMP[21], the prior for the encoder network that encourages the encoding of structural information for images in the same projection class $P_i$ can be expressed as

$$J_1(f) = \sum_{X_i} \frac{1}{K} \sum_{f(X_j) \in \text{kNN}(f(X_i)), P_j = P_i}^{K}$$
$$\log\left(1 + ||f(X_i(V_i, P_i, u_i)) - f(X_j(V_j, P_j, u_j))||^2\right)$$
$$+ \sum_{X_i} \frac{1}{N} \sum_{X_j \neq X_i, P_j = P_i}^{N} \log\left(1 + \frac{1}{||f(X_i(V_i, P_i, u_i)) - f(X_j(V_j, P_j, u_j))||^2}\right), \quad (5)$$

where kNN refers to the $k$ nearest neighbors of the image measured by Euclidean distance in latent space, $K$ refers to the number of nearest neighbors, and the second summation in the second term is over all images in the same class as image $X_i$ except itself, and $N$ refers to the size of the projection class of image $X_i$. For images with similar codes, the attracting prior reaches a minimum when $f(X_i) = f(X_j)$. By contrast, the minimum of the repelling prior is obtained at $||f(X_i) - f(X_j)|| \to \infty$, thus the encoder is forced to amplify the structural differences between $X_i$ and $X_j$.

Next let us consider the interclass prior aiming to improve the clustering of images according to structural information while reducing the impact of projection poses. This is achieved by collating the latent codes of different projections of the same 3D structure. The projections with similar latent codes are considered to be from the same structure. The attracting force can be added among them. To encourage the separation of different 3D structures, a repelling prior is added to an image and its $k$ farthest point in latent space. Formally, let the kNN of the image $X_i$ in latent space be kNN$(f(X_i))$, the projection class of image $X_i$ be $P_i$, the $k$ farthest point of the image $X_i$ in latent space be kFP$(f(X_i))$, then our prior for encouraging the clustering of images with similar latent codes and different poses can be expressed as

$$J_2(f) = \sum_{X_i} \frac{1}{K} \sum_{f(X_j) \in \text{kNN}(f(X_i)), P_j \neq P_i}^{K}$$
$$\log\left(1 + ||f(X_i(V_i, P_i, u_i)) - f(X_j(V_j, P_j, u_j))||^2\right)$$
$$+ \sum_{X_i} \frac{1}{N} \sum_{f(X_j) \in \text{kFP}(f(X_i)), P_j \neq P_i}^{N} \log\left(1 + \frac{1}{||f(X_i(V_i, P_i, u_i)) - f(X_j(V_j, P_j, u_j))||^2}\right), \quad (6)$$

where the second summation in both terms is over all images that do not belong to the same projection class as $X_i$, $K$ refers to the number of nearest neighbors of $X_i$ and $N$ refers to the number of farthest points of $X_i$, which are both tunable hyperparameters. The complete structural

disentanglement prior is the sum of equation (5) and equation (6), which can be written as

$$J(f) = \sum_{X_i} \frac{1}{K_1} \sum_{f(X_j) \in \text{kNN}(f(X_i)), P_j = P_i}^{K}$$
$$\log\left(1 + ||f(X_i) - f(X_j)||^2\right)$$
$$+ \frac{1}{N_1} \sum_{X_j \neq X_i, P_j = P_i} \log\left(1 + \frac{1}{||f(X_i) - f(X_j)||^2}\right) \quad (7)$$
$$+ \frac{1}{K_2} \sum_{f(X_j) \in \text{kNN}(f(X_i)), P_j \neq P_i} \log\left(1 + ||f(X_i) - f(X_j)||^2\right)$$
$$+ \frac{1}{N_2} \sum_{f(X_j) \in \text{kFP}(f(X_i)), P_j \neq P_i}^{N} \log\left(1 + \frac{1}{||f(X_i) - f(X_j)||^2}\right),$$

where the first two terms for images with $P_j = P_i$ are the intraclass comparison, while the last two terms for images with $P_j \neq P_i$ are the interclass comparison.

To compute the structural disentanglement latent prior, equation (5) can be approximated using a batch of images from the same projection class (intraclass). For equation (6), given that a cryo-EM dataset contains hundreds of thousands of images, it is computationally inefficient to compute kNN and kFP for each image over all images. We compute them approximately for each query by using a subset of images from projection classes other than the query's projection class to ensure interclass comparison. This subset is randomly sampled from a memory bank that stores all latent codes. The distances between the query latent code and samples are then computed and sorted in ascending order. The first $K_2$ latent codes with the shortest distances are designated as the nearest neighbors of the query. The last $N_2$ latent codes with the longest distances are denoted as the farthest points of the query.

## Data augmentation

The final ingredient to increase the robustness of OPUS-DSD to contrast changes is a data augmentation pipeline. The CTF blurs the cryo-EM 2D projection while strongly modulating its contrast[36]. The encoder might learn to recognize the pattern of the CTF instead of the signal of the underlying structure. To make the network more robust to the defocus variations, we propose to corrupt the pattern of the CTF by constructing input and output image pairs with randomized amplitudes of Fourier transforms. Specifically, the input and output are two images that have been independently constructed by multiplying the random Gaussian function with the Fourier transform of the original image. That is, let the Fourier transform of the constructed image be $\mathcal{X}'_{h,k}$ and the Fourier transform of the original image be $\mathcal{X}_{h,k}$, and the data augmentation process is defined as

$$\mathcal{X}'_{h,k} = e^{-b\pi^2 s^2} \mathcal{X}_{h,k}, \quad (8)$$

where $b$ is a uniform random $B$ factor in the range [−0.5, 0.5] and $s = \sqrt{h^2 + k^2}$ is the modulus of the spatial frequency vector of the Fourier transform. The image contrast is sharpened when $b < 0$, while the image contrast is blurred when $b > 0$. The $B$ factors applied to input and output are two independently sampled values. We further multiply the Gaussian blurred or sharpened input image with a uniform random constant in the range [0.75, 1.25] to change its overall contrast and brightness. The input image is then fed into the encoder, while the decoder is asked to reconstruct the output image. The neural network thus is forced to reconstruct an image using input with different contrast values. This data augmentation pipeline increases the robustness of the encoder network to the contrast variations of inputs, and reduces the impact of defocus parameters on latent codes.

## Training objectives

The smoothness of latent space is of paramount importance to the generalizability of the generative model. To generate plausible samples

during a traversal through the latent space, the generative model needs to smoothly interpolate between training samples. One way to achieve this is by VAE[14]. Instead of producing a deterministic latent code, the encoder approximates a posterior distribution of latent code conditioned on image $X$, $f(z|X)$, which is often assumed to be a diagonal Gaussian distribution for simplicity. The encoder then parameterizes this diagonal Gaussian distribution by outputting its mean and variance. The smoothness of latent space is encouraged by increasing the overlapping between the approximate posterior distributions of images, namely, restraining the deviation of approximate posterior distributions from the standard Gaussian distribution. The default VAE assumes that the reconstruction loss and restraints are of equal weights, while β-VAE introduces a tunable parameter to control the strength of the standard Gaussian restraints[13]. Furthermore, to generate a plausible 3D volume, total variation and LASSO priors are imposed on the reconstructed 3D volume to encourage its smoothness and sparseness[31,32].

As mentioned in the Structural Disentanglement Prior subsection, the pose assignment errors from consensus refinement have an adverse effect on the quality of the 3D reconstruction output by the neural network. We have therefore designed a reconstruction loss that is robust to pose assignment errors by comparing an experimental image with multiple reconstructions obtained at its pose from consensus refinement and neighboring poses. The objective function leveraged by OPUS-DSD with the aforementioned considerations is as follows. Using β-VAE, let $θ$ be the projection angle determined by a consensus 3D refinement for each image, $f$ be the encoder network, $g$ be the decoder network, $F$ represent the image formation process given the 3D volume $V$, projection angle $θ'$ and defocus parameters $u$, let the dimension of the 3D volume be $N^3$, and the objective function for learning a neural network to resolve 3D structural heterogeneity can be expressed as

$$
\min_{f,g} E_{f(z|X)} - \log \sum_{θ' \in NN(θ)} \exp -\frac{1}{2}||X - F(g(V(\mathbf{x})|z), θ', u)||^2
$$
$$
+ βD_{KL}\left(f(z|X)||\mathcal{N}(0,I)\right) + λJ(f(z|X)) \tag{9}
$$
$$
+ \frac{λ_{tv}}{N}\sum_{\mathbf{x}}\left\|\left\|\frac{\partial g(V(\mathbf{x})|z)}{\partial \mathbf{x}}\right\|\right\|_2 + \frac{λ_{l_1}}{N}\sum_{\mathbf{x}}||g(V(\mathbf{x})|z)||_1,
$$

where $NN(θ)$ represents the nearest neighbors of the projection angle $θ$, the first term is the expectation of errors between the ground-truth image $X$ and the reconstruction $F(g(V|z), θ', u)$ over the distribution of latent code $z$, $\mathcal{N}(0,I)$ is the standard Gaussian distribution, $D_{KL}$ is the Kullback–Leibler (KL) divergence between the distribution given by the encoder network $f$ and the standard Gaussian $\mathcal{N}(0,I)$, the term $J(f(z|X))$ is the structural disentanglement prior to encourage the encoding of the 3D structural information in latent space, $\frac{\partial g(V(\mathbf{x})|z)}{\partial \mathbf{x}}$ denotes the gradient of the 3D volume at grid point $\mathbf{x}$, $β$ is the restraint strength for $D_{KL}$, $λ$ is the restraint strength for the structural disentanglement prior, $λ_{tv}$ is the restraint strength for total variation, and $λ_{l_1}$ is the restraint strength for LASSO. Computing the expectation in equation (9) is intractable, and we use a reparameterization trick[14] to approximate the expectation integral.

### Training

Our network was trained using 2D images. The projection pose parameters of 2D images were determined by the consensus 3D refinement in RELION or cryoSPARC[6,41]. The 2D images were randomly split into a training set and a validation set using a specified split ratio. The images in the validation set were used to evaluate the reconstruction quality of the neural network only. For the spliceosome and $Pf$80S ribosome with a large amount of data and good signal-to-noise ratios, we used 20% of the images for validation. For the $Sc$80S ribosome and NEXT complex data, 10% of the images were used for validation. The computation of the structural disentanglement prior requires us to perform intraclass and interclass comparisons. Hence, the training images were classified

into 48 different projection classes according to their first two Euler angles using HEALPix[40] before being sent to training. At each iteration a projection class was selected uniformly and a batch of images in the selected projection class was sampled by a customized batch sampler. The structural disentanglement prior can be easily computed using such a batching process. The image sample was passed to the encoder, which outputs a sample of the latent distribution $f(z|X)$. The decoder reconstructs a 3D volume according to the latent code. The total variations and LASSO priors were imposed on the reconstructed 3D volume to encourage its smoothness and sparseness[31,32]. The 3D volume was rendered into a 2D reconstruction with predetermined pose according to the cryo-EM image formation model. The inputs and reconstructions together with the latent codes were used to construct the training loss as in equation (9), which was minimized using the Adam optimizer[42].

### Implementation

The neural network of OPUS-DSD was implemented in PyTorch with automatic differentiation support[43]. The major computation bottleneck is caused by the 3D convolution autoencoder given that every 2D image has to go through a series of 3D operations. The run time and memory cost of OPUS-DSD was hence almost irrelevant to the size of the 2D images. Using four Nvidia V100 GPUs (graphics processing units), the training of 1,000 images took around 1 min. For the pre-catalytic spliceosome dataset containing 262,816 images, the training took 3 h per epoch (an epoch represents a loop through the whole training set once) using four Nvidia V100 GPUs, and the best results were obtained between 12 and 16 epochs. The 3D convolutional architecture incurred a larger memory cost than cryoDRGN. When the output 3D volume is of a size of $256^3$, an Nvidia V100 GPU with 32 GB memory could process around 16 images at once. However, OPUS-DSD enables the user to specify a smaller output volume to save memory, which was accomplished by resampling intermediate tensors: for example, the resampling of the tensor of $256 \times 8^3$ in Fig. 1c to $256 \times 6^3$ yields an output volume of $192^3$, thus saving memory. Despite the larger computational and memory cost compared with cryoDRGN, the run time of OPUS-DSD should still be acceptable given that it generally converges well before the 20th epoch. For example, on the $Sc$80S ribosome, cryoDRGN took 8.75 h to converge using four GPUs, while OPUS-DSD took 11 h to obtain the results presented here. On the synthetic NEXT complex, cryoDRGN took 3 h with four GPUs to converge, while OPUS-DSD took 8 h to obtain the results reported. Moreover, OPUS-DSD was able to very quickly reconstruct a 3D volume for a given latent code because it generated the whole volume in one pass. In contrast, cryoDRGN is required to reevaluate the neural network at each voxel to produce a final volume. Last, OPUS-DSD implements a routine to estimate the signal-to-noise ratio of a dataset and automatically balance the strengths of regularization penalties and the reconstruction loss. Users can specify the parameters using an absolute scale without worrying about the actual signal-to-noise ratios of the dataset.

### Hyperparameter settings

In our experiments we set the restraint strength for KL divergence to be proportional to the signal-to-noise ratio of the dataset, specifically $λ_{KL} = 2 \times$ signal-to-noise ratio. OPUS-DSD can automatically estimate the signal-to-noise ratio during training. The restraint strength for the structural disentanglement prior was also set according to the signal-to-noise ratios of the dataset. For the spliceosome and $Pf$80S ribosome with a high signal-to-noise ratio we set $λ = 2$. For the $Sc$80S ribosome we set $λ = 1$. For the synthetic data and real NEXT complex datasets we set $λ = 0.5$. Furthermore, previous restraint strengths were weighted by a cyclical annealing schedule to avoid posterior collapse[44]. We fixed the restraint strength for the total variation of the 3D volume as $λ_{tv} = 1$ and the restraint strength for LASSO as $λ_{l_1} = 0.3$. The learning rate was set to $10^{-4}$ and decayed by 0.95 at each epoch for all experiments. The batch size was set to 72 during the training. All training

sessions were performed on four Nvidia V100 GPUs. The number of samples for interclass comparison was set to 12,800. The number of kNNs for interclass or intraclass comparison was set to 128 or 4, respectively, while the number of kFPs for interclass comparison was set to 512. The latent codes of particles were updated with a momentum of 0.7. The dimension of latent space can limit the amount of information flowing from encoder to decoder, and was often set to around 10 in the 2D to 3D translation in our experiments.

## Computation protocol

For each system the structural heterogeneity analyses were performed on the same consensus refinement result for different methods. For cryoDRGN, due to the lack of validation metrics, the number of training epochs was chosen based on the quality of the density maps reconstructed. By contrast, OPUS-DSD selected models for further analyses according to the average reconstruction errors using the validation set. The structural heterogeneity was determined by analyzing the 3D density maps reconstructed from the centroids of clusters in latent space generated by the simple Kmeans clustering algorithm[45]. The latent space of different methods was also visualized in 2D using UMAP[21]. The structural heterogeneity was also analyzed using PCA. Kmeans clustering and PCA were carried out in scikit-learn 1.3.0 (ref. 46). All movies were generated by Chimera[47].

## Cryo-EM data collection

Recombinant proteins (MTR4, RBM7 and ZCCHC8) of the NEXT complex were overexpressed separately in HEK293 GnTI⁻ cells. Cells were infected with 10% volume of recombinant baculovirus at 37 °C, 95 rpm for 20 h before adding sodium butyrate to a final concentration of 10 mM, and then transferred to 30 °C for another 40 h. Cells were collected by centrifugation at 800g for 10 min at 4 °C. Recombinant protein cell pellets were mixed and suspended in lysis buffer containing 20 mM HEPES, pH 7.4, 300 mM NaCl, 4% glycerol, 1 mM TCEP and 1% NP-40, 0.1‰ nuclease and lysed using high-pressure homogenizer before centrifugation at 44,000g for 30 min at 4 °C. The supernatant was incubated with 1 ml (50% slurry) poly-flag-tag beads for 1 h at 4 °C and then poured into a gravity flow column, and beads were washed with 20 mM HEPES, pH 7.4, 300 mM NaCl and 4% glycerol. Bound proteins were eluted in the buffer containing 20 mM HEPES, pH 7.4, 300 mM NaCl, 4% glycerol and 400 µg ml⁻¹ poly-flag-peptide. The NEXT complex was further purified using Superdex 200 Increase 10/300 (GE Healthcare) in a buffered solution containing 20 mM HEPES, pH 7.4, 300 mM NaCl and 4% glycerol. The desired fraction was concentrated to 50 µl. Protein cross-linking was achieved by density gradient centrifugation (10–30% glycerol, 0.2% glutaraldehyde) at 47,000 rpm (SW-55Ti, Beckman) for 8.5 h at 4 °C. Sample after centrifugation was concentrated to a minimal volume and changed to the buffer containing 20 mM HEPES, pH 7.4, and 150 mM NaCl by dialysis. Three microliters of the complex sample were applied to a Cryo Matrix amorphous alloy film R1.2/1.3 300 mesh (Zhenjiang Lehua Technology) that had been glow-discharged and vitrified using Vitrobot (Thermo Fisher Scientific) at 4 °C and 100% humidity. The experimental NEXT dataset was collected using a Titan Krios G3i 300 KV electron microscope (Thermo Fisher Scientific) equipped with a BioQuantum-K3 Summit camera (Gatan) with a 20 eV slit and in super-resolution mode. Micrographs were captured at a nominal magnification of 81,000 and a calibrated pixel size of 0.55 Å. Each video stack received a total electron dose of 50 e⁻ per Å² in a 2.19-s exposure.

## Synthetic data preparation

The synthetic NEXT dataset consisted of 64,000 2D images of eight different conformations that were generated from the starting conformation by shifting the MTR4 helicase domains relative to the ZCCHC8 center. The density maps for these eight conformations were generated using the molmap command in Chimera from the corresponding atomic models[47]. The 2D projection was obtained by projecting the density map of a conformation onto the 2D plane along a randomly sampled projection angle. The 2D projection was then convolved with a randomly sampled CTF and contaminated by specific levels of noise to generate the final synthetic 2D images. Each conformation contained 8,000 2D images produced in this way.

## Reporting summary

Further information on research design is available in the Nature Portfolio Reporting Summary linked to this article.

## Data availability

We used the following publicly available datasets: EMPIAR-10180 (structure of a pre-catalytic spliceosome), EMPIAR-10028 (cryo-EM structure of a *Pf*80S ribosome bound to the anti-protozoan drug emetine) and EMPIAR-10002 (*Sc*80S ribosome direct electron detector dataset). Synthetic and real NEXT datasets are deposited in Zenodo at https://doi.org/10.5281/zenodo.8093296 (ref. 48). Cryo-EM density maps for the real NEXT complex are deposited in the Electron Microscopy DataBank (EMDB) with the accession number EMD-37262. Trained models and heterogeneity analysis results are deposited in Zenodo at https://doi.org/10.5281/zenodo.8143779 (ref. 49). Source data are provided with this paper.

## Code availability

OPUS-DSD software is deposited in Code Ocean at https://doi.org/10.24433/CO.3046690.v1 (ref. 50) and is also available at https://github.com/alncat/opusDSD.

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

## Acknowledgements

We thank C. Jia for providing the cryo-EM dataset for the NEXT complex. We also thank the staff at MRICS Cryo-EM Center at Fudan University for their help with data collection. We also acknowledge the open-source software cryoDRGN, upon which OPUS-DSD was built. The research was partially supported by the National Key Research and Development Program of China (No. 2021YFF1200400), and Shanghai Municipal Science and Technology Major Project (No. 2018SHZDZX01) and ZJLab.

## Author contributions

Z.L., Q.W. and J.M. conceived the work. Z.L. designed the algorithm and implemented the software. Z.L. and F.N. performed the experiments. All authors wrote and approved the final paper.

## Competing interests

Q.W. is an employee of Harcam Biomedicines who is bound by confidentiality agreements that prevent her from disclosing the competing interests in this work. The other authors declare no competing interests.

## Additional information

**Extended data** are available for this paper at https://doi.org/10.1038/s41592-023-02031-6.

**Correspondence and requests for materials** should be addressed to Jianpeng Ma.

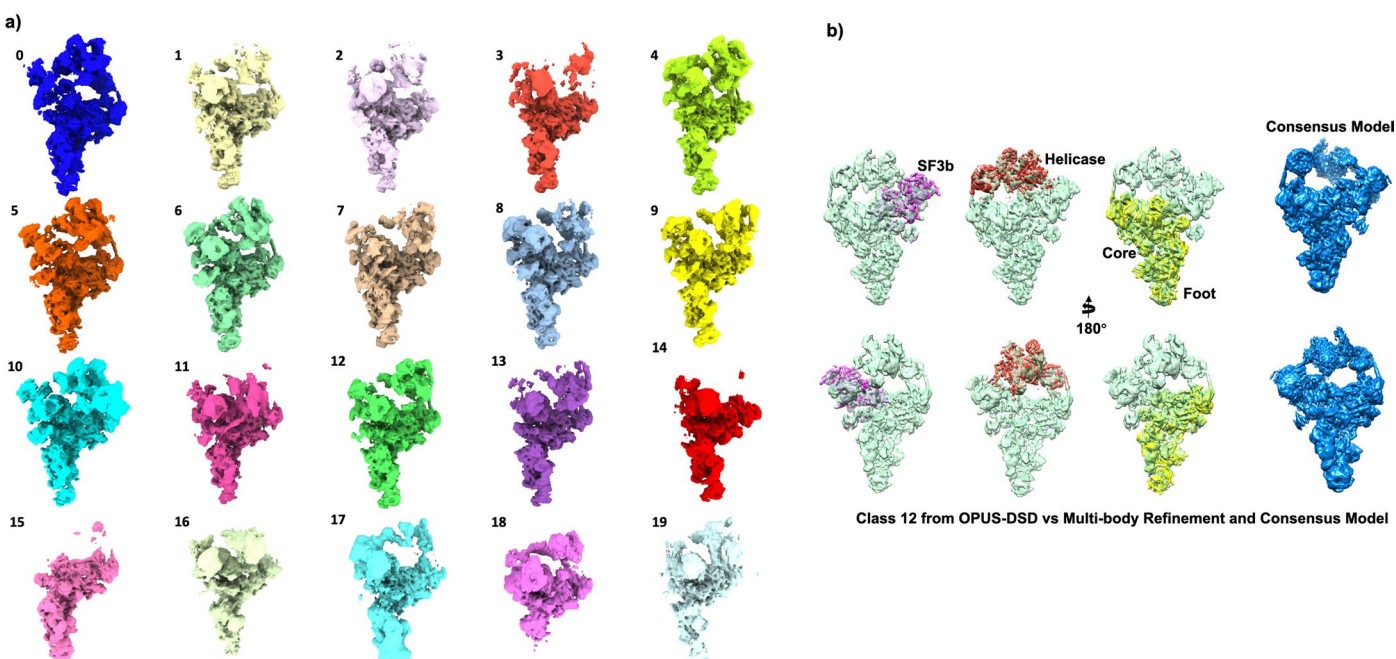

**Extended Data Fig. 1 | Density maps of the pre-catalytic spliceosome reconstructed by OPUS-DSD's decoder. a).** Twenty density maps of the pre-catalytic spliceosome generated by OPUS-DSD's decoder at cluster centers shown in Fig. 2a as inputs. **b).** Comparisons between densities of SF3b, Helicase and Core in Class 12 from OPUS-DSD and high-resolution structure from multi-body refinement, together with the consensus model. Class 12 is shown in green and transparent. The high-resolution structures from multi-body refinement are shown in solid colors inside Class 12. Left-to-right three columns: the comparisons focusing on SF3b (purple), Helicase (red) and Core/Foot (yellow), respectively. The consensus structure is shown in solid blue on the right as a reference.

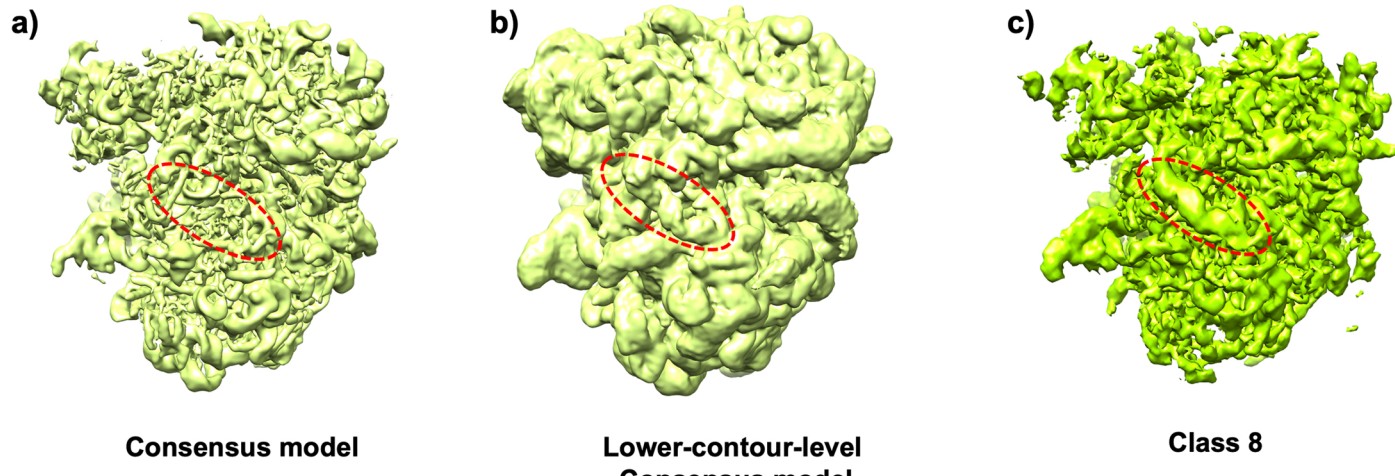

**Consensus model**

**Lower-contour-level
Consensus model**

**Class 8**

**Extended Data Fig. 2 | A highly mobile RNA identified in Class 8 of Pf80S ribosome by OPUS-DSD. a)** and **b).** The consensus map shown at two different contour levels. **c).** The density map of Class 8 identified by OPUS-DSD. The maps in a) and c) are at the same contour level, which the map at b) is at a much lower contour level. The RNA strand is visible in Class 8 from OPUS-DSD (c) and the consensus map at a very low contour level (b), while it is absent in consensus map on the same contour level as Class 8 (a), suggesting its high flexibility.

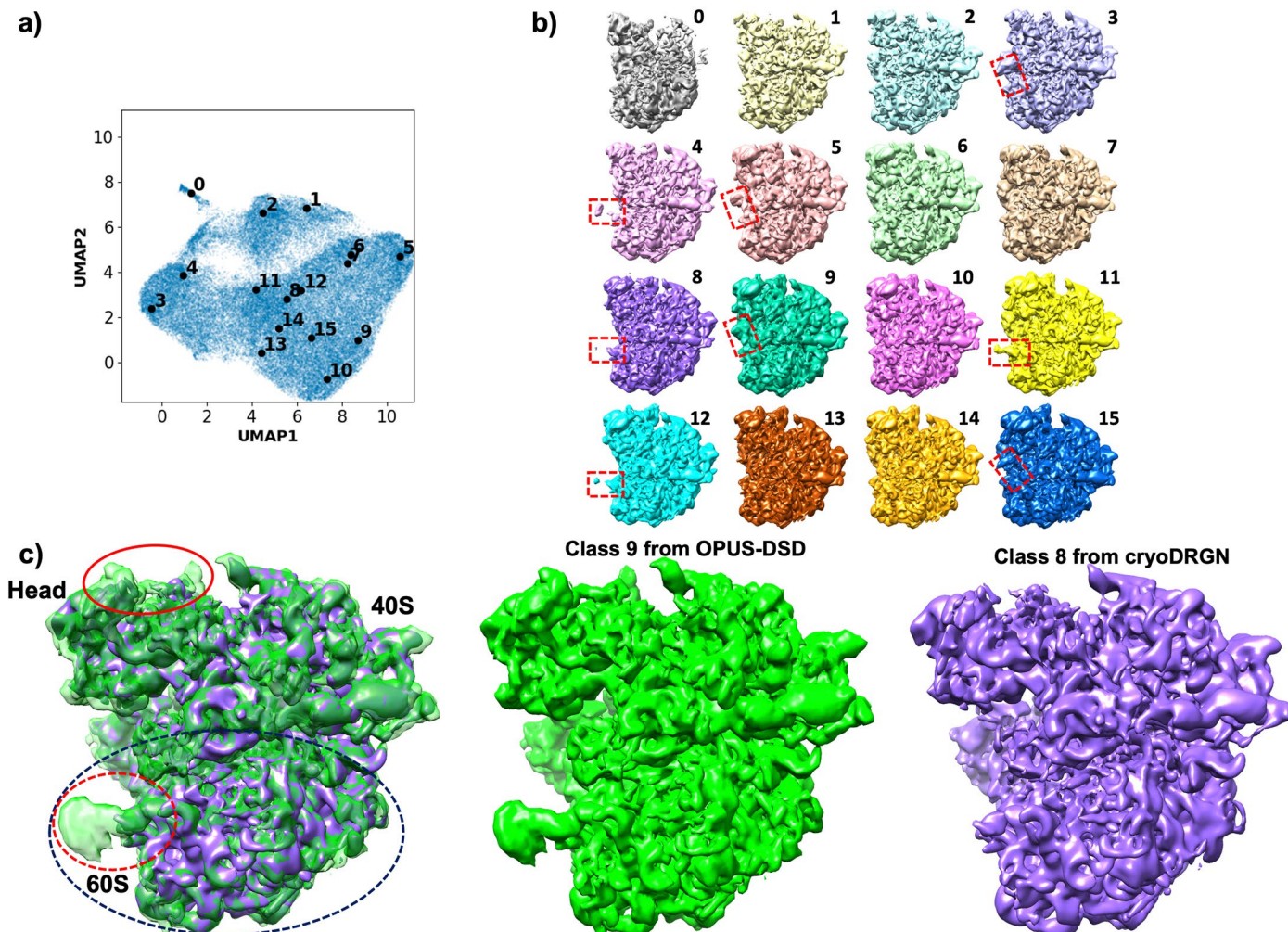

**Extended Data Fig. 3 | Heterogeneity analysis on S.cerevisiae 80S (S.c80S) ribosome (EMPIAR: 10002) by cryoDRGN. a)**. UMAP visualization of the 8-dimensional latent space of all particles encoded by cryoDRGN. Solid black dots represent the cluster centers for labeled classes. **b)**. Sixteen density maps reconstructed by cryoDRGN's decoder using Kmeans clustering centers in its 8-dimensional latent space as inputs. The swinging RNA strand which was revealed in the results of OPUS-DSD can also be observed in cryoDRGN's results here (red dashed boxes). **c)**. Comparison between the density maps of Class 9 from OPUS-DSD (in green color) and Class 8 from cryoDRGN (in purple color). The density map of Class 9 from OPUS-DSD is more complete in regions marked by red dashed ellipse. Both maps are contoured at the same level.

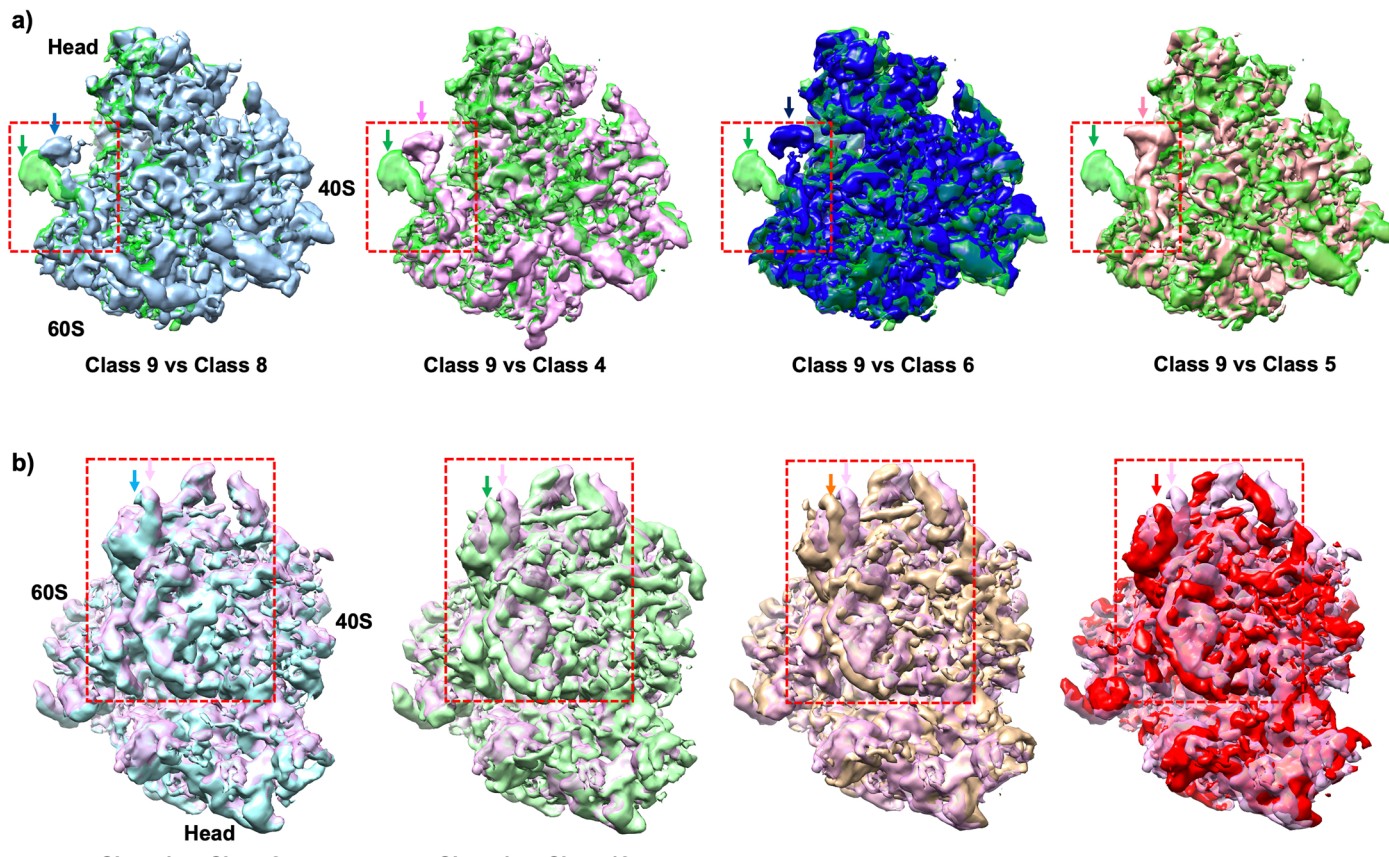

Extended Data Fig. 4 | The movements of an RNA strand and the 40S subunit in S.c80S ribosome revealed by OPUS-DSD. a). The movements of an RNA strand in S.c80S ribosome revealed by OPUS-DSD. The reference density map of Class 9 from OPUS-DSD is colored in green and semi-transparent. The density maps shown in order are Classes 8, 4, 6, and 5, exhibiting increasingly larger displacements in the RNA (as marked by the red dashed boxes) relative to its position in Class 9. The arrows in the same color as the density maps are drawn to highlight the displacements. b). The movement of the 40S subunit in S.c80S ribosome revealed by OPUS-DSD. The reference density map of Class 4 from OPUS-DSD is colored in violet and semi-transparent. The density maps shown in order are Classes 2, 13, 7, and 15, displaying increasingly larger displacements in 40S subunit (marked by red dashed boxes) relative to its position in Class 4. The arrows in the same color as the density maps are drawn to highlight the displacements.

**a)** Reconstruction by OPUS-DSD decoder

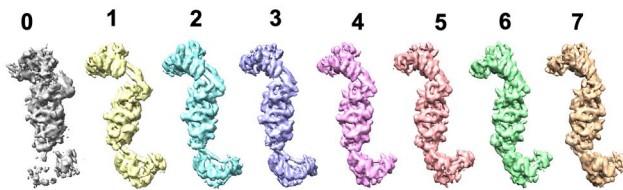

**b)** Reconstruction by cryoDRGN decoder

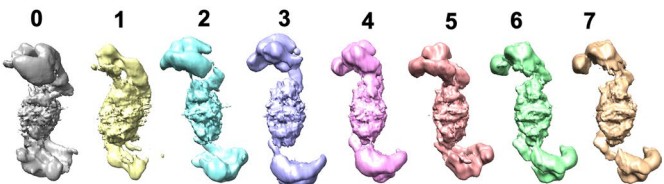

**c)** Reconstruction by cryoSPARC using classes from OPUS-DSD

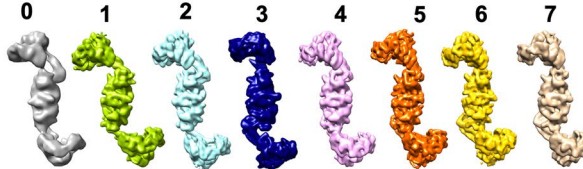

**d)** Reconstruction by cryoSPARC using classes from cryoDRGN

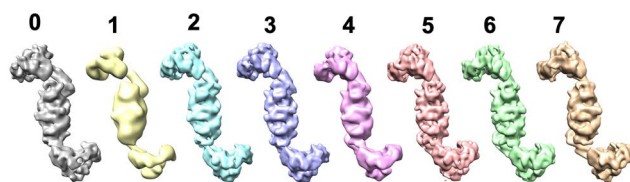

**Extended Data Fig. 5 | Reconstructions of OPUS-DSD and cryoDRGN on synthetic dataset of NEXT complex with a signal-to-noise ratio of 0.05. a)**. 3D density maps reconstructed by the decoder of OPUS-DSD using cluster centers shown in Fig. 6b. **b)**. 3D density maps reconstructed by the decoder of cryoDRGN using cluster centers shown in Fig. 6b. **c)**. 3D density maps reconstructed by cryoSPARC with particles in each class from OPUS-DSD. **d)**. 3D density maps reconstructed by cryoSPARC with particles in each class from cryoDRGN.

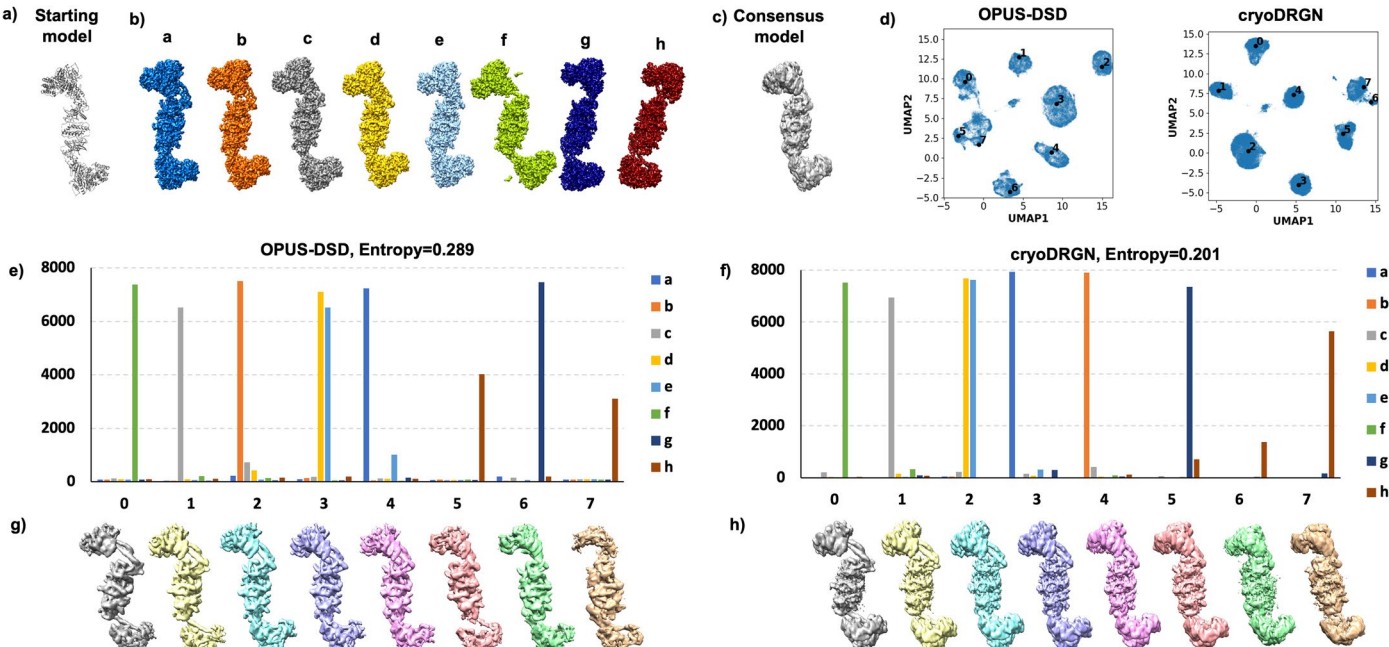

**Extended Data Fig. 6 | Heterogeneity analysis on synthetic dataset of NEXT complex with signal-to-noise ratio of 0.1. a)**. The starting atomic model of NEXT complex. **b)**. Ground-truth density maps of NEXT complexes. **c)**. Consensus model reconstructed by cryoSPARC using all particles. **d)**. UMAP visualizations of the 8-dimensional latent spaces of all particles encoded by OPUS-DSD (left panel) and cryoDRGN (right panel). **e)**. Distribution of particles in clusters in the latent space of OPUS-DSD. Bars are colored to match the profiles of the ground-truth density maps in b). Entropy is defined as the average of $-\sum_i p_i \log p_i$ of each cluster, where $p_i = \frac{n_i}{\sum_i n_i}$, and $n_i$ is the number of particles in the ground-truth conformation $i$. **f)**. Distribution of particles in clusters in the latent space of cryoDRGN. **g)**. 3D density maps reconstructed by particles in each class from OPUS-DSD. **h)**. 3D density maps reconstructed by particles in each class from cryoDRGN.

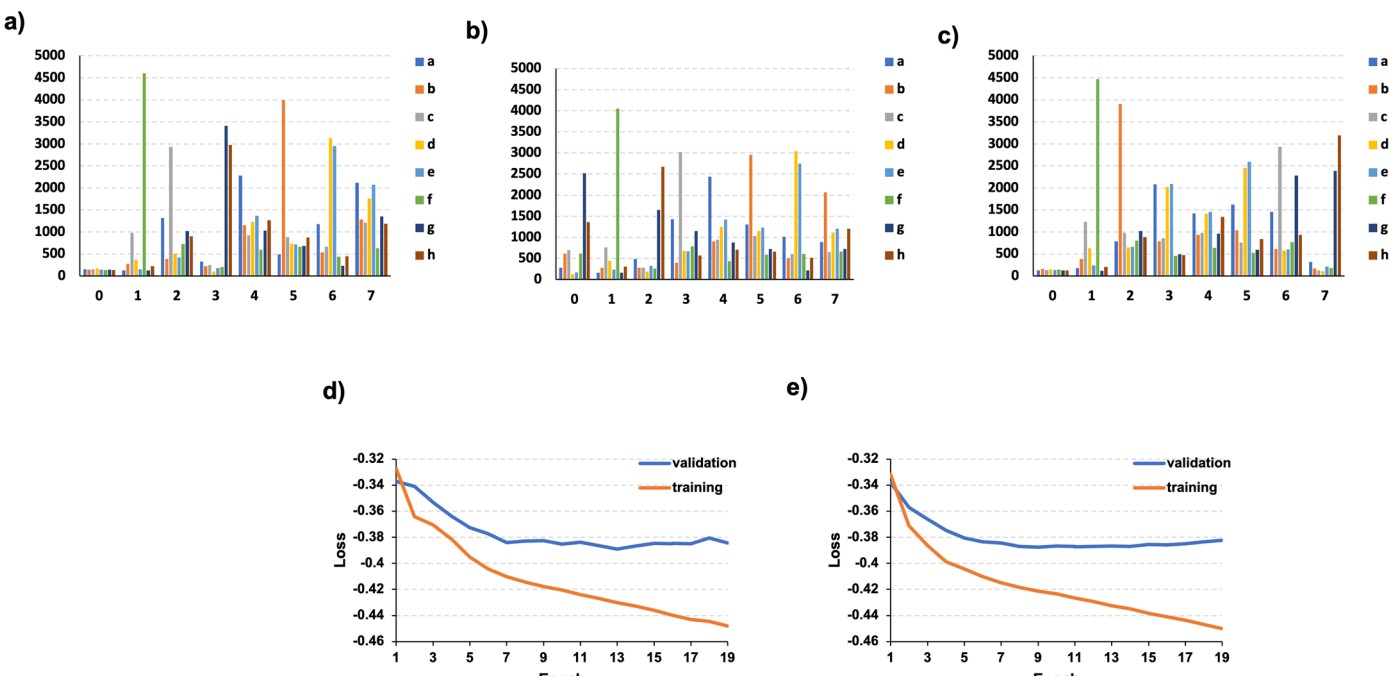

**Extended Data Fig. 7 | Contributions of disentanglement prior and data augmentation to the performance of OPUS-DSD. a)**. Distribution of particles in the latent space of OPUS-DSD with disentanglement prior and data augmentation on the synthetic data of NEXT complex. **b)**. Distribution of particles in the latent space of OPUS-DSD without disentanglement prior. **c)**. Distribution of particles in the latent space of OPUS-DSD without data augmentation. **d)**. Validation and training losses of S.c80S ribosome with data augmentation in OPUS-DSD. **e)**. Validation and training losses of S.c80S ribosome without data augmentation in OPUS-DSD.

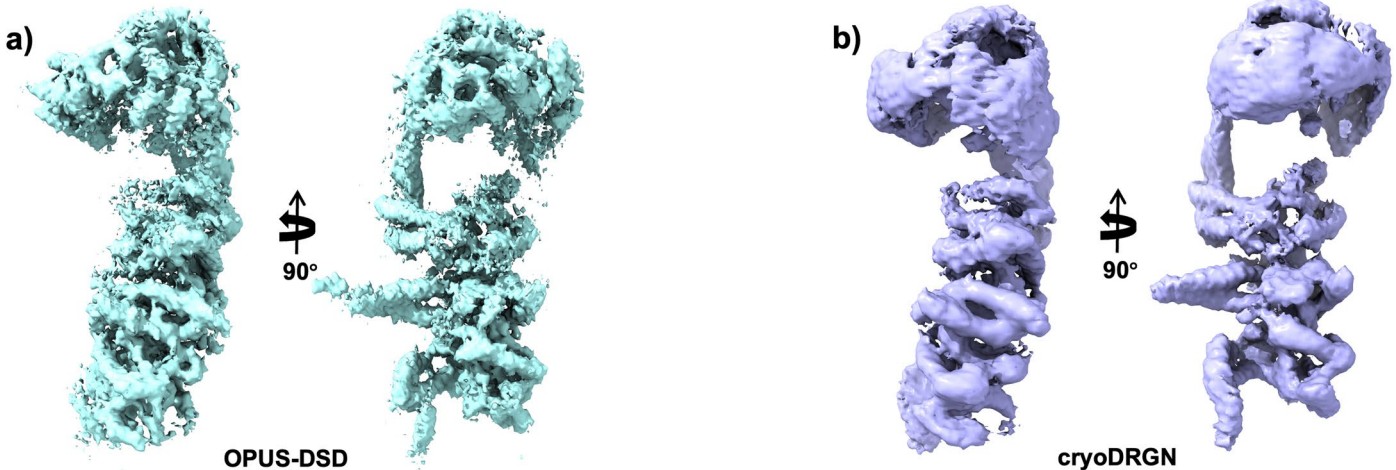

**Extended Data Fig. 8 | Comparison of unmasked half maps. a).** Unmasked half maps from the results of heterogeneity analyses by OPUS-DSD on the 224,354 particles of NEXT complex. **b).** Unmasked half maps from the results of heterogeneity analyses by cryoDRGN on the 224,354 particles of NEXT complex.

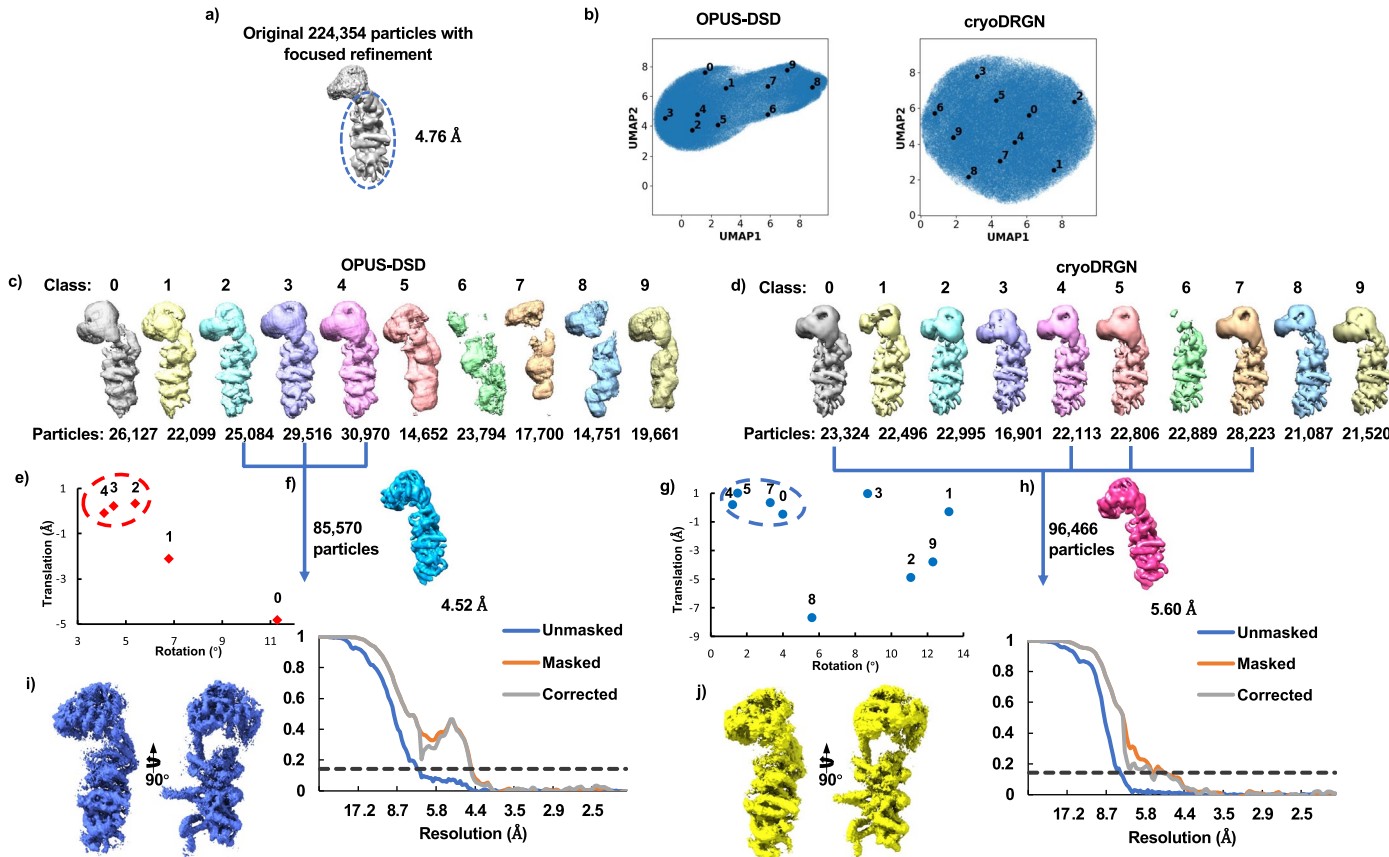

**Extended Data Fig. 9 | Heterogeneity analysis on NEXT complex with focused refinement. a)**. Starting consensus model which was refined with a mask without the MTR4 domain using RELION. **b)**. UMAP visualizations of latent spaces learned by OPUS-DSD (left panel) and cryoDRGN (right panel). The dots with numbers are cluster centers found by Kmeans algorithm. **c)**. Ten classes reconstructed by OPUS-DSD at points shown in **b** and the corresponding number of particles in each class. **d)**. Ten classes reconstructed by cryoDRGN and the corresponding number of particles in each class. **e)**. Translation–rotation plot for reconstructions from OPUS-DSD. The movement is measured for the MTR4 domain relative to the ZCCHC8 domain of each conformation by Dyndom using

the consensus model as a reference. **f)**. Density map and gold-standard FSCs for 85,570 particles by combining OPUS-DSD's clusters with small structural variations (grouped by red circle in **e**). **g)**. Translation–rotation plot for reconstructions from cryoDRGN. The movement is measured for the MTR4 domain relative to the ZCCHC8 domain of each conformation by Dyndom using the consensus model as a reference. **h)**. Density map and gold-standard FSCs for 96,466 particles by combining cryoDRGN's clusters with small structural variations (grouped by blue circle in **g**). **i)**. Unmasked half maps for OPUS-DSD's cluster in **f**). **j)**. Unmasked half maps for cryoDRGN's cluster in **h**). All maps are contoured at the same level.

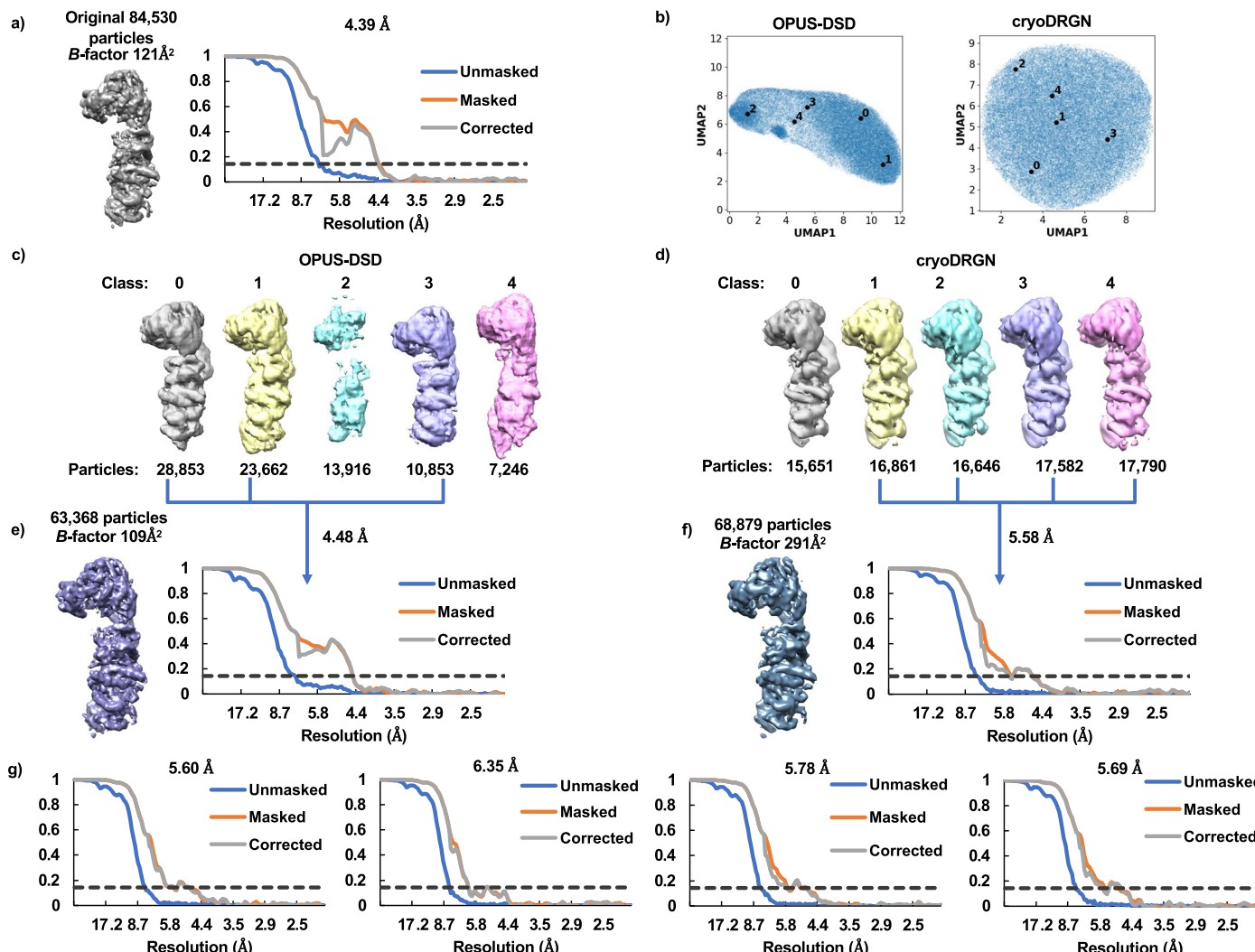

**Extended Data Fig. 10 | Heterogeneity analysis of 84,530 particles of NEXT complex. a).** Starting consensus model and its gold-standard FSCs for all particles. **b).** UMAP visualizations of latent spaces learned by OPUS-DSD (left panel) and cryoDRGN (right panel). The dots with numbers are cluster centers found by Kmeans algorithm. **c).** Five classes reconstructed by OPUS-DSD and the corresponding number of particles in each class. **d).** Five classes reconstructed by cryoDRGN and the corresponding number of particles in each class. **e).** Density map and gold-standard FSCs for 63,368 particles by combining OPUS-DSD's Classes 0, 1 and 3. **f).** Density map and gold-standard FSCs for 68,879 particles by combining cryoDRGN's Classes 1 - 4. **g).** FSCs for reconstructions after randomly discarding a quarter of 84,530 particles in four parallel trials.

# Reporting Summary

## Statistics

For all statistical analyses, confirm that the following items are present in the figure legend, table legend, main text, or Methods section.

| n/a | Confirmed | |
|---|---|---|
| ☐ | ☒ | The exact sample size (*n*) for each experimental group/condition, given as a discrete number and unit of measurement |
| ☒ | ☐ | A statement on whether measurements were taken from distinct samples or whether the same sample was measured repeatedly |
| ☒ | ☐ | The statistical test(s) used AND whether they are one- or two-sided *Only common tests should be described solely by name; describe more complex techniques in the Methods section.* |
| ☒ | ☐ | A description of all covariates tested |
| ☒ | ☐ | A description of any assumptions or corrections, such as tests of normality and adjustment for multiple comparisons |
| ☐ | ☒ | A full description of the statistical parameters including central tendency (e.g. means) or other basic estimates (e.g. regression coefficient) AND variation (e.g. standard deviation) or associated estimates of uncertainty (e.g. confidence intervals) |
| ☒ | ☐ | For null hypothesis testing, the test statistic (e.g. *F*, *t*, *r*) with confidence intervals, effect sizes, degrees of freedom and *P* value noted *Give P values as exact values whenever suitable.* |
| ☐ | ☒ | For Bayesian analysis, information on the choice of priors and Markov chain Monte Carlo settings |
| ☒ | ☐ | For hierarchical and complex designs, identification of the appropriate level for tests and full reporting of outcomes |
| ☒ | ☐ | Estimates of effect sizes (e.g. Cohen's *d*, Pearson's *r*), indicating how they were calculated |

*Our web collection on statistics for biologists contains articles on many of the points above.*

## Software and code

Policy information about availability of computer code

| Data collection | Our data for NEXT complex was collected by SerialEM 4.0. |
|---|---|
| Data analysis | The following tools were used; KMeans clustering algorithm and principal component analysis in scikit-learn 1.3.0, Chimera 1.16, ChimeraX 1.6.1, cryoSPARC v2.4, Relion 3.0.8, umap v0.4.1, python 3.8, pytorch 1.10, cryoDRGN v1.0.0, and OPUS-DSD which is available at https://doi.org/10.24433/CO.3046690.v1 and https://github.com/alncat/opusDSD. |

For manuscripts utilizing custom algorithms or software that are central to the research but not yet described in published literature, software must be made available to editors and reviewers. We strongly encourage code deposition in a community repository (e.g. GitHub). See the Nature Portfolio guidelines for submitting code & software for further information.

## Data

Policy information about availability of data

All manuscripts must include a data availability statement. This statement should provide the following information, where applicable:
- Accession codes, unique identifiers, or web links for publicly available datasets
- A description of any restrictions on data availability
- For clinical datasets or third party data, please ensure that the statement adheres to our policy

We used the following publicly available datasets: EMPIAR-10180 (structure of a pre-catalytic spliceosome), EMPIAR-10028 (cryo-EM structure of a Plasmodium falciparum 80S ribosome bound to the anti-protozoan drug emetine) and EMPIAR-10002 (S.cerevisiae 80S ribosome direct electron detector dataset). Synthetic

and real NEXT datasets were deposited in Zenodo at https://doi.org/10.5281/zenodo.7748967. Trained models and heterogeneity analysis results are deposited at https://doi.org/10.5281/zenodo.8143779 .

## Human research participants

Policy information about studies involving human research participants and Sex and Gender in Research.

| | |
|---|---|
| Reporting on sex and gender | N/A |
| Population characteristics | N/A |
| Recruitment | N/A |
| Ethics oversight | N/A |

Note that full information on the approval of the study protocol must also be provided in the manuscript.

# Field-specific reporting

Please select the one below that is the best fit for your research. If you are not sure, read the appropriate sections before making your selection.

☒ Life sciences        ☐ Behavioural & social sciences        ☐ Ecological, evolutionary & environmental sciences

For a reference copy of the document with all sections, see nature.com/documents/nr-reporting-summary-flat.pdf

# Life sciences study design

All studies must disclose on these points even when the disclosure is negative.

| | |
|---|---|
| Sample size | For publicly available dataset used in this work, the sample size is as it is provided. For synthetic dataset generated in this work, the sample size is 64k, which is of comparable size to previous studies. For example,  cryoDRGN1.0 generates a synthetic dataset with 50k particles. For NEXT complex dataset collected by our own group, the sample size is 224k. The number of particles of our NEXT complex dataset is also comparable to another published research of NEXT complex. In "Structural basis for RNA surveillance by the human nuclear exosome targeting (NEXT) complex" by M. Puno et.al (DOI:https://doi.org/10.1016/j.cell.2022.04.016) , 270k particles were used for 3D refinement. |
| Data exclusions | During training, part of data is randomly excluded from analysis and serves to validate our model. This is a standard technique known as cross-validation in machine learning. |
| Replication | The parameter settings for experiments are detailed in Methods section. We deposited models and codes on codeocean for replication with doi https://doi.org/10.24433/CO.3046690.v1 . |
| Randomization | Neural Networks are trained starting from random initialization weights and the dataset is randomly permuted at each iteration of training. The 3D refinement is also performed with random split of dataset. The training and validation sets are also randomly split. |
| Blinding | Researchers are blinded to expected results as the study is based on unsupervised learning. |

# Reporting for specific materials, systems and methods

We require information from authors about some types of materials, experimental systems and methods used in many studies. Here, indicate whether each material, system or method listed is relevant to your study. If you are not sure if a list item applies to your research, read the appropriate section before selecting a response.

## Materials & experimental systems

| n/a | Involved in the study |
|---|---|
| ☒ ☐ | Antibodies |
| ☒ ☐ | Eukaryotic cell lines |
| ☒ ☐ | Palaeontology and archaeology |
| ☒ ☐ | Animals and other organisms |
| ☒ ☐ | Clinical data |
| ☒ ☐ | Dual use research of concern |

## Methods

| n/a | Involved in the study |
|---|---|
| ☒ ☐ | ChIP-seq |
| ☒ ☐ | Flow cytometry |
| ☒ ☐ | MRI-based neuroimaging |

