## [Peer Review File · Nature Methods]

Peer Review Information

Manuscript Title: OPUS-DSD: Deep Structural Disentanglement for cryo-EM Single Particle Analysis

Corresponding author name(s): Jianpeng Ma

Editorial Notes: n/a

Reviewer Comments & Decisions:

Decision Letter, initial version:

Dear Professor Ma,

I am very sorry for our delay in reaching a decision on your Article, "OPUS-DSD: Deep Structural Disentanglement for cryo-EM Single Particle Analysis". It has now been seen by 3 reviewers. As you will see from their comments below, although the reviewers find your work of considerable potential interest, they have raised a number of important concerns. We are interested in the possibility of publishing your paper in Nature Methods, but would like to consider your response to these concerns before we reach a final decision on publication.

We therefore invite you to revise your manuscript to address these concerns. In particular, we ask that you:

- * Analyze performance of OPUS-DSD using at least one more "standard" dataset (Reviewer #1 suggests the splicesome).
- * Show movies as suggested by Reviewer #1.
- * Please ensure that you are utilizing optimal parameters for running cryoDRGN in your comparisons, and that you run comparisons using the same datasets. We request that you also perform comparisons to e2gmm (<https://www.nature.com/articles/s41592-021-01220-5>).
- * Address all other technical issues.

* Tone down strong claims of superiority to cryoDRGN (see in particular the comments from Reviewer #3).

When revising your paper, please also:

- * include a point-by-point response to the reviewers and to any editorial suggestions
- * underline/highlight any additions to the text or areas with other significant changes to facilitate review of the revised manuscript
- * address the points listed described below to conform to our open science requirements
- * ensure it complies with our general format requirements as set out in our guide to authors at www.nature.com/naturemethods
- * resubmit all the necessary files electronically by using the link below to access your home page

[Redacted] This URL links to your confidential home page and associated information about manuscripts you may have submitted, or that you are reviewing for us. If you wish to forward this email to co-authors, please delete the link to your homepage.

We hope to receive your revised paper within 4 months. If you cannot send it within this time, please let us know. In this event, we will still be happy to reconsider your paper at a later date so long as nothing similar has been accepted for publication at Nature Methods or published elsewhere.

OPEN SCIENCE REQUIREMENTS

REPORTING SUMMARY AND EDITORIAL POLICY CHECKLISTS

Please note that these forms are dynamic ‘smart pdfs’ and must therefore be downloaded and completed in Adobe Reader. We will then flatten them for ease of use by the reviewers. If you would like to reference the guidance text as you complete the template, please access these flattened versions at <http://www.nature.com/authors/policies/availability.html>.

DATA AVAILABILITY

We strongly encourage you to deposit all new data associated with the paper in a persistent repository where they can be freely and enduringly accessed. We recommend submitting the data to discipline-specific and community-recognized repositories; a list of repositories is provided here:

<http://www.nature.com/sdata/policies/repositories>

All novel DNA and RNA sequencing data, protein sequences, genetic polymorphisms, linked genotype and phenotype data, gene expression data, macromolecular structures, and proteomics data must be deposited in a publicly accessible database, and accession codes and associated hyperlinks must be provided in the “Data Availability” section.

To further increase transparency, we encourage you to provide, in tabular form, the data underlying the graphical representations used in your figures. This is in addition to our data-deposition policy for specific types of experiments and large datasets. For readers, the source data will be made accessible directly from the figure legend. Spreadsheets can be submitted in .xls, .xlsx or .csv formats. Only one (1) file per figure is permitted: thus if there is a multi-paneled figure the source data for each panel should be clearly labeled in the csv/Excel file; alternately the data for a figure can be included in multiple, clearly labeled sheets in an Excel file. File sizes of up to 30 MB are permitted. When submitting source

data files with your manuscript please select the Source Data file type and use the Title field in the File Description tab to indicate which figure the source data pertains to.

Please include a “Data availability” subsection in the Online Methods. This section should inform readers about the availability of the data used to support the conclusions of your study, including accession codes to public repositories, references to source data that may be published alongside the paper, unique identifiers such as URLs to data repository entries, or data set DOIs, and any other statement about data availability. At a minimum, you should include the following statement: “The data that support the findings of this study are available from the corresponding author upon request”, describing which data is available upon request and mentioning any restrictions on availability. If DOIs are provided, please include these in the Reference list (authors, title, publisher (repository name), identifier, year). For more guidance on how to write this section please see: <http://www.nature.com/authors/policies/data/data-availability-statements-data-citations.pdf>

CODE AVAILABILITY

Please include a “Code Availability” subsection in the Online Methods which details how your custom code is made available. Only in rare cases (where code is not central to the main conclusions of the paper) is the statement “available upon request” allowed (and reasons should be specified).

For more information on our code sharing policy and requirements, please see: <https://www.nature.com/nature-research/editorial-policies/reporting-standards#availability-of-computer-code>

MATERIALS AVAILABILITY

ORCID

Sincerely yours,
Allison

Allison Doerr, Ph.D.
Chief Editor
Nature Methods

Reviewers' Comments:

Reviewer #1:

Remarks to the Author:

Luo et al. describe a new algorithm (OPUS-DSD) that employs deep neural networks and an encoder-decoder architecture to reconstruct density maps from cryo-EM images of heterogeneous samples. encoder-decoder architecture based solutions to this problem have been described in the past (e.g. cryoDRGN) but the described method differs in overall architecture and also through additional regularizations that are included to keep the latent well separated between groups, but located for similar classes. In addition, the method keeps the images and volumes in real-space which is a significantly different approach. In order to demonstrate the results of their algorithm, the authors demonstrate it on 3 different datasets of the NEXT complex, one synthetic, one experimental, and

thirdly the experimental dataset after focused classification. For each of these, they make comparisons to cryoDRGN. They also demonstrate the results on a ribosome dataset.

In general the algorithm appears well designed and thought out. If it functions as claimed it is certainly of general interest to the field, and the additional regularizations and different architecture appear to be interesting and well described.

My concern is that the results as presented make it very hard to validate the claims and effectiveness of the method. In particular, all the results presented are essentially demonstrated as classifications, whereby the authors attempt to find clusters in latent space, then make reconstructions with those clusters. As the authors themselves state in the introduction, 3D classification has existed in the cryo-EM field for many years, and neural network approaches are really designed to handle more complex heterogeneity such as continuous motion. None of the examples provided demonstrate this, the synthetic example is specifically designed with 8 classes, whereas similar published work all includes a synthetic example with continuous motion. Another example is that similar methods all demonstrate traversal of the latent space and provide movies displaying the structural changes that occur. These movies provide evidence that the method is more than a simple 3D classifier, and it would be great to see similar examples and movies included for this method.

Much space is given to being “better” than cryoDRGN on the NEXT examples. Personally I do not believe this to be helpful for a number of reasons. Firstly, it relies on the authors being able to run cryoDRGN as well as they can run their own method, and as well as the authors of cryoDRGN could run it which is not likely. Secondly it is only a singular example, so not very informative as to the general behavior. It would be much more useful to provide results on standard datasets that have been used in the field, e.g. splicesome - this then enables comparison to results authors of different methods calculated themselves and the characteristics of these datasets are well understood. For example, for the ribosome dataset the authors state that “CryoDRGN performed structural heterogeneity analysis on a large 80S ribosome dataset (EMPIAR-10028) which contained 104,280 images in its original publication. Here OPUS-DSD is used to analyze the structural heterogeneity of a smaller dataset of 80S ribosome containing 60,363 images (EMPIAR-10002)”. Why not use the same dataset, particularly as so much time is given to a comparison with cryoDRGN.

I have a number of specific comments that the authors may wish to address :-

1. As above, my main comment is that the provided examples are not sufficient to validate the claims. For example, in the abstract the authors state their method can “reliably reconstruct the structural landscape”. The provided examples do not really demonstrate this. What is needed to validate this is an example with continuous motion, where the authors actually traverse their latent space and

demonstrate they can obtain reconstructions along the continuum (e.g. showing movies). More examples on publicly available and well studied datasets with complex heterogeneity would also be highly beneficial, for example the EMPIAR-10180 spliceosome dataset that has been included as an example in the cryoDRGN paper, the Relion multi-body refinement paper, the cryoSPARC methods papers for 3dflex and 3DVA, the EMAN2 mixed-dimensional Gaussian technique and likely others I am not remembering.

2. The FSCs shown for their results on the NEXT complexes all look worrying and artifactual. The corrected FSCs have a severe dip that in many cases just stays above 0.143, then rises back up, hovers around 0.5 for a while before finally dropping. The unmasked FSCs for both their and the cryoDRGN results are similar, it would be expected then that the masked FSCs would also be similar, but the cryoDRGN results have lower resolution, though do not have the artifacts stated above. The authors should explain these FSCs and perhaps provide unmasked half maps so that they can be validated.

3. As in other methods, the decoder presented here creates volumes directly from latent space variables. However, no examples are provided either as an image, or as movies traversing the space. What are the limitations of these volumes, and what is their quality? E.g. as I understand it, the geometry of the network is fixed. How high resolution can reconstructions be made directly from the latent space, is it possible to obtain high resolution in the generated maps? it would be nice to have some discussion of this and to show some examples.

4. The method included regularization to push different volumes to be separated in latent space, and similar volumes to be grouped together. My impression is that this leads to better clustering in latent space, and is likely beneficial for separating defined states. How does this method affect more continuous motions where defined clusters should not exist, does it push them into clusters, or do they remain as a continuum?

4. "To make the network to be invariant to the defocus variation, the network uses input image with its contrast and brightness randomly adjusted to reconstruct the original image." More information should be provided on this, how is it done exactly.

5. Clusters are identified via k-means clustering and the manuscript states "The informative latent space with well separated clusters should be easily clustered by the simple KMeans clustering algorithm". Is it possible that the results obtained from cryoDRGN are simply a failure of the clustering, perhaps their latent space is not as well organized for example? Is it possible to find better reconstructions for cryoDRGN by simply calculating 3Ds for various regions of the latent space?

6. In all examples the latent space dimension is set to 8, is 8 the optimum number for every experiment, a small discussion on this would be appreciated.

7. The authors should at least mention other neural network based techniques designed to solve similar problems such as the EMAN2 GMM technique and 3Dflex.
8. On page 17 there is a reference which simply states "Error! Reference source not found."

Reviewer #2:

Remarks to the Author:

The manuscript by Luo et al. presents a deep learning based method to analyze structural heterogeneity of macromolecules from single-particle cryo-EM data.

Structural heterogeneity is still a major challenge in cryo-EM structure determination and is important for two reasons:

1) Analyzing structural heterogeneity provides information on conformational dynamics which is often related to function, and 2) disentangling different structural states will help increase the resolution (for each state).

The presented method (OPUS-DSD) is therefore a valuable and timely addition to the existing cryo-EM toolbox. OPUS-DSD was tested on simulated and experimental data and overall the performance is well documented.

The results from OPUS-DSD are compared to those from the previous published method cryoDRGN. From reading the text, it sounds like cryoDRGN and OPUS-DSD are two separate methods. However, by looking at the code, it becomes clear that OPUS-DSD is based on cryoDRGN and includes substantial amount of code from cryoDRGN. I would therefore consider OPUS-DSD more like an extension/improvement of cryoDRGN. Therefore, the relationship of OPUS-DSD with cryoDRGN should be made clearer in the main text.

To be clear, there are indeed substantial changes with respect to cryoDRGN which clearly improve the results and warrants separate publication. In particular, OPUS-DSD introduces a prior in latent space, which helps to separate variations due to orientational changes and defocus from the structural changes of the particle, leading to improved results.

Other comments:

- Fig 2b shows the 8 different structural states (a-h) that were used for the simulated dataset. Fig 2d shows the resulting Umap visualization with cluster centers. Would it be possible to add the 8 input structures also to the Umap visualizations? This would show much better how well the clustering captures the actual (ground-truth) structures.
 - in addition or alternatively: It would be good to show rotation/translation plots in Fig 2 similar to Fig 3e. This would allow for a direct comparison of the ground-truth structures (the 8 states) with the states obtained by OPUS-DSD and cryoDRGN.
 - Fig. 4e: why are only 5 out of the 10 states shown in this plot?
 - Figs. 3f, 4f, 5e: The "masked" and "corrected" FSC curves from the OPUS-DSD look strange, (the unmasked is fine), whereas these curves look ok for the corresponding cryoDRGN results. Can this be explained?
 - Figs 2g,h; 3c,d; 4c,d; 5c,d; how was the density threshold chosen when comparing the different maps?
 - it might be helpful to proofread the manuscript as it contains a number of grammatical errors and missing articles.
- minor points:
- Title Figure 4: "Heterogeneous analyses", maybe better "Heterogeneity analysis" (since the analysis is not heterogeneous).

Reviewer #3:

Remarks to the Author:

The work presented here is interesting, promising, and partly innovative.

This work follows the general architecture of CryoDRGN, which is a popular implementation of continuous heterogeneity analysis in cryo-EM that introduced a VAE-like architecture to the field. The current work introduces several interesting new ideas which seem to provide material advancements overall. The main innovations:

* The encoder component of the VAE architecture, which in cryoDRGN takes only the image, is replaced with an encoder that takes the image and a pose. Inside the encoder, it cleverly back-projects from 2D to 3D, then uses a 3D convolutional neural network to produce a latent variable.

* The decoder is a 3D voxel-based decoder. Voxels are the traditional way of representing volumes in cryoEM. I believe that a conceptually similar component has been used in Multi-CryoGAN, I don't recall seeing it in VAE-type work in cryo-EM though.

* A clever regularization (referred to as prior) on some of the variability in the latent space and volume space.

* Data augmentation to mitigate the effect of contrast variations (invariance to contrast transfer function(CTF) is claimed).

There are a few experiments comparing the new software, mainly with cryoDRGN. The results are compelling. There are no standard benchmarks in this community, and there are many parameters that can be adjusted, so proving that one tool is really superior would mean clearing a very high bar. One may ask for additional experiments, but I don't think that the experiments presented here are unreasonable (other than the lack of run time information - see below) to support the relevance of this method - superiority would mean clearing a much higher bar.

That being said, some additional experiments, especially experiments that reproduce the exact setup given in cryoDRGN examples (where the authors of cryoDRGN may have also optimized the parameters), would make the authors' argument more convincing (if some of the current experiments do this, it would be good to emphasize this fact).

However, the claims made throughout the paper, especially in the discussion, are not supported in the paper. The paper needs extensive revisions in the presentation of the work. My opinion of the work is overall favorable. The presentation of the work (and things that are not supported in the work) is my concern.

I would divide my remarks into two categories:

* The issues with clarity in inaccurate statements are unfortunate (very unfortunate given their prevalence), and diminish the quality of a paper about otherwise nice work. That being said, they are somewhat harmless. The same might apply to the omissions of the background and state of the art. I do not think that the extent of these inaccuracies is appropriate for a paper in this journal, but they seem to be harmless.

* The statements related to a strict superiority over cryoDRGN effectively say that there are overwhelming peer-reviewed arguments and evidence that this software should replace cryoDRGN. While this reviewer does not rule out that this tool might be strictly superior (pose updates aside), this reviewer disagrees that such claims are supported by evidence in this paper.

Given the extent of the issues in this paper, my recommendation is to reject the paper, leaving the door open for the resubmission of a heavily revised manuscript. Alternatively, the paper can go to major revisions, with extensive rewriting.

This reviewer has no reason to favor cryoDRGN. This reviewer would be excited to see software that supersedes cryoDRGN. The examples in my review below are not presented in order to justify my recommendation to reject the paper (I think the summary above would have been sufficient), but rather in the hope that the authors would take some of the comments into consideration in rewriting the paper, resulting in a timely publication of a high-quality paper about this interesting work.

I should note that it is likely that I missed a detail here and there, but my overall impression of the accuracy of the claims is pretty strong. The central claim about the comparison to cryoDRGN alone is sufficient.

Technical questions:

=====

Backprojection of the image to a volume: please explain which numerical approach is used.

The reference to a Spatial transformer and neural volume, with the citation provided for it, is unclear. Do the authors refer to one of the standard volume rotation computations or to an actual "fancy" transformer? In either case, they need to be more specific. Does the decoder produce a grid that is then rotated, or a more elaborate neural volume?

Data augmentation: what exactly is varied, and how?

There are some unclear references to total variation and LASSO in the paper. Are these implemented or suggested? Where do they fit in? This would mean that there are many different forms of regularization going on. How do these compare or fit in with the other regularization proposed by the authors? How sensitive is the algorithm to the weights on these various regularizations?

Comparison questions:

=====

Which version of CryoDRGN is used? The current version of cryoDRGN allows pose updates, which the proposed software does not allow, and which the paper seems to ignore.

Run time (and other resources): the statement about timing for 1000 particles (at what image size? is this a single iteration?) is insufficient and potentially misleading. It also does not provide any comparison to the CryoDRGN benchmark. There can be serious concerns about timing for all the usual reasons, but also since this algorithm uses 3D representations where the CryoDRGN benchmark uses 2D representations. The authors may have found an efficient way to implement the operations, and the optimization may require a small number of epochs/iterations, but this is not obvious.

This reviewer recognizes that precise timing reports do not make sense here: there is no obvious point at which to stop the iterations in cryoDRGN (there is a method for stopping, which is not fully explained, available in the current work). However, some information about the time it took to run some of the examples in the paper would be appropriate (and to make the comparison clearer, it might be useful to show to significantly fewer iterations do not yield a good result).

This reviewer finds this software potentially useful even if it is found to be significantly slower than CryoDRGN, but the information should be made available to the reader. Surely, if the software is considerably slower, it is more difficult to argue that the software is strictly superior.

On a related note, what GPUs are needed to run this software at reasonable image sizes? The 3D grids could require a considerable amount of GPU RAM. The only discussion of hardware here mentions relatively high-end hardware - does the software work adequately with a single GPU? With lower-end GPUs? Again, the software can be useful even if it requires higher-end hardware, but the bar is much higher if the authors claim the strict superiority claimed here.

Similarly, RAM/storage might also be an issue if some of the backprojections are stored.

The authors make some arguments about certain latent space representations (their results) being advantageous over others (cryoDRGN results). What is a "correct" or "good" representation is actually a complicated question. The authors could state certain properties that may be advantageous in some aspects (e.g., some visual separability when a dimensionality reduction algorithm is used downstream, may be preferable from the point of view of a practitioner). They actually provide reasonable motivation around line 126, but these are different from the arguments made in the comparison to cryoDRGN. The comparisons to cryoDRGN are not unreasonable, they just require some additional discussion of motivation.

Can the authors comment on the importance of different components (where they tried)? There are many statements about the role of these components made in the paper, but most of them are not supported in the paper. At most, the paper demonstrates some overall potential advantages but stops short of showing what these potential advantages could be attributed to. Many of the claims about a particular component contributing in a particular way sound like hypotheses at best. It is difficult to quantify such hypotheses with so many moving parts, but it might be possible to provide evidence for some of these claims with some additional work. Stating so many of these hypotheses as clear

undisputable facts is a stretch. Stating these as clear advantages over cryoDRGN does not seem to be supported by evidence in the work, despite the evidence in the experiments for potential advantages over cryoDRGN of the software as a whole.

In other words, I wonder which of these components are necessary and the extent to which they help. I don't think that the authors should try every possible combination of parts of CryoDRGN and their software. Instead, they can rephrase some of their hypotheses and add a few small experiments. A couple of experiments that seem plausible without too much work:

1. Could the authors present an example with the contrast data augmentation turned off?
2. Could the authors turn off the loss function on the latent space embedding? This might throw the latent variable all over the place, so if the authors happen to have an implementation of the classic VAE loss used in cryoDRGN it would be nice to see how that compares (if the implementation takes a lot of work I don't think it is necessary).

Scope of discussion

=====

The software requires fixed poses to be provided from upstream algorithms. This can be a critical issue for large heterogeneity and for higher resolution reconstructions. The more recent cryoDRGN can update poses (in fact, it has ab-initio functionality), and cryoDRGN is not the first to demonstrate pose updates. It would be legitimate to say that the implementation of this feature in this software is left for future work. The current discussion of the issue (or lack thereof) is insufficient, if not borderline misleading. The paper seems to suggest that the algorithm might have some immunity to inaccurate upstream poses, a claim for which I don't see any supporting evidence. If the authors argue that the software is immune to the issue, much stronger evidence is needed. I think that the actual claim about immunity to upstream pose inaccuracy is more subtle, but this is not clear and not supported by evidence (and I'm not sure it is correct either).

Some remarks about the statements and claims:

=====

Given the large number of issues, I will discuss only some of the problems I found. The main issues appear in the discussion section, but the most significant problematic claim about the superiority of this software is in the introduction. I put "[sic?]" where I think that language editing might be needed (in addition to my comment about the content of the statement).

Introduction:

The paper correctly identifies CryoDRGN as a good benchmark in the context of deep learning (although somewhat similar concepts of getting 3D volumes from 2D images can be traced back even to the covariance method). However, the area of continuous heterogeneity (using deep learning methods and other methods) is broad, some of the contributions attributed to cryoDRGN are not unique to it, and it would be appropriate to add some citations about other current and previous work in the area, especially work related to the ideas here (e.g., multicryoGAN in the context of volume representation if applicable, previous discussions of regularization of the volume, etc.).

The current presentation of the state of the art is lacking.

“By systematically testing on synthetic and real cryo-EM datasets, we demonstrated the superior performance of OPUS-DSD on [sic?] resolving structural heterogeneity, in terms [sic?] of both compositional changes and conformational dynamics, in cryo-EM datasets compared to cryoDRGN. OPUS-DSD can not only provide the necessary structural heterogeneity information to deepen our understanding about [sic?] the dynamics of macromolecular systems, but also improve the resolution of highly flexible system[sic?] by providing a more homogenous[?] dataset.” The experiments are nice, and it's appropriate to say that they demonstrate the potential advantages of this work, but this community doesn't have a truly systematic test or even an accepted test. The tests here are encouraging but anecdotal. The statements here should be more modest. The lack of discussion of run time makes this claim even more questionable.

Otherwise, while I would suggest revisiting the introduction for better clarity, it conveys much of the motivation for this work without making other outrageous claims.

Discussion

The discussion overstates many things and should be rewritten. Almost every sentence is either unclear, overstates the results, makes claims that are not supported by evidence in the paper or misrepresent other work, or is at least somewhat ambiguous and may be misinterpreted by the readers. Some of the items are treated a little more carefully in the preceding sections, but the statements in the discussion are questionable.

“The ill-posedness is alleviated by encouraging the smoothness of the latent space w.r.t structural variations and the invariance of latent codes w.r.t poses and defocus. A smooth latent space guarantees that similar 3D structures are encoded by similar latent codes, while a pose invariant latent space makes the latent encodings of 2D images solely depend on their 3D structures.” The general idea of this contribution of smoothness is discussed in previous works on the continuous representation of continuous heterogeneity. It is also presumably present implicitly even in cryoDRGN, through the implicit/inductive bias of the deep network. The current paper does introduce new ideas for implementing explicit regularization - this is not my most significant issue with the discussion.

“Therefore, the quality of 3D density map can be improved by 2D supervisions with diverse views and similar underlying 3D structures during training.” Not a very clear sentence.

“OPUS-DSD also leveraged data augmentation to learn a defocus invariant latent encoding.” The augmentation potentially reduces the dependence on contrast, but this does not immediately mean that it reduces the dependence on CTF (which is not uniform over all frequencies), let alone making it “invariant”. Achieving strict mathematical invariance is difficult - I doubt that if the authors ran an experiment, they would see actual invariance. It would be encouraging (and interesting!) if they do see reduced sensitivity. The most that can be said is something like “mitigation,” “reduced sensitivity,” or perhaps a colloquial “more invariant” - but even this is not really demonstrated in the paper, only hypothesized (Maybe it is better supported by experiments that the authors did not include?). More generally, most mentions of “invariance” in the paper are too liberal and not supported by evidence in the work. They should be more accurate.

“The neural network in cryoDRGN also relies on an explicit pose encoding to model the 2D projection.” This is no longer true for recent versions of CryoDRGN (and CryoDRGN is not the first to demonstrate this feature). The authors go beyond this to say that this dependence on pose is a weakness of cryoDRGN - there is very little in the current paper that addresses this issue. The authors have a clever idea about backprojecting the image to 3D - it is a very pleasant surprise to me to see that this works despite the missing data (it is easier to see in the Fourier domain that the backprojection does not solve the missing data problem than it is in the spatial domain) that goes into the convolutional network, but to say that it solves problems such as wrong pose going into the algorithm is much more than is actually demonstrated in this paper. Maybe I don't understand what the word “this” means when they say “This approach may be susceptible to the pose assignment errors”: I would say that this issue applies equally to both methods as long as there is no further evidence. There seem to be two issues packed into this:

the difference in what is viewed from each pose and the inaccuracy in the pose. It is not clear what point is made about either of these.

``_These_ features of cryoDRGN yield unsatisfactory 3D density maps on datasets with low SNRs." Maybe I don't understand this correctly, but weren't the experiments with variable SNR synthetic experiments where the ground truth viewing direction is known? In that case, how is the pose inaccuracy relevant?

``The systematic errors in 3D density maps will propagate into latent space during training and ultimately lead to noninformative latent encodings, which yields spurious structural heterogeneity resolving results." Not a clear or supported statement. Again, not my most significant concern.

``These architectural drawbacks in cryoDRGN are demonstrated using both synthetic and real datasets in our study here." Not that I have seen. The authors demonstrate some potential advantages but have not shown where they would be traced to. For all I know, it can just be a choice of parameters. Again, I am not saying that the authors cherry-picked, or even that they cannot demonstrate potential with some level of cherry-picking, I am saying that the evidence is not sufficient for such a strong claim. The results are nice enough without statements like these. I would say that the experiments demonstrate something about performance, but I don't think that it's sufficient for a statement about architectural drawbacks.

``On the other hand [sic?], OPUS-DSD overcomes these peculiarities by adopting a 3D convolutional architecture [10]." Is this a statement about the encoder or the decoder? It's not clear what contrast is drawn here.

``... Instead of representing the 2D projection using explicit pose encoding, OPUS-DSD reconstructs the 2D projection by directly projecting a 3D volume." I assume that this is about the encoder. It's a clever idea, but I don't see how it overcomes issues such as misaligned poses without further evidence. As far as I can tell, it is as likely to improve things as it is likely to make them worse (the misaligned pose is now fed into the encoder, not only the decoder). Something seems to work nicely here in practice in this software, but the claims are too strong and not supported with evidence.

Saying ``Instead of representing the 2D projection using explicit pose encoding, OPUS-DSD reconstructs..." makes it sound like CryoDRGN encodes the pose from the image (it doesn't) or perhaps that this encoder does not consider the pose or is invariant to it. In fact, technically speaking, the CryoDRGN encoder is ``invariant" to the inaccurate pose input (it does not take in the pose as an input variable), whereas the encoder in this work is not invariant (it backprojects using the pose). This does not make either of them better or worse, it is just very unclear what the authors meant to say here. I think I can see where the authors are going with this, but it's not what is written in the paper.

“This convolutional architecture appears to be robust to high level of noises [sic?] and incorrect pose assignments in 2D cryo-EM images.” Where is this shown? I think that it is fair to say that the software performs well in high noise (maybe robust is ok), but I am not sure where there is evidence for robustness to incorrect poses. Furthermore, the more recent CryoDRGN updates pose estimates, an issue that is completely ignored here.

My inclination would actually be to attribute the SNR performance to the decoder (implicit regularization), not the encoder. Where is the role of this particular component of the encoder shown? The authors might want to point to something specific such as quality of class assignment as some potential evidence for the argument about SNR.

“It also focuses on capturing large-scale structural variations while being resilient to the irrelevant high-resolution structural information and noises.” Unclear. Is the same point repeated?

“An advantage of this approach is that it allows one to further improve the fidelity of the 3D density maps by using the traditional sparseness and smoothness regularizers such as total variation and LASSO (15, 23, 24).” Not clear. Are we talking about the regularizer now? Are these implemented? Where?

“Moreover, OPUS-DSD employs a validation set to objectively monitoring [sic] the neural network training process, thus further reducing the problem of overfitting nonstructural heterogeneity.” Mentioned only later in the paper, and not really explained, as far as I can see. It is difficult to say if the validation set really validates, but it would be very interesting and useful even if it only gives an indication of when to stop.

“In such cases, rigid-body analysis such as multibody refinement (6) would be less efficient”. This statement might be interpreted as a unique feature of this software, which it is not. It applies to cryoDRGN and many other approaches to continuous heterogeneity. Since a comparison to multi-body refinement isn't presented here, I suggest that plausible statements like these would be phased as “likely”, “potentially” etc. or rely on a citation of some other papers.

“... More importantly, OPUS-DSD has the advantage of exploring conformational and compositional changes in a unified framework.” Again, a reader might mistakenly think that this is a new property of this software. It applies to many (but not all) of the approaches to continuous heterogeneity.

“In conclusion, OPUS-DSD is a robust, accurate and versatile tool for resolving structural variations. It can not only deepen our understandings about [sic?] the dynamics of biological systems but also facilitate structure determination by providing more homogenous datasets to achieve higher resolutions. ”

Again, it is too early to say these things based on anecdotal. I might agree that it is promising. What are "more homogenous datasets" ?

Other comments

"The decoder reconstructs a 3D volume according to the latent code. Total variation and LASSO priors are imposed on the reconstructed 3D volume to encourage its smoothness and sparseness (23, 24)". Please explain where these come in.

"The number of samples for interclass comparison is " Please explain (term comes up here the first time).

"However, OPUS-DSD selected models for further analyses according to the reconstruction error on the validation set." Please explain.

"The informative latent space with well separated clusters should be easily clustered by the simple KMeans clustering algorithm (34)" Did you verify that the clusters produced were consistent? It's not always the case with kmeans, but it might be the case here.

It might be worthwhile to say that cryoDRGN works in the Fourier domain, so it does not need to explicitly integrate 3D volumes. This is relevant for the computational cost, but it might not be self-explanatory to every reader without further discussion, so it is not crucial to add this.

"Prior" terminology. I am not sure that the proposed regularization easily translates to formal prior. "Regularization" might be more appropriate. I don't have a very strong opinion about this as long as the actual procedure is explained.

Author Rebuttal to Initial comments

Dear Dr. Doerr,

Thank you very much for conducting a very productive review on our manuscript! We have followed your suggestions and the suggestions of the reviewers and thoroughly revised the manuscript. Here are the point-to-point responses to all comments.

1. Analyze performance of OPUS-DSD using at least one more "standard" dataset (Reviewer #1 suggests the spliceosome).

Response: Thank you very much for the suggestion! We have additionally analyzed EMPIAR-10180 spliceosome and EMPIAR-10028 Pf80S ribosome. The results further validated the performance of our method.

2. Show movies as suggested by Reviewer #1.

Response: Thank you very much for this suggestion! We have included four movies generated by OPUS-DSD and one movie by cryoDRGN in this revision. These movies clearly demonstrate that OPUS-DSD can reconstruct continuous dynamics. In addition, the improved reconstruction quality of OPUS-DSD allows the recovery of compositional changes accompanying large-scale structural dynamics.

3. Please ensure that you are utilizing optimal parameters for running cryoDRGN in your comparisons, and that you run comparisons using the same datasets. We request that you also perform comparisons to e2gmm (<https://www.nature.com/articles/s41592-021-01220-5>).

Response: For EMPIAR-10180 spliceosome and EMPIAR-10028 Pf80S ribosome that cryoDRGN and other methods used, we directly compared the results from OPUS-DSD with those reported in their respective publications. For NEXT complex, we systematically optimized the parameters and used the optimal values for running cryoDRGN in our comparison. All comparisons are performed on the same datasets.

Lastly, the results of OPUS-DSD on EMPIAR-10180 spliceosome were also compared to those of e2gmm. In the case of EMPIAR-10180 spliceosome, both OPUS-DSD and cryoDRGN detected large compositional heterogeneities, which were not seen by e2gmm. The movie generated by OPUS-DSD also exhibits much higher resolutions than those from cryoDRGN or e2gmm.

4. Address all other technical issues.

Response: Yes, all technical issues have been addressed in the responses below and incorporated in the manuscript.

5. Tone down strong claims of superiority to cryoDRGN (see in particular the comments from Reviewer #3).

Response: Yes, we revised the texts accordingly.

Reviewer #1

1. As above, my main comment is that the provided examples are not sufficient to validate the claims. For example, in the abstract the authors state their method can “reliably reconstruct the structural landscape”. The provided examples do not really demonstrate this. What is needed to validate this is an example with continuous motion, where the authors actually traverse their latent space and demonstrate they can obtain reconstructions along the continuum (e.g. showing movies). More examples on publicly available and well studied datasets with complex heterogeneity would also be highly beneficial, for example the EMPIAR-10180 spliceosome dataset that has been included as an example in the cryoDRGN paper, the Relion multi-body refinement paper, the cryoSPARC methods papers for 3dflex and 3DVA, the EMAN2 mixed-dimensional Gaussian technique and likely others I am not remembering.

Response: Thank you for your wonderful suggestions. We included the pre-catalytic spliceosome dataset (EMPIAR-10180) and Pf80S ribosome (EMPIAR-10028) in our revised manuscript. For each system, we traversed the first principal component of latent space and obtained expected dynamics. For example, for EMPIAR-10180 spliceosome, traversal along the PC1 in **Mov.1** reveals the continuous dynamics between the open and closed conformations, highlighting the concerted movements of the subcomplexes, i.e., the Helicase bends towards the Foot while the SF3b folds onto the Core. There are also some conformational changes which were not reported in previous analyses. Overall, the resolutions of structures shown in **Mov.1** are higher than those demonstrated by cryoDRGN or e2gmm. In the case of EMPIAR-10028 ribosome, the UMAP of the latent space learned by OPUS-DSD shows large differences compared to that by cryoDRGN. **Mov.2** generated by OPUS-DSD clearly demonstrates how an RNA slides out of its original site in SSU and docks onto LSU. **Mov. 3** generated by OPUS-DSD shows movements with higher continuity and larger range-of-motion than the movie generated by cryoDRGN in the same region.

The main advantage of the end-to-end methods such as cryoDRGN and OPUS-DSD is that they can explore compositional and conformational heterogeneity in a unified framework, and resolve concerted movements. However, the advantage of multi-body refinement is that it can obtain high-resolution structures for dynamic subcomplexes. In the future, there should be a way to combine the advantages of end-to-end methods and multi-body refinement method, thus obtaining high-resolution concerted movements.

2. The FSCs shown for their results on the NEXT complexes all look worrying and artifactual. The corrected FSCS have a severe dip that in many cases just stays above 0.143, then rises back up, hovers around 0.5 for a while before finally dropping. The unmasked FSCs for both their and

the cryoDRGN results are similar, it would be expected then that the masked FSCs would also be similar, but the cryoDRGN results have lower resolution, though do not have the artifacts stated above. The authors should explain these FSCs and perhaps provide unmasked half maps so that they can be validated.

Response: Thank you very much for your questions. The described behavior that “the corrected FSCs have a severe dip that in many cases just stays above 0.143, then rises back up, hovers around 0.5 for a while before finally dropping” may not be purely artifacts, rather it may represent meaningful improvements in this resolution range due to the reduction of structural variations in the MTR4 domain by OPUS-DSD. The following observations seem to support this explanation. First, the unmasked half maps of clusters obtained by heterogeneity analyses of NEXT complex by OPUS-DSD (**Fig.S9a**) or cryoDRGN (**Fig.S9b**) suggest the improvement brought by OPUS-DSD over that of cryoDRGN. Second, the more normal-looking unmasked FSC for OPUS-DSD’s result (**Fig.6f**) displays clear improvements over that of cryoDRGN (**Fig.6h**). Specifically, the unmasked FSC of OPUS-DSD’s result drops to 0.143 at 7.0 Å, and hovers at around 0.1 from 7.0 Å to 5.1 Å (**Fig.6f**), while that of cryoDRGN’s result drops to 0.143 at 7.5 Å (**Fig.6h**), and to nearly zero at about 6.0 Å. Third, the reconstructed density map from OPUS-DSD (**Fig.6f**) did contain more structural details than that from cryoDRGN (**Fig.6h**). The same is true for comparison of OPUS-DSD (**Fig.S11a**) vs cryoDRGN (**Fig.S11b**), and **Fig.S10e** vs **Fig.S10f** in focused refinement of NEXT complex, and **Fig.S12e** vs **Fig.S12f** with 84k particles of NEXT complex. Additionally, NEXT complex is an extremely dynamic complex. The dips in the FSCs for the NEXT complex are not only seen in the analysis of our own data, but also in a different dataset of NEXT complex by Puno et al. (**Fig.S6** in Puno MR, Lima CD. Structural basis for RNA surveillance by the human nuclear exosome targeting (NEXT) complex. *Cell*. 2022;185(12):2132-47. e26).

3. As in other methods, the decoder presented here creates volumes directly from latent space variables. However, no examples are provided either as an image, or as movies traversing the space. What are the limitations of these volumes, and what is their quality? E.g. as I understand it, the geometry of the network is fixed. How high resolution can reconstructions be made directly from the latent space, is it possible to obtain high resolution in the generated maps? it would be nice to have some discussion of this and to show some examples.

Response: Thank you for pointing out our inadequacy! We added movies for the pre-catalytic spliceosome, the Pf80s ribosome and the S.c80s ribosome by traversing along the first principal component of the latent space (**Mov.1~4**). The resolutions of those reconstructions are comparable to the reconstructions produced during the 3D refinement of Relion. For spliceosome, we showed the comparisons among the structure output by the decoder of OPUS-DSD, the multi-body refined subcomplexes, and the consensus model refined by Relion in **Fig.S2**. They overall agree quite well with each other. The resolutions of the subcomplexes are not as high as those obtained by multi-body refinement since OPUS-DSD reconstructs them

without knowing their poses. However, the movies generated by OPUS-DSD exhibit much higher resolutions than those from cryoDRGN or e2gmm.

We also compared the density map reconstructed by OPUS-DSD to the consensus model (run_class001.mrc) for S.c80S ribosome (**Fig.4d**). The density map reconstructed by OPUS-DSD is of similar resolution as the consensus model, but the OPUS-DSD's density map shows a more complete structure for 40S subunit.

4. The method included regularization to push different volumes to be separated in latent space, and similar volumes to be grouped together. My impression is that this leads to better clustering in latent space, and is likely beneficial for separating defined states. How does this method affect more continuous motions where defined clusters should not exist, does it push them into clusters, or do they're main as a continuum?

Response: This is a great question. In essence, the smoothness of latent space is critical to both resolving continuous dynamics and clustering. In fact, lots of efforts in OPUS-DSD are devoted to improving the smoothness of latent space, including the variational autoencoder (VAE, also used by cryoDRGN) and a disentanglement prior with balanced repulsive forces and attractive forces. Hence, OPUS-DSD does not aggressively compress particles around a cluster into a very small dot, but strives to uncover the underlying patterns within each cluster. This effect is evident in the case of the synthetic data of NEXT with SNR=0.1 (**Fig.S8** in the manuscript). Although both OPUS-DSD and cryoDRGN produce distinct clusters, the UMAP of latent space learned by OPUS-DSD shows much finer patterns within each cluster compared to that by cryoDRGN. Furthermore, the movies output by traversing the latent spaces learned by OPUS-DSD clearly show continuous dynamics. Therefore, OPUS-DSD appears to perform well on both clustering and resolving continuous dynamics. Based on these results, if the cluster doesn't exist and the distribution of conformation is uniform, OPUS-DSD is expected to maintain the uniform distribution. However, for such a dataset with large-scale uniform heterogeneities, we suspect that it can only be resolved to poor resolutions.

5. "To make the network to be invariant to the defocus variation, the network uses input image with its contrast and brightness randomly adjusted to reconstruct the original image." More information should be provided on this, how is it done exactly.

Response: Thank you for the wonderful suggestion. We added detailed explanation in the revised manuscript (Page 26, section title "Data augmentation"). As the defocus variation mainly brings about the contrast changes of an image, we introduced this classic data-augmentation technique in deep learning to improve the robustness of neural network to the contrast changes in images.

6. Clusters are identified via k-means clustering and the manuscript states “The informative latent space with well separated clusters should be easily clustered by the simple KMeans clustering algorithm”. Is it possible that the results obtained from cryoDRGN are simply a failure of the clustering, perhaps their latent space is not as well organized for example? Is it possible to find better reconstructions for cryoDRGN by simply calculating 3Ds for various regions of the latent space?

Response: From our observations, the improvements brought about by OPUS-DSD vs, for example, cryoDRGN, are mostly due to the improved accuracy of OPUS-DSD on encoding the structural variations. For instance, on the synthetic dataset with SNR at 0.1, cryoDRGN can classify particles to high accuracies. But the classification accuracies of cryoDRGN quickly deteriorate as the SNR drops to 0.05, whilst OPUS-DSD maintains its classification accuracy under low SNR. Theoretically, in a latent space which captures structural variations well enough, the structure given by the decoder at a cluster center should represent the consensus structure when performing consensus refinement with particles in the cluster. In fact, grouping clusters with similar structures at cluster centers do yield best results.

7. In all examples the latent space dimension is set to 8, is 8 the optimum number for every experiment, a small discussion on this would be appreciated.

Response: Thank you for a great question! For the 2D-to-3D translation problem, according to both the experiments with cryoDRGN and OPUS-DSD, a latent space dimension around 10 is enough. As can be seen in the revised manuscript, we used the latent space dimension of 12 for the pre-catalytic spliceosome and Pf80s ribosome, and 8 for *S.cerevisiae* 80S ribosome and NEXT complex, all generate satisfactory results. Due to the strong regularization effect of the KL divergence in VAE, the impact of the dimension of latent space is actually well regularized. It is hard to overfit a VAE. By setting the latent space dimension to small values, the information passing from encoder to decoder is greatly restricted. The architecture will then be forced to learn more global variations among dataset. A short discussion has been included in Methods under “Hyperparameter settings” in the revised manuscript.

8. The authors should at least mention other neural network based techniques designed to solve similar problems such as the EMAN2GMM technique and 3Dflex.

Response: Thank you for pointing out this! We added descriptions and references to those techniques in Introduction in the revised manuscript. 3Dflex actually fits the displacement fields of a 3D structure instead of a neural network representation for 3D structure.

9. On page 17 there is a reference which simply states “Error! Reference source not found.”

Response: Sorry about this editing issue, which has been corrected in this revision.

Reviewer #2

1. Fig 2b shows the 8 different structural states (a-h) that were used for the simulated dataset. Fig 2d shows the resulting Umap visualization with cluster centers. Would it be possible to add the 8 input structures also to the Umap visualizations? This would show much better how well the clustering captures the actual (ground-truth) structures. In addition or alternatively: It would be good to show rotation/translation plots in Fig 2 similar to Fig 3e. This would allow for a direct comparison of the ground-truth structures (the 8 states) with the states obtained by OPUS-DSD and cryoDRGN.

Response: Thank you very much for these great suggestions! Since adding 8 structures to UMAP may make it over-crowded, we used the rotation-translation plots in the revised manuscript and added the rotation-translation plot of the ground-truth structures to make the structural comparison much clearer (**Fig.5g, 5h** in the revised main manuscript). The rotation-translation plots clearly demonstrate that the conformations recovered by OPUS-DSD are much more closely resembling those of ground-truth structures.

2. Fig. 4e: why are only 5 out of the 10 states shown in this plot?

Response: The other 5 states (in Fig.4e in previous manuscript, which becomes Fig.S10e in this revision) show incomplete densities in the MTR4 domain, thus making it impossible to calculate the rotation and translation of MTR4 in relative to the ZCCH8 domain.

3. Figs. 3f, 4f, 5e: The "masked" and "corrected" FSC curves from the OPUS-DSD look strange (the unmasked is fine), whereas these curves look ok for the corresponding cryoDRGN results. Can this be explained?

Response: Thank you very much for your questions. The "masked" and "corrected" FSC curves look strange for the presence of a severe dip followed by a bump before finally dropping. Multiple lines of evidence suggest that these may represent meaningful improvements in this resolution range due to the reduction of structural variations in the MTR4 domain by OPUS-DSD. First, the unmasked half maps of clusters obtained by heterogeneity analyses of NEXT complex by OPUS-DSD (**Fig.S9a**) or cryoDRGN (**Fig.S9b**) suggest the improvement brought by OPUS-DSD over that of cryoDRGN. Second, the more normal-looking unmasked FSC for OPUS-DSD's result (**Fig.6f**) displays clear improvements over that of cryoDRGN (**Fig.6h**). Specifically, the unmasked FSC of OPUS-DSD's result drops to 0.143 at 7.0 Å, and hovers at

around 0.1 from 7.0 Å to 5.1 Å (**Fig.6f**), while that of cryoDRGN's result drops to 0.143 at 7.5 Å (**Fig.6h**), and to nearly zero at about 6.0 Å. Third, the reconstructed density map from OPUS-DSD (**Fig.6f**) did contain more structural details than that from cryoDRGN (**Fig.6h**). The same is true for comparison of OPUS-DSD (**Fig.S11a**) vs cryoDRGN (**Fig.S11b**), and **Fig.S10f** vs **Fig.S10h** in focused refinement of NEXT complex, and **Fig.S12e** vs **Fig.S12f** with 84k particles of NEXT complex. Additionally, NEXT complex is an extremely dynamic complex. The dips in the FSCs for the NEXT complex are not only seen in the analysis of our own data, but also in a different dataset of NEXT complex by Puno et al. (**Fig.S6** in Puno MR, Lima CD. Structural basis for RNA surveillance by the human nuclear exosome targeting (NEXT) complex. *Cell*. 2022;185(12):2132-47. e26).

4. Figs 2g,h; 3c,d; 4c,d; 5c,d; how was the density threshold chosen when comparing the different maps?

Response: Thank you for the question. The density thresholds are set to make those volumes to display approximately the same structural features. For volumes with similar resolution, setting those thresholds to n times their standard deviations work well.

5. Title Figure 4: “Heterogeneous analyses”, maybe better “Heterogeneity analysis” (since the analysis is not heterogeneous).

Response: Thank you very much for spotting the errors! We renamed those titles accordingly.

Reviewer #3

1. Backprojection of the image to a volume: please explain which numerical approach is used.

Response: The backprojection is performed in real space. For an image of size $N \times N$, we first repeat the image along Z dimension for N times, thus forming a pseudo 3D volume of size $N \times N \times N$. The pseudo 3D volume can be rotated as in projection. Suppose the projection angle of image is θ , and the corresponding rotation matrix is R_θ , we rotate the pseudo 3D volume by the inverse of R_θ . This operation can be completed by a simple 3D volume resampling. The backprojection can be seen as a step to recover the lost 3D pose information in 2D image implicitly. In fact, the Fourier transform of pseudo 3D volume is still a slice of the complete 3D Fourier transform. We believe this can also be done via cryoDRGN's approach by supplementing pose encodings into the encoder.

2. The reference to a Spatial transformer and neural volume, with the citation provided for it, is unclear. Do the authors refer to one of the standard volume rotation computations or to an actual "fancy" transformer? In either case, they need to be more specific. Does the decoder produce a grid that is then rotated, or a more elaborate neural volume?

Response: Thank you for the questions. In this paper spatial transformer refers to the standard volume transformation computations, which was proposed in 2015 by M Jaderberg. Since every volume transformation can be represented by relating indices of source volume with indices of transformed volume, it takes a grid and a volume, and transforms the volume according to the indices specified in the grid. It is more complicated than the affine transformations such as rotation and translation, and is not the transformer proposed in nature language processing. As for "neural volumes", here is the first convolutional architecture to represent complex natural 3D scenes as far as we knew. Actually, the success of this architecture lies in its initialization method. We have experimented using other conventional initialization methods, which were all failed. To sum it up, the decoder based on "neural volumes" just produces a volume, then a grid is computed according to the projection parameters and the "spatial transformer" rotates the volume according to the grid.

3. Data augmentation: what exactly is varied, and how?

Response: Thank you for the question. We added a detailed explanation in the revised version of our manuscript (Page 26, section title "Data augmentation"). To simply put, it consists of a random B-factor sharpening or blurring process, and also simply changes the scale of pixel values, the purpose of which is mainly to improve the contrast of image (most computer vision models do this).

4. There are some unclear references to total variation and LASSO in the paper. Are these implemented or suggested? Where do they fit in? This would mean that there are many different forms of regularization going on. How do these compare or fit in with the other regularization proposed by the authors? How sensitive is the algorithm to the weights on these various regularizations?

Response: Thank you for noting this! These are implemented. During training, the decoder will produce a volume for each input image. We then compute the total variation (TV) and LASSO of this volume, and add them to the training loss. The weights of these priors are fixed for all experiments we have done so far. We found those fixed values works well for all systems we have tested, and they did improve the reconstructions. These priors mainly serve to improve the quality of volume, and are pretty robust to different levels of SNRs of datasets. By setting their weights on a reasonable scale, the volume will not be overly smoothed or too noisy. The original

work of “neural volumes” also leverages similar regularization techniques, but its regularizers are different and more suited for rendering natural objects.

5. Which version of CryoDRGN is used? The current version of cryoDRGN allows pose updates, which the proposed software does not allow, and which the paper seems to ignore.

Response: Thank you for the questions! We used the 1.0.0 version of cryoDRGN throughout since it was the version available when this study started. The pose update is a complicated problem. In the current version of our software, we didn't update pose of each image. The reason is that, although the pose estimated by consensus refinement contain errors, the errors should be small and mostly local. Therefore, we mitigate pose errors using local angular sampling instead. An additional benefit of the 3D volume representation is that it is easy to compute projections at other angles without reevaluating neural network. For an image with given pose, our method produces other projections at the 8 neighboring projection angles of the given pose, and the reconstruction loss is computed between the image and those 8 projections. But it is worth incorporating pose update into the future versions of our method. More discussions are included in the revised manuscript (Page 18, section title “Discussion”; Page 23, section title “Structural disentanglement prior”; Page 28, section title “Training”).

6. Run time (and other resources): the statement about timing for 1000 particles (at what image size? is this a single iteration?) is insufficient and potentially misleading. It also does not provide any comparison to the CryoDRGN benchmark. There can be serious concerns about timing for all the usual reasons, but also since this algorithm uses 3D representations where the CryoDRGN benchmark uses 2D representations. The authors may have found an efficient way to implement the operations, and the optimization may require a small number of epochs/iterations, but this is not obvious. This reviewer recognizes that precise timing reports do not make sense here: there is no obvious point at which to stop the iterations in cryoDRGN (there is a method for stopping, which is not fully explained, available in the current work). However, some information about the time it took to run some of the examples in the paper would be appropriate (and to make the comparison clearer, it might be useful to show to significantly fewer iterations do not yield a good result). This reviewer finds this software potentially useful even if it is found to be significantly slower than CryoDRGN, but the information should be made available to the reader. Surely, if the software is considerably slower, it is more difficult to argue that the software is strictly superior. On a related note, what GPUs are needed to run this software at reasonable image sizes? The 3D grids could require a considerable amount of GPU RAM. The only discussion of hardware here mentions relatively high-end hardware - does the software work adequately with a single GPU? With lower-end GPUs? Again, the software can be useful even if it requires higher-end hardware, but the bar is much higher if the authors claim the strict superiority claimed here. Similarly, RAM/storage might also be an issue if some of the backprojections are stored.

Response: Thank you for these important questions! OPUS-DSD is relatively more computationally expensive in terms of run time and memory requirement compared to cryoDRGN. There are two aspects to consider here. First, OPUS-DSD uses an almost fixed network configuration, and 3D representations throughout. The run time of OPUS-DSD is almost irrelevant to the image size. In contrast, the run time of cryoDRGN depends on the image size. So, if the image is 256x256, the cryoDRGN needs 21 mins to process 100,000 particles on 1 Nvidia V100 GPU using its default 1024x3 network, then the run time of OPUS-DSD is $4/(21/100) = 19$ times of the run time of cryoDRGN. If switching the network of cryoDRGN to 1024x10, the run time of OPUS-DSD is $4/(56.1/100)=7$ times of the run time of cryoDRGN. On the other hand, OPUS-DSD usually converges after around 10 epochs, while cryoDRGN often needs 30~40 epoches to converge. Therefore, in reality, the differences for both methods in run time and memory requirement are drastically smaller. For example, on the S.c80S ribosome, cryoDRGN takes 8.75 hours with 4GPUs to converge, while OPUS-DSD takes 11 hours with 4GPUs to get the result presented in our manuscript. On the synthetic NEXT complex, cryoDRGN takes 3 hours with 4GPUs to converge, while OPUS-DSD takes 8 hours with 4GPUs to get the result presented in our manuscript. As for the amount of GPU RAM, the V100 GPU with 32GB memory can process around 16 images with OPUS-DSD model at the same time. But we can also make the architecture smaller by setting the output volume to a smaller size. For example, if the output volume is of size 192x192x192, then it requires only half of the memory size compared to output volume of size 256x256x256.

On a separate note, when it comes to inference (though it is less relevant), OPUS-DSD is much faster than cryoDRGN since it can output the 3D volume by one pass while cryoDRGN needs to evaluate the volume voxel by voxel.

More descriptions are added under section title “Implementation” in Methods.

7. The authors make some arguments about certain latent space representations (their results) being advantageous over others (cryoDRGN results). What is a “correct” or “good” representation is actually a complicated question. The authors could state certain properties that may be advantageous in some aspects (e.g., some visual separability when a dimensionality reduction algorithm is used downstream, may be preferable from the point of view of a practitioner). They actually provide reasonable motivation around line 126, but these are different from the arguments made in the comparison to cryoDRGN. The comparisons to cryoDRGN are not unreasonable, they just require some additional discussion of motivation.

Response: Thank you for the question. The comparison to cryoDRGN is mainly to demonstrate the robustness of OPUS-DSD under different levels of SNRs. The UMAP visualizations of

latent spaces between OPUS-DSD and cryoDRGN on the same dataset are mostly similar. We added some additional discussion of motivation in the revised manuscript.

8. Can the authors comment on the importance of different components (where they tried)?

There are many statements about the role of these components made in the paper, but most of them are not supported in the paper. At most, the paper demonstrates some overall potential advantages but stops short of showing what these potential advantages could be attributed to. Many of the claims about a particular component contributing in a particular way sound like hypotheses at best. It is difficult to quantify such hypotheses with so many moving parts, but it might be possible to provide evidence for some of these claims with some additional work. Stating so many of these hypotheses as clear undisputable facts is a stretch. Stating these as clear advantages over cryoDRGN does not seem to be supported by evidence in the work, despite the evidence in the experiments for potential advantages over cryoDRGN of the software as a whole. In other words, I wonder which of these components are necessary and the extent to which they help. I don't think that the authors should try every possible combination of parts of CryoDRGN and their software. Instead, they can rephrase some of their hypotheses and add a few small experiments. A couple of experiments that seem plausible without too much work:

- 1). Could the authors present an example with the contrast data augmentation turned off?
- 2). Could the authors turn off the loss function on the latent space embedding? This might throw the latent variable all over the place, so if the authors happen to have an implementation of the classic VAE loss used in cryoDRGN it would be nice to see how that compares (if the implementation takes a lot of work I don't think it is necessary).

Response: Thank you for the great questions! We didn't do many ablation studies since these techniques are mainly theory motivated and the performance of the end product, OPUS-DSD, is satisfactory. Nevertheless, we followed your suggestions and did some ablation studies for the disentanglement prior and the data augmentation process.

The strength of structural disentanglement prior can actually be tuned by specifying a parameter in our program. We studied the effects using synthetic NEXT complex data with SNR=0.05 with or without the disentanglement prior. We found that turning off the structural disentanglement prior tends to decrease the classification accuracies of clustering results, most significantly for Classes 1, 2, 3, 5 and 7 (comparing **Fig.S14a** and **Fig.S14b**).

As for data augmentation, we also did an ablation study using the synthetic NEXT complex data with SNR=0.05. Again, the classification accuracies are slightly dropped in the latent space of OPUS-DSD without data augmentation, most significantly for Classes 2, 3, 5, 6, and 7 (comparing **Fig.S14a** and **Fig.S14c**).

We also found that the presence of data augmentation decreases the validation error and improve generalization on S.c80S ribosome (comparing **Fig.S14d** and **Fig.S14e**).

9. The paper correctly identifies cryoDRGN as a good benchmark in the context of deep learning (although somewhat similar concepts of getting 3D volumes from 2D images can be traced back even to the covariance method). However, the area of continuous heterogeneity (using deep learning methods and other methods) is broad, some of the contributions attributed to cryoDRGN are not unique to it, and it would be appropriate to add some citations about other current and previous work in the area, especially work related to the ideas here (e.g., multicryoGAN in the context of volume representation if applicable, previous discussions of regularization of the volume, etc.). The current presentation of the state of the art is lacking. "By systematically testing on synthetic and real cryo-EM datasets, we demonstrated the superior performance of OPUS-DSD on [sic?] resolving structural heterogeneity, in terms [sic?] of both compositional changes and conformational dynamics, in cryo-EM datasets compared to cryoDRGN. OPUS-DSD can not only provide the necessary structural heterogeneity information to deepen our understanding about [sic?] the dynamics of macromolecular systems, but also improve the resolution of highly flexible system[sic?] by providing a more homogenous[?] dataset." The experiments are nice, and it's appropriate to say that they demonstrate the potential advantages of this work, but this community doesn't have a truly systematic test or even an accepted test. The tests here are encouraging but anecdotal. The statements here should be more modest. The lack of discussion of run time makes this claim even more questionable. Otherwise, while I would suggest revisiting the introduction for better clarity, it conveys much of the motivation for this work without making other outrageous claims.

Response: Thank you for pointing out our inadequacy in citations and claims. MutlicryoGAN is a GAN based method which is notoriously hard to train, especially in the case of cryo-EM images with low SNR. They only provided demonstrations on synthetic dataset. Hence, we primarily used cryoDRGN as a good benchmark. We have revised the manuscript accordingly, including the addition of MutlicryoGAN citation and tone down the claims.

10. "Therefore, the quality of 3D density map can be improved by 2D supervisions with diverse views and similar underlying 3D structures during training." Not a very clear sentence.

Response: Thank you for pointing out this! We mean that the quality of 3D density map at a given latent code depends on the number of supervisions available at that code. Therefore, if more 2D images with the same underlying structures but different views to the same latent code are mapped, the quality of 3D density map will be higher at that point. This is also why the smoothness of latent space is critical to the quality of 3D density map output by decoder. In a smooth latent space, similar 3D structures will have similar latent codes. Hence, images from similar structures will be clustered together to supervise the training of decoder at the corresponding latent codes. The logic of that paragraph was not very clear in the first submission. We revised it extensively in this new manuscript.

11. ``OPUS-DSD also leveraged data augmentation to learn a defocus invariant latent encoding." The augmentation potentially reduces the dependence on contrast, but this does not immediately mean that it reduces the dependence on CTF (which is not uniform overall frequencies), let alone making it ``invariant". Achieving strict mathematical invariance is difficult - I doubt that if the authors ran an experiment, they would see actual invariance. It would be encouraging (and interesting!) if they do see reduced sensitivity. The most that can be said is something like ``mitigation," ``reduced sensitivity," or perhaps a colloquial ``more invariant" - but even this is not really demonstrated in the paper, only hypothesized (Maybe it is better supported by experiments that the authors did not include?). More generally, most mentions of ``invariance" in the paper are too liberal and not supported by evidence in the work. They should be more accurate.

Response: Thank you very much for your advice! We agree that it is hard to achieve strict invariance. We revised those statements according to your suggestion to make them more accurate.

12. ``The neural network in cryoDRGN also relies on an explicit pose encoding to model the 2D projection." This is no longer true for recent versions of CryoDRGN (and CryoDRGN is not the first to demonstrate this feature). The authors go beyond this to say that this dependence on pose is a weakness of cryoDRGN - there is very little in the current paper that addresses this issue. The authors have a clever idea about backprojecting the image to 3D - it is a very pleasant surprise to me to see that this works despite the missing data (it is easier to see in the Fourier domain that the backprojection does not solve the missing data problem than it is in the spatial domain) that goes into the convolutional network, but to say that it solves problems such as wrong pose going into the algorithm is much more than is actually demonstrated in this paper. Maybe I don't understand what the word ``this" means when they say ``This approach may be susceptible to the pose assignment errors": I would say that this issue applies equally to both methods as long as there is no further evidence. There seem to be two issues packed into this: the difference in what is viewed from each pose and the inaccuracy in the pose. It is not clear what point is made about either of these.

Response: Thank you for pointing out the newest update of cryoDRGN and our inadequacy on the argument here. In ``This approach may be susceptible to the pose assignment errors", we referred to the fact that the decoder in cryoDRGN is a function of the pose encoding. Hence, the decoder of cryoDRGN will generate inaccurate reconstruction given inaccurate pose, namely, the second issue (the inaccuracy in the pose) you mentioned here. The pose update might be a remedy to the pose assignment error for cryoDRGN if it can iteratively improve the accuracy of pose assignment during training. On the other hand, in OPUS-DSD, its 3D volume representation allows us to design another way to treat pose assignment error. We can perform

local angular sampling with the 3D volume presentation very efficiently. Since every pass of OPUS-DSD will generate a 3D volume, we can project this volume along a few more different angles around the given pose. We then train OPUS-DSD by fitting this ensemble of reconstructions w.r.t the experimental images. This approach should also increase the quality of reconstruction in the presence of pose assignment error.

13. ``_These_ features of cryoDRGN yield unsatisfactory 3D density maps on datasets with low SNRs." Maybe I don't understand this correctly, but weren't the experiments with variable SNR synthetic experiments where the ground truth viewing direction is known? In that case, how is the pose inaccuracy relevant?

Response: Thank you for your question to allow us to clarify. Even in the synthetic experiments, we first did a consensus refinement before heterogeneity analysis. The consensus refinement result is then supplemented into heterogeneity analyses. Hence, the viewing direction used in heterogeneity analysis is not ground truth. The settings of cryoDRGN on dataset with SNR=0.1 did show good results. Applying the same architecture of cryoDRGN to the dataset with SNR=0.05, as used in the dataset with SNR=0.1, resulted in worse quality of 3D density map output by cryoDRGN (Comparing Fig.5 and Fig.S7 for SNR=0.05 vs Fig.S8 for SNR=0.1).

14. ``The systematic errors in 3D density maps will propagate into latent space during training and ultimately lead to noninformativelatent encodings, which yields spurious structural heterogeneity resolving results." Not a clear or supported statement. Again, not my most significant concern. ``These architectural drawbacks in cryoDRGN are demonstrated using both synthetic and real datasets in our study here." Not that I have seen. The authors demonstrate some potential advantages but have not shown where they would be traced to. For all I know, it can just be a choice of parameters. Again, I am not saying that the authors cherry-picked, or even that they cannot demonstrate potential with some level of cherry-picking, I am saying that the evidence is not sufficient for such a strong claim. The results are nice enough without statements like these. I would say that the experiments demonstrate something about performance, but I don't think that it's sufficient for a statement about architectural drawbacks.

Response: Thank you for correcting us! Our test cases only represent a specific complex with a specific geometry. We agreed with your comments. We revised the corresponding statement in the new manuscript.

15. ``On the other hand [sic?], OPUS-DSD overcomes these peculiarities by adopting a 3D convolutional architecture [10]." Is this a statement about the encoder or the decoder? It's not clear what contrast is drawn here. ``... Instead of representing the 2D projection using explicit pose encoding, OPUS-DSD reconstructs the 2D projection by directly projecting a 3D volume." I assume that this is about the encoder. It's a clever idea, but I don't see how it overcomes issues

such as misaligned poses without further evidence. As far as I can tell, it is as likely to improve things as it is likely to make them worse (them is aligned pose is now fed into the encoder, not only the decoder). Something seems to work nicely here in practice in this software, but the claims are too strong and not supported with evidence. Saying ``Instead of representing the 2D projection using explicit pose encoding, OPUS-DSD reconstructs...'' makes it sound like CryoDRGN encodes the pose from the image (it doesn't) or perhaps that this encoder does not consider the pose or is invariant to it. In fact, technically speaking, the CryoDRGN encoder is ``invariant'' to the inaccurate pose input (it does not take in the pose as an input variable), whereas the encoder in this work is not invariant (it backprojects using the pose). This does not make either of them better or worse, it is just very unclear what the authors meant to say here. I think I can see where the authors are going with this, but it's not what is written in the paper.

Response: So sorry for the confusion! The following are the responses to each question.

“On the other hand, OPUS-DSD overcomes these peculiarities by adopting a 3D convolutional architecture [10].” This is a statement about the whole architecture since both encoder and decoder are convolutional and in 3D.

“Instead of representing the 2D projection using explicit pose encoding, OPUS-DSD reconstructs the 2D projection by directly projecting a 3D volume.” This statement was referred to decoder. Saying ``Instead of representing the 2D projection using explicit pose encoding, OPUS-DSD reconstructs...'' makes it sound like CryoDRGN encodes the pose from the image. Here we referred to the fact that the decoder of cryoDRGN leverages explicit pose encoding to represent the 2D slice in Fourier space. Therefore, the pose errors in consensus refinement have adverse effect on the reconstruction of cryoDRGN. But OPUS-DSD reconstructs a whole 3D volume and projects it according to a given pose. Hence, it doesn't require such kind of positional encodings as cryoDRGN.

Saying ``Instead of representing the 2D projection using explicit pose encoding, OPUS-DSD reconstructs...'' makes it sound like CryoDRGN encodes the pose from the image (it doesn't) or perhaps that this encoder does not consider the pose or is invariant to it. Pose invariant refers to property that the encoder generates the same encodings for the 2D projections from the same 3D structure.

“the CryoDRGN encoder is ``invariant'' to the inaccurate pose input (it does not take in the pose as an input variable), whereas the encoder in this work is not invariant (it backprojects using the pose).” We thought that though the encoder of cryoDRGN doesn't take in the pose as an input variable, the encoder of cryoDRGN is not invariant to the pose variations since the pose information is implicitly presented in the 2D image (the 2D image is a function of both 3D structure and projection pose, $X(V, P)$). Unless the encoder is trained by 2D images with known labels for each 3D structure, it is hard to achieve pose invariance (this argument is true for both methods). In practice, both methods leverage the encoder-decoder architecture. Since both

decoders in cryoDRGN and OPUS-DSD model the pose variation by reconstructing a 2D projection with a given pose, the decoders can be regarded as pose invariant. The encoders of both methods are then coupled with pose invariant decoders. But the encoder of OPUS-DSD still has certain advantages. OPUS-DSD supplements the pose information to encoder using the backprojection process. The pose information is not 100% accurate but still useful. Plus, the pose assignment errors are mainly small (otherwise the 3D consensus refinement cannot converge). Hence, the encoder of OPUS-DSD is easier to train to encode the structural information by incorporating the pose information into 2D input (in practice, we found that our network can often generate good reconstruction with less epochs than cryoDRGN). Using the notation proposed in the Structural disentanglement prior section in our manuscript, the encoder of cryoDRGN is a function of the form $f(X(V, P))$, while the encoder of OPUS-DSD is a function of the form $f(X(V, P), P)$. Hence, the advantage of the encoder of OPUS-DSD is that it can learn a function to account for the variation associated with pose.

Lastly, we agree that this paragraph is unclear in its original writing. We have revised it in our new manuscript.

16. "This convolutional architecture appears to be robust to high level of noises [sic?] and incorrect pose assignments in 2D cryo-EM images." Where is this shown? I think that it is fair to say that the software performs well in high noise (maybe robust is ok), but I am not sure where there is evidence for robustness to incorrect poses. Furthermore, the more recent CryoDRGN updates pose estimates, an issue that is completely ignored here. My inclination would actually be to attribute the SNR performance to the decoder (implicit regularization), not the encoder. Where is the role of this particular component of the encoder shown? The authors might want to point to something specific such as quality of class assignment as some potential evidence for the argument about SNR.

Response: Thank you for your question and suggestions! We agree with your suggestion that the improvements brought about by OPUS-DSD are likely the consequence of the improved quality of class assignment. On the other hand, since our comparison was based on cryoDRGN version 1.0.0, the pose update function in the recent version was not considered. However, we mentioned this feature of cryoDRGN and incorporated corresponding changes in the revised manuscript.

17. "It also focuses on capturing large-scale structural variations while being resilient to the irrelevant high-resolution structural information and noises." Unclear. Is the same point repeated?

Response: Thank you for pointing out this! We simplified this sentence in the new manuscript.

18. "An advantage of this approach is that it allows one to further improve the fidelity of the 3D density maps by using the traditional sparseness and smoothness regularizers such as total variation and LASSO (15, 23, 24)." Not clear. Are we talking about the regularizer now? Are these implemented? Where?

Response: Sorry for the confusion! Yes, they are implemented in OPUS-DSD. The description was in the "Training objective" subsection in the Methods section. To clarify, we further added descriptions in the Discussion in the revised manuscript.

19. "Moreover, OPUS-DSD employs a validation set to objectively monitoring [sic] the neural network training process, thus further reducing the problem of overfitting nonstructural heterogeneity." Mentioned only later in the paper, and not really explained, as far as I can see. It is difficult to say if the validation set really validates, but it would be very interesting and useful even if it only gives an indication of when to stop.

Response: Thank for your comments! We agree that the validation set is a good indicator of when to stop. To help better describe the validation set, we have expanded the description in the "Training" subsection in Methods in this revised manuscript.

20. "In such cases, rigid-body analysis such as multibody refinement (6) would be less efficient". This statement might be interpreted as a unique feature of this software, which it is not. It applies to cryoDRGN and many other approaches to continuous heterogeneity. Since a comparison to multi-body refinement isn't presented here, I suggest that plausible statements like these would be phased as "likely", "potentially" etc. or rely on a citation of some other papers. "... More importantly, OPUS-DSD has the advantage of exploring conformational and compositional changes in a unified framework. "Again, a reader might mistakenly think that this is a new property of this software. It applies to many (but not all) of the approaches to continuous heterogeneity." In conclusion, OPUS-DSD is a robust, accurate and versatile tool for resolving structural variations. It can not only deepen our understandings about [sic?] the dynamics of biological systems but also facilitate structure determination by providing more homogenous datasets to achieve higher resolutions. "Again, it is too early to say these things based on anecdotal. I might agree that it is promising. What are "more homogenous datasets" ?

Response: Thank you for your questions and the opportunity to correct and clarify! We revised these statements extensively in our new manuscript.

21. "The decoder reconstructs a 3D volume according to the latent code. Total variation and LASSO priors are imposed on the reconstructed 3D volume to encourage its smoothness and sparseness (23, 24)". Please explain where these come in.

Response: Thank you for your questions! The decoder of OPUS-DSD will output a 3D volume for each image. We then compute the total variation and LASSO loss of the 3D volume, and add them to the total loss. We discussed these two terms in the “Training objective” subsection in Methods.

22. “The number of samples for interclass comparison is ” Please explain (term comes up here the first time).

Response: Thank you for pointing out this! We added explanation in the Methods section under Equation 7. “Interclass comparison” refers to the last two term in Equation 7.

23. “The informative latent space with well separated clusters should be easily clustered by the simple KMeans clustering algorithm (34)” Did you verify that the clusters produced were consistent? It's not always the case with kmeans, but it might be the case here.

Response: Thank you for the question! Yes, the clusters produced by Kmeans clustering from both cryoDRGN or OPUS-DSD are consistent with the ground-truth conformations on synthetic data with SNR 0.1 (**Fig.S8**). Thus, kmeans clustering is sufficient to produce accurate classification if the latent space is informative.

24. It might be worthwhile to say that cryoDRGN works in the Fourier domain, so it does not need to explicitly integrate 3D volumes. This is relevant for the computational cost, but it might not be self-explanatory to every reader without further discussion, so it is not crucial to add this.

Response: Thanks! We added more descriptions when comparing computational cost.

Decision Letter, first revision:

Dear Dr. Ma,

Thank you for submitting your revised manuscript "OPUS-DSD: Deep Structural Disentanglement for cryo-EM Single Particle Analysis" (NMETH-A51024A). My apologies for the delay! It has now been seen by the original referees and their comments are below. The reviewers find that the paper has improved in revision, and therefore we'll be happy in principle to publish it in Nature Methods, pending minor revisions to satisfy the referees' final requests and to comply with our editorial and formatting guidelines.

TRANSPARENT PEER REVIEW

Nature Methods offers a transparent peer review option for new original research manuscripts submitted from 17th February 2021. We encourage increased transparency in peer review by publishing the reviewer comments, author rebuttal letters and editorial decision letters if the authors agree. Such peer review material is made available as a supplementary peer review file. Please state in the cover letter 'I wish to participate in transparent peer review' if you want to opt in, or 'I do not wish to participate in transparent peer review' if you don't. Failure to state your preference will result in delays in accepting your manuscript for publication.

ORCID

Sincerely yours,
Allison

Allison Doerr, Ph.D.
Chief Editor
Nature Methods

Reviewer #1 (Remarks to the Author):

The authors did a reasonable job of responding to reviewer comments. The extra examples and results presented considerably improve the manuscript and make it much easier for readers to compare with other methods, as does the addition of movies. I also appreciate that there is a little more detail in the description of the method now. I think the method and paper are interesting and I support publication.

I do believe it is essential the authors should add a statement saying their method is based on cryoDRGN 1.0, and all comparisons in the paper are with cryoDRGN 1.0 (this was a great point mentioned by another reviewer) - this is particularly true as “xxx is clearly better than cryoDRGN” remains prevalent in the paper, but who knows how it compares against the most recent version of cryoDRGN.

I have a couple of comments, which the authors may wish to address, but I do not feel must be addressed for publication :-

1. The strong claims of superiority to cryoDRGN do not seem that toned down to me. This is a shame, as I don't think they are needed - the examples provided are good and readers will be able to appreciate the results without the comment “This is better than cryoDRGN”, which occurs very frequently. In the end, the success of this method, and how it is viewed relative to other groups will be determined by the results it provides for many groups over many datasets. It is for the authors to decide whether to tone it down more, but I think they do themselves a disservice to highlight this so much.

2. I appreciate they have now mentioned the strange FSCs that I and reviewer 2 highlighted in the manuscript. The main gist of their discussion appears to be “yes, they look strange, but we really do think our maps are better than cryoDRGN.” Even if the maps are better, I still feel the FSCs are artifactual. I wonder if they have tried different masking to try and address this? Ultimately, I leave it to the authors to decide whether they wish to publish with these FSCs, but I am sure many people will notice this as we did, and the results may actually be better served by having less artifactual FSCs that report lower resolution.

Reviewer #2 (Remarks to the Author):

My main comment:

"The results from OPUS-DSD are compared to those from the previous published method cryoDRGN. From reading the text, it sounds like cryoDRGN and OPUS-DSD are two separate methods. However, by looking at the code, it becomes clear that OPUS-DSD is based on cryoDRGN and includes substantial amount of code from cryoDRGN. I would therefore consider OPUS-DSD more like an

extension/improvement of cryoDRGN. Therefore, the relationship of OPUS-DSD with cryoDRGN should be made clearer in the main text."

does not seem to have been addressed. I think it is important to clearly state that OPUS-DSD builds on cryoDRGN.

Also the author's explanation of the abnormal shape of the corrected FSC curve for the OPUS-DSD results is not very convincing (new Figs. 6f, S10f, S12e, and paragraph following page 14, line 323). The authors argue that visually comparing the OPUS-DSD density with the cryoDRGN density shows an improvement. However, the question is not whether the density has improved, but why we see this dip. I cannot see how this curve can be caused by an actual improvement in the structure, that for some reason only happens in this specific spatial frequency range. Given that the unmasked curves look ok, there must be something wrong with either the correction procedure or something else.

I am fine with the replies to my other comments.

Reviewer #3 (Remarks to the Author):

The revised manuscript is better. However, the paper could have been more readable. I find it regrettable that some of the more interesting and innovative ideas are buried in a way that makes it more difficult for a reader to appreciate them.

There are various parts that are not very clear and some statements that are not entirely accurate (along the lines of the issues mentioned in my original review), but these are not crucial.

Various items appear without proper context. Among other issues, the paper is not written to be read linearly. This does not seem to be unavoidable. The Discussion is written assuming that the reader has already read the Methods that come after it. One of many easily avoidable examples (not the worst) is the reference to β -VAE, which is mentioned for the first time, without context, in the Discussion (earlier references I found are only to VAE).

The authors added some discussion of previous work in the area. They do mention some of the software that is more readily available to users. While I don't think they need to write a complete survey of the work on heterogeneity, there should be a broader discussion (or reference to review papers).

I am not aware of a consensus on the statement "The direction of using neural network to represent a complete 3D volume pioneered by cryoDRGN should be a more promising approach to uncover complete compositional and conformational heterogeneities in cryo-EM datasets." Although I could

come up with some potential advantages for the representations in the cryoDRGN and current paper. I doubt the authors wish to cover these nuances.

Thank you for providing some clarifications about the use of spatial transformers. I am not sure that the citation provided is sufficient for explaining the role of the spatial transformer when a pose is provided to it as input. I don't think it is a huge leap, but it would be nice to have a small discussion or a reference to spatial transformers in a slightly closer context. The authors seem to state that the spatial transformer is used here as a trainable component, not a predetermined (pose dependent) rotation operator. I can see that this trainable component could have interesting properties in the encoder. I am not sure that it is clarified to the reader of the paper that this is not a simple rotation transformation: for example, I can read "brings the 2D image closer to the corresponding 3D structure" as "rotates," but the operation is more subtle and I think the authors want to argue that it could potentially do something analogous to refining the rotation and maybe even modify the rotated volume to fit some proxy of the target volume. I suspect that there is another nuance here, where this implicit "proxy volume" doesn't actually have to be trivially related to the actual volume since it is only used in the encoder. I do not think that these nuances that are very consistent with the choice of words in the paper. A more careful choice of words and some additional discussion or introduction of spatial transformers in this context could clarify this without covering all these nuances.

It is not very clear if/why this is also a trained component in the decoder.

Please explain how the validation set is used and the procedure for stopping the optimization.

Minor comments

* Abstract: "reducing particle heterogeneity in a given dataset" Please interpret what you mean here. I think it is worth rephrasing at least this part of the abstract.

* The authors make reference to the equivariance of convolution. The authors present a standard proof of the equivariance of (continuous) convolution in Methods (without context or an internal reference, or a statement that this is a standard proof, I believe). In practice, the authors use a CNN, which involves pooling and subsampling. Is the network as a whole still equivariant? This is a technicality, the general idea is clear.

* The paragraph starting with "The success of OPUS-DSD should..." and especially the "comparison" to cryoDRGN here are not very clear. The issue is not the statements. It is the relationship between them. However, this seems to be harmless.

- * $R^{|z|}$ does not seem to be a great mathematical notation for saying $z \in R^d$, although it might be interpreted correctly by a reader.
- * The authors say that the encoder outputs z , but later also mention a variance σ that also comes out. It might be appropriate to organize these concepts better. It might be appropriate to cite the original work on VAEs (or a later survey) and perhaps point the reader to a convenient overview (while noting that this is a modified VAE) with some general statement about μ , σ and sampling from the distribution being managed like in VAEs, but I think the description in the paper can be a little cleaner.
- * "supervisions" authors might want to consider a different term.
- * "The structural heterogeneity power" - explain.
- * "equivalents to" - Consider rephrasing.
- * Explain the translation-rotation plot: translation and rotation of what, determined how?
- * "OPUS-DSD is expected..." -perhaps say "We expect OPUS-DSD to be..."?
- * "In this paper, the ill-posedness is first alleviated by conditioning the encoder on the projection pose via the back-projection step, which recovers the lost 3D pose information in the 2D image and helps the encoder disentangle the structural heterogeneities from the 2D inputs." I understand what the authors mean, but it is written as if it suggests that the back-projection itself recovers 3D _information_ from 2D (back projection in itself produces something 3D, but can't add any _information_ per se. Consider, for example, the Data processing inequality implication). I can see how this can be debated, but I think this is an easily avoidable technicality.
- * If the code is indeed based on CryoDRGN's code, I think it would be appropriate to acknowledge that in Acknowledgement or even in the body of the paper.

Author Rebuttal, first revision:

Dear Dr. Doerr,

Thank you very much for the further feedbacks on our manuscript! We have followed your suggestions and the suggestions of the reviewers and thoroughly revised the manuscript. Here are the point-to-point responses to all comments.

Reviewer #1

I do believe it is essential the authors should add a statement saying their method is based on cryoDRGN 1.0, and all comparisons in the paper are with cryoDRGN 1.0 (this was a great point mentioned by another reviewer) - this is particularly true as “xxx is clearly better than cryoDRGN” remains prevalent in the paper, but who knows how it compares against the most recent version of cryoDRGN.

Response: Thank you for your suggestion! We have added the statement that all comparisons are with cryoDRGN 1.0 in multiple places in the revised manuscript.

The strong claims of superiority to cryoDRGN do not seem that toned down to me. This is a shame, as I don't think they are needed - the examples provided are good and readers will be able to appreciate the results without the comment “This is better than cryoDRGN”, which occurs very frequently. In the end, the success of this method, and how it is viewed relative to other groups will be determined by the results it provides for many groups over many datasets. It is for the authors to decide whether to tone it down more, but I think they do themselves a disservice to highlight this so much.

Response: Thank you for your suggestion! We have toned down the claims throughout in this revised manuscript.

I appreciate they have now mentioned the strange FSCs that I and reviewer 2 highlighted in the manuscript. The main gist of their discussion appears to be “yes, they look strange, but we really do think our maps are better than cryoDRGN.” Even if the maps are better, I still feel the FSCs are artifactual. I wonder if they have tried different masking to try and address this? Ultimately, I leave it to the authors to decide whether they wish to publish with these FSCs, but I am sure many people will notice this as we did, and the results may actually be better served by having less artifactual FSCs that report lower resolution.

Response: Thank you for pointing out this! After checking cryoSPARC's refinement log, now we are aware that there was an FSC-mask auto-tightening procedure which was enabled by default during the refinement in cryoSPARC. The FSCs presented in the earlier version of our manuscript were obtained with this FSC-mask auto-tightening procedure in cryoSPARC. As you have suggested, the dips in the corrected and masked FSCs appear to be artifactual and related to the FSC-mask auto-tightening. By using the default mask parameters without auto-tightening in cryoSPARC, the dips in these FSCs were

largely reduced or almost disappeared, while the reported resolutions only dropped slightly. The FSC figures (Fig.6a, f, h; Extended Data Fig.9f, 9h; and Extended Data Fig.10a, e, f, g) and corresponding resolution values are all updated in this revision. We have also removed the discussion about the dips in the manuscript.

Reviewer #2

"The results from OPUS-DSD are compared to those from the previous published method cryoDRGN. From reading the text, it sounds like cryoDRGN and OPUS-DSD are two separate methods. However, by looking at the code, it becomes clear that OPUS-DSD is based on cryoDRGN and includes substantial amount of code from cryoDRGN. I would therefore consider OPUS-DSD more like an extension/improvement of cryoDRGN. Therefore, the relationship of OPUS-DSD with cryoDRGN should be made clearer in the main text." does not seem to have been addressed. I think it is important to clearly state that OPUS-DSD builds on cryoDRGN.

Response: Thank you for your suggestion! We added the statement in multiple places in this revision. We also acknowledged the adoption of cryoDRGN's code in the Acknowledgements section.

Also the author's explanation of the abnormal shape of the corrected FSC curve for the OPUS-DSD results is not very convincing (new Figs. 6f, S10f, S12e, and paragraph following page 14, line 323). The authors argue that visually comparing the OPUS-DSD density with the cryoDRGN density shows an improvement. However, the question is not whether the density has improved, but why we see this dip. I cannot see how this curve can be caused by an actual improvement in the structure, that for some reason only happens in this specific spatial frequency range. Given that the unmasked curves look ok, there must be something wrong with either the correction procedure or something else.

Response: Thank you for pointing out this! After checking cryoSPARC's refinement log, now we are aware that there was an FSC-mask auto-tightening procedure which was enabled by default during the refinement in cryoSPARC. The FSCs presented in the earlier version of our manuscript were obtained with this FSC-mask auto-tightening procedure in cryoSPARC. As you have suggested, the dips in the corrected and masked FSCs appear to be artifactual and related to the FSC-mask auto-tightening. By using the default mask parameters without auto-tightening in cryoSPARC, the dips in these FSCs were largely reduced or almost disappeared, while the reported resolutions only dropped slightly. The FSC figures (Fig.6a, f, h; Extended Data Fig.9f, 9h; and Extended Data Fig.10a, e, f, g) and corresponding resolution values are all updated in this revision. We have also removed the discussion about the dips in the manuscript.

Reviewer #3

The revised manuscript is better. However, the paper could have been more readable. I find it regrettable that some of the more interesting and innovative ideas are buried in a way that makes it more difficult for a reader to appreciate them.

There are various parts that are not very clear and some statements that are not entirely accurate (along the lines of the issues mentioned in my original review), but these are not crucial.

Various items appear without proper context. Among other issues, the paper is not written to be read linearly. This does not seem to be unavoidable. The Discussion is written assuming that the reader has already read the Methods that come after it. One of many easily avoidable examples (not the worst) is the reference to β -VAE, which is mentioned for the first time, without context, in the Discussion (earlier references I found are only to VAE).

Response: Thank you for your advice! We have thoroughly revised the manuscript in the hope of improving the overall readability. In addition, we also introduced VAE and β -VAE in the Introduction, and referred reader to Training objectives subsection for detailed explanations.

The authors added some discussion of previous work in the area. They do mention some of the software that is more readily available to users. While I don't think they need to write a complete survey of the work on heterogeneity, there should be a broader discussion (or reference to review papers).

I am not aware of a consensus on the statement "The direction of using neural network to represent a complete 3D volume pioneered by cryoDRGN should be a more promising approach to uncover complete compositional and conformational heterogeneities in cryo-EM datasets." Although I could come up with some potential advantages for the representations in the cryoDRGN and current paper. I doubt the authors wish to cover these nuances.

Response: Thanks for your advice! The said statement was removed in this revision.

Thank you for providing some clarifications about the use of spatial transformers. I am not sure that the citation provided is sufficient for explaining the role of the spatial transformer when a pose is provided to it as input. I don't think it is a huge leap, but it would be nice to have a small discussion or a reference to spatial transformers in a slightly closer context. The authors seem to state that the spatial transformer is used here as a trainable component, not a predetermined (pose dependent) rotation operator. I can see that this trainable component could have interesting properties in the encoder. I am not sure that it is clarified to the reader of the paper that this is not a simple rotation transformation: for example, I can read "brings the 2D image closer to the corresponding 3D structure" as "rotates," but the operation is more subtle and I think the authors want to argue that it could potentially do something analogous to refining the rotation and maybe even modify the rotated volume to fit some proxy of the target volume. I suspect that there is another nuance here, where this implicit "proxy volume" doesn't actually have to be trivially related to the actual volume since it is only used in the encoder. I do not

think that these nuances that are very consistent with the choice of words in the paper. A more careful choice of words and some additional discussion or introduction of spatial transformers in this context could clarify this without covering all these nuances.

It is not very clear if/why this is also a trained component in the decoder.

Response: Thanks for your advice! Spatial transformer is a very general module which can cover lots of transformations. In our paper, the spatial transformer is not a trained component in the encoder or decoder. We simply used it to perform the rotation transformation with predetermined pose here. We added a short introduction about the spatial transformer to clarify this.

Please explain how the validation set is used and the procedure for stopping the optimization.

Response: Thank you for pointing out this! After each training epoch ended, the images reserved in the validation set was sent to the neural networks, and went through the whole forward process to compute average reconstruction error. Note that the validation set was used for evaluating the reconstruction errors only, and did not participate in the backward passes to update the parameters of neural network. We set to terminate the optimization if the average reconstruction error on validation set stopped decreasing in 5 consecutive epochs.

Abstract: ``reducing particle heterogeneity in a given dataset" Please interpret what you mean here. I think it is worth rephrasing at least this part of the abstract.

Response: Thank you for pointing out this problem! We intended to stress that our method can cluster particles with similar conformations together but this phrase was unclear. We revised the abstract extensively to make it more precise and clearer.

The authors make reference to the equivariance of convolution. The authors present a standard proof of the equivariance of (continuous) convolution in Methods (without context or an internal reference, or a statement that this is a standard proof, I believe). In practice, the authors use a CNN, which involves pooling and subsampling. Is the network as a whole still equivariant? This is a technicality, the general idea is clear.

Response: Thank you for your advice! Yes, it is a standard proof in many literatures. We removed it to shorten the methods section. We didn't use pooling in our network but did use subsampling. The subsampling will preserve the spatial arrangement of activations. Since the encoder consists of stacks of convolutional layers which are translational equivariant, the encoder should be translational equivariant. Since the input is projected from a 2D image into a low-dimensional vector, the equivariance holds for more global translations.

The paragraph starting with "The success of OPUS-DSD should..." and especially the "comparison" to cryoDRGN here are not very clear. The issue is not the statements. It is the relationship between them. However, this seems to be harmless.

Response: Thank you for pointing out this! We revised this paragraph.

$R^{\{|z|\}}$ does not seem to be a great mathematical notation for saying $z \in R^d$, although it might be interpreted correctly by a reader.

Response: Thank you for your great suggestion! We revised this notation for z .

The authors say that the encoder outputs z , but later also mention a variance σ that also comes out. It might be appropriate to organize these concepts better. It might be appropriate to cite the original work on VAEs (or a later survey) and perhaps point the reader to a convenient overview (while noting that this is a modified VAE) with some general statement about μ , σ and sampling from the distribution being managed like in VAEs, but I think the description in the paper can be a little cleaner.

Response: Thank you for this great advice! We introduced VAE in Introduction, and directly refers to the fact that encoder estimates the distribution of latent code in this part. This should make the description of VAE in our manuscript cleaner.

"supervisions" authors might want to consider a different term.

Response: Thank you for your advice! We dropped this term and used different terms such as 'images' in this revision.

"The structural heterogeneity power" - explain.

Response: Thank you pointing out this error! The correct phrase should be 'the resolving power for structural heterogeneity'. We intended to mention the ability to resolving structural heterogeneity. It has been corrected in this revision.

* "equivalents to" - Consider rephrasing.

Response: Thanks for your advice! We rephrased those sentences involving this term in the revised manuscript.

Explain the translation-rotation plot: translation and rotation of what, determined how?

Response: Thank you for the question! For a conformation, we split it into a fixed domain and a moving domain. The conformational change is then defined as the inter-domain movement between the moving domain and the fixed domain. For example, in this paper, the ZCCHC8 is denoted as the fixed domain, while the MTR4 head is denoted as the moving domain. The rotation-translation plot shows the rotation and translation of MTR4 in relative to ZCCHC8 in a conformational change by comparing to the reference conformation. We added more details of the translation-rotation plot in the legends of Figure 5 and 6 in the manuscript.

* ``OPUS-DSD is expected...'' -perhaps say ``We expect OPUS-DSD to be...''?

Response: Thank you for your advice! We revised it accordingly.

* ``In this paper, the ill-posedness is first alleviated by conditioning the encoder on the projection pose via the back-projection step, which recovers the lost 3D pose information in the 2D image and helps the encoder disentangle the structural heterogeneities from the 2D inputs.'' I understand what the authors mean, but it is written as if it suggests that the back-projection itself recovers 3D _information_ from 2D (back projection in itself produces something 3D, but can't add any _information_ per se. Consider, for example, the Data processing inequality implication). I can see how this can be debated, but I think this is an easily avoidable technicality.

Response: Thank you for pointing out this point, especially the data processing inequality! We revised the manuscript accordingly.

If the code is indeed based on CryoDRGN's code, I think it would be appropriate to acknowledge that in Acknowledgement or even in the body of the paper.

Response: That's a great advice! We acknowledged cryoDRGN's work in both the main text and the Acknowledgements section of this revision!

Final Decision Letter:

Dear Jianpeng,

I am pleased to inform you that your Article, "OPUS-DSD: Deep Structural Disentanglement for cryo-EM Single Particle Analysis", has now been accepted for publication in Nature Methods. Your paper is tentatively scheduled for publication in our November print issue, and will be published online prior to that. The received and accepted dates will be 22 November 2022 and 24 August 2023. This note is

intended to let you know what to expect from us over the next month or so, and to let you know where to address any further questions.

Over the next few weeks, your paper will be copyedited to ensure that it conforms to Nature Methods style. Once your paper is typeset, you will receive an email with a link to choose the appropriate publishing options for your paper and our Author Services team will be in touch regarding any additional information that may be required.

You will receive a link to your electronic proof via email with a request to make any corrections within 48 hours. If, when you receive your proof, you cannot meet this deadline, please inform us at rjsproduction@springernature.com immediately.

Please note that *Nature Methods* is a Transformative Journal (TJ). Authors may publish their research with us through the traditional subscription access route or make their paper immediately open access through payment of an article-processing charge (APC). Authors will not be required to make a final decision about access to their article until it has been accepted. [Find out more about Transformative Journals](https://www.springernature.com/gp/open-research/transformative-journals)

Your paper will now be copyedited to ensure that it conforms to Nature Methods style. Once proofs are generated, they will be sent to you electronically and you will be asked to send a corrected version within 24 hours. It is extremely important that you let us know now whether you will be difficult to contact over the next month. If this is the case, we ask that you send us the contact information (email, phone and fax) of someone who will be able to check the proofs and deal with any last-minute problems.

If, when you receive your proof, you cannot meet the deadline, please inform us at rjsproduction@springernature.com immediately.

Once your manuscript is typeset and you have completed the appropriate grant of rights, you will receive a link to your electronic proof via email with a request to make any corrections within 48 hours. If, when you receive your proof, you cannot meet this deadline, please inform us at rjsproduction@springernature.com immediately.

Once your paper has been scheduled for online publication, the Nature press office will be in touch to confirm the details.

Once your paper has been scheduled for online publication, the Nature press office will be in touch to confirm the details.

Content is published online weekly on Mondays and Thursdays, and the embargo is set at 16:00 London time (GMT)/11:00 am US Eastern time (EST) on the day of publication. If you need to know the exact publication date or when the news embargo will be lifted, please contact our press office after you have submitted your proof corrections. Now is the time to inform your Public Relations or Press Office about your paper, as they might be interested in promoting its publication. This will allow them time to prepare an accurate and satisfactory press release. Include your manuscript tracking number NMETH-A51024B and the name of the journal, which they will need when they contact our office.

About one week before your paper is published online, we shall be distributing a press release to news organizations worldwide, which may include details of your work. We are happy for your institution or funding agency to prepare its own press release, but it must mention the embargo date and Nature Methods. Our Press Office will contact you closer to the time of publication, but if you or your Press Office have any inquiries in the meantime, please contact press@nature.com.

Nature Portfolio journals [encourage authors to share their step-by-step experimental protocols](https://www.nature.com/nature-research/editorial-policies/reporting-standards#protocols) on a protocol sharing platform of their choice. Nature Portfolio 's Protocol Exchange is a free-to-use and open resource for protocols; protocols deposited in Protocol Exchange are citable and can be linked from the published article. More details can found at www.nature.com/protocolexchange/about.

Best regards,
Allison

Allison Doerr, Ph.D.
Chief Editor
Nature Methods